Resource

# Pan-cancer profiling of tumor-infiltrating natural killer cells through transcriptional reference mapping

Herman Netskar [1,2,13], Aline Pfefferle [3,13] ✉, Jodie P. Goodridge[4], Ebba Sohlberg [3], Olli Dufva[5], Sarah A. Teichmann [6,7], Demi Brownlie [8], Jakob Michaëlsson[3], Nicole Marquardt [8], Trevor Clancy [9,10], Amir Horowitz[11,12,14] & Karl-Johan Malmberg [1,2,3,14] ✉

The functional diversity of natural killer (NK) cell repertoires stems from differentiation, homeostatic, receptor–ligand interactions and adaptive-like responses to viral infections. In the present study, we generated a single-cell transcriptional reference map of healthy human blood- and tissue-derived NK cells, with temporal resolution and fate-specific expression of gene-regulatory networks defining NK cell differentiation. Transfer learning facilitated incorporation of tumor-infiltrating NK cell transcriptomes (39 datasets, 7 solid tumors, 427 patients) into the reference map to analyze tumor microenvironment (TME)-induced perturbations. Of the six functionally distinct NK cell states identified, a dysfunctional stressed CD56$^{bright}$ state susceptible to TME-induced immunosuppression and a cytotoxic TME-resistant effector CD56$^{dim}$ state were commonly enriched across tumor types, the ratio of which was predictive of patient outcome in malignant melanoma and osteosarcoma. This resource may inform the design of new NK cell therapies and can be extended through transfer learning to interrogate new datasets from experimental perturbations or disease conditions.

NK cells are innate lymphocytes that play a vital role in the immune response through their ability to directly kill transformed and virus-infected cells by orchestrating the early phase of the adaptive immune response[1]. NK cells are commonly divided into two functionally distinct subsets: CD56$^{bright}$ and CD56$^{dim}$ NK cells[2,3]. However, this is an oversimplified view of the repertoire. Mass cytometry profiling of NK cell repertoires at the single-cell level revealed an extensive phenotypic diversity comprising up to 100,000 unique subsets in healthy individuals[4]. Much of this diversity is based on combinatorial expression of stochastically expressed, germline-encoded activating and inhibitory receptors that bind to human leukocyte antigen (HLA) class I and tune NK cell function in a process termed NK cell education[5,6]. Another layer of diversity reflects the continuous differentiation through well-defined

intermediate phenotypes from the naive CD56$^{bright}$ NK cells through CD62L$^+$NKG2A$^+$KIR$^-$CD57$^-$CD56$^{dim}$ NK cells to terminally differentiated, adaptive CD62L$^-$NKG2C$^+$CD57$^+$KIR$^+$CD56$^{dim}$ NK cells, associated with past infection with cytomegalovirus (CMV)[7–10]. Given the increasing interest in harnessing the cytolytic potential of NK cells in cell therapy against cancer, it is of fundamental importance to understand the molecular programs and gene-regulatory circuits driving NK cell differentiation and the underlying functional diversification of the human NK cell repertoire.

Utilizing single-cell RNA sequencing (scRNA-seq), Crinier et al. discovered organ-specific signatures in human spleen NK cells and two major transcriptional clusters in blood-derived NK cells (PB-NK), corresponding to CD56$^{dim}$ (NK1) and CD56$^{bright}$ (NK2) NK cell subsets[2].

Bulk RNA and chromatin immunoprecipitation sequencing identified dominant transcription factor (TF) axes defining CD56[bright] (*TCF1-LEF-MYC*) and CD56[dim] (*PRDM1*) phenotypic subsets, respectively[11]. Later research reported additional diversity with unique transcriptional clusters, including interleukin (IL)-2- and type I interferon (IFN)-responding NK cell subsets[12] and an intermediate CD56[dim]GzmK[+] stage, potentially bridging CD56[bright] and CD56[dim] NK cells[13]. A comprehensive analysis unveiled a role for Bcl11b in driving NK cell differentiation toward the adaptive state, reciprocally suppressing early TFs such as *RUNX2* and *ZBTB16* (ref. 14). Combining gene expression analysis, chromatin accessibility and lineage tracing via mitochondrial DNA mutations, Rückert et al. revealed clonal expansions and a distinct inflammatory memory signature in adaptive NK cells[15]. Using a pan-cancer, single-cell atlas approach, Tang et al.[16] identified a tumor-enriched dysfunctional CD56[dim]CD16[hi] NK cell population interacting with LAMP3[+] dendritic cells in the TME. Hence, scRNA-seq and bulk RNA-seq usage have defined major transcriptional regulatory hubs during NK cell differentiation and identified a persistent memory state in human innate immunity. However, it remains unclear how the regulatory gene circuits that operate under homeostasis in healthy tissues are affected by cellular and/or soluble cues in the TME, resulting in perturbed functional states within tumor-infiltrating NK (TiNK) cells.

In the present study we established a single-cell transcriptional reference map that resolves gene expression trends and dominating TF–target interactions during NK cell differentiation in blood and normal tissues. Reference mapping enabled the analysis of cellular differences and gene programs in diseases and various conditions by contextualizing new datasets within a healthy transcriptional reference, facilitating the identification of new states not found in the literature[17]. We utilized our NK cell reference map, compiled from 44,640 PB-NK cells (12 donors) and 27,732 tissue-resident NK (TrNK) cells (136 donors), to query the regulons and functional states, as defined through gene expression signatures, of TiNK cells derived from 427 patients with 7 distinct solid tumors (38,982 TiNKs). We found that TrNK and TiNK cells have a clear tissue-residency signature but still share the dominant regulons of blood CD56[bright] and CD56[dim] NK cells. Of the six functional states identified in our pan-cancer atlas and confirmed in a spatial transcriptomics dataset, a dysfunctional stressed CD56[bright] state susceptible to TME-associated cellular communication and a cytotoxic effector CD56[dim] state were commonly enriched across tumor types. Stratification of patient survival data identified a high ratio of effector CD56[dim] to stressed CD56[bright] state to correlate with improved survival in patients with osteosarcoma and melanoma. This resource provides a granular view of cancer-specific alterations of solid TiNK cells, identifying how the TME can lead to NK cell dysfunction and may inspire new strategies to engineer cell therapy products with robust functional phenotypes resistant to TME-induced suppressive mechanisms.

## Results

### NK cell subset annotation using predictive gene signatures

To establish a pan-cancer atlas of TiNK cells, we first defined NK cell differentiation at the transcriptional level. We performed scRNA-seq of the total NK cell population from seven healthy donors and integrated our transcriptomes with five publicly available donor datasets[2,18] using scVI (single-cell Variational Inference)[19] (Supplementary Table 1). By retaining only cell-to-cell variation independent of sample-to-sample variation, the cells that initially clustered by donor and laboratory origin were successfully integrated into a homogeneous population and visualized using diffusion maps[20] to preserve the continuous trajectories observed with biological differentiation (Fig. 1a). Although NK cell differentiation is best described as a continuum, CD56[bright] and CD56[dim] NK cells represent two distinct stages of differentiation. By performing gene signature scoring using AUCell[21], we identified cells at the top of the diffusion map embedding scoring high for the CD56[bright] gene signature[2], whereas the main body of the embedding exhibited increasing intensity of the CD56[dim] signature[2] (Fig. 1b). Scoring of two independent gene signatures based on the CD56[bright/dim] regulon[11] and proteome[22] confirmed our results (Extended Data Fig. 1a,b).

The relatively large and heterogeneous population of CD56[dim] NK cells is commonly phenotypically defined into functionally distinct subsets based on a selected number of inhibitory and activating receptors contributing to the functional tuning[7]. To identify predictive gene signatures associated with these functional stages encompassing NK cell differentiation, we sorted and sequenced equal numbers of CD56[bright] NK cells and four CD56[dim] NK cell subsets (NKG2A[+]KIR[-]CD57[-], NKG2A[-]self-KIR[+]CD57[-], NKG2A[-]nonself-KIR[+]CD57[-], NKG2A[-]self-KIR[+]CD57[+] or NKG2A[-/+]self-KIR[+]CD57[+]NKG2C[+]) from two donors, one without and one with a large adaptive NK cell expansion (Fig. 1c and Extended Data Fig. 1c,d). Transcriptionally, the adaptive NK cell subset was the most distinct because the remaining CD56[dim] subsets exhibited a high degree of transcriptional overlap, while still ordering themselves along the previously defined maturation scheme (Fig. 1c). As previously observed in bulk RNA-seq data[23], the transcriptomes of self and nonself KIR[+] NK cells were highly similar even at the single-cell level and thus merged for subsequent analysis (Fig. 1c). The five transcriptionally distinct NK cell subsets were renamed to reflect their maturation stage: 'CD56[bright]', 'early CD56[dim]', 'intermediate CD56[dim]', 'late CD56[dim]' and 'adaptive' (Fig. 1c).

We next trained a semi-supervised model, scANVI (single-cell ANnotation using Variational Inference)[24], to leverage our identified NK cell subset gene signatures to predict and infer subset annotation of compiled bulk NK cell scRNA-seq datasets. We first tested the accuracy of the prediction model (M1) on 15% of the subset-sorted NK cells (Fig. 1c) that were not included in the training of the model. Transcriptionally distinct subsets (CD56[bright], adaptive) were annotated with high accuracy, whereas subsets exhibiting higher transcriptional overlap were annotated with slightly reduced accuracy (Fig. 1d). Using this model, we could annotate the total NK cell dataset comprising 23,253 single-cell transcriptomes across 12 donors at the subset level (Fig. 1e). The transcriptional profiles of the subsets are captured by the model and used to identify differentially expressed genes (DEGs). The overlapping sets of genes illustrate the transition between the subsets. (Fig. 1f). To validate our annotation model, we performed unbiased clustering (Leiden) of the total NK cell dataset (12 donors), identifying 5 clusters closely matching our annotated 5 NK cell subsets (Fig. 1g). A small

**Fig. 1 | NK cell differentiation at the transcriptional level. a**, Integration process of scRNA-seq data of NK cells from 12 donors and 4 different laboratories using scVI showing a UMAP based on the scVI latent representation, followed by a UMAP based on the diffusion map components. **b**, AUCell scores of gene signatures for CD56[bright] and CD56[dim] NK cell subsets. **c**, UMAP representation of five sorted subsets from a donor with an adaptive expansion (left) and a donor without an adaptive expansion (right). **d**, Heatmap depicting accuracy of our prediction model for subset annotation tested on the held-out 15% of cells from the subset-specific dataset (two donors). **e**, UMAP of the scANVI representation of both bulk and sorted NK cells, showing original annotation of NK cells (12 donors, left) and subset labels predicted (right) using the scANVI model trained with sorted subset data (2 donors). **f**, Dot plots showing the top three up- and downregulated genes between all pairs of subsets (*x* and *y* axes) as identified by the differential expression module in scANVI. These top genes were then visualized across all NK cell subsets within the differentiation spectrum (*x* axis), to highlight the continuous nature of NK cell differentiation. **g**, Diffusion map representation showing the predicted subset labels for the bulk data (top) and depicting Leiden clustering of the 12 donor NK cell dataset (bottom). **h**, Heatmap showing distribution of our annotated 12 donor NK cell subsets over the 5 Leiden clusters. **i**, Frequency (freq.) of annotated late CD56[dim] and adaptive NK cell subsets in donors with and without an adaptive NK cell expansion. Int., intermediate.

portion of intermediate CD56^dim-annotated NK cells clustered together with late CD56^dim-annotated NK cells in cluster 4 (Fig. 1h), probably corresponding to more mature cells within the population. The subset stratification obtained through training of our model based on subset signatures, as well as the unbiased Leiden clustering, harmonizes well

with the recently proposed NK1–3 nomenclature[25] (Extended Data Fig. 1e). Having confirmed the validity of our five NK cell subsets, M1 was utilized to identify donors with an adaptive NK cell expansion, which were all confirmed to be CMV seropositive (Fig. 1i). Thus, this first scANVI model forms a basis to interrogate cellular states layered

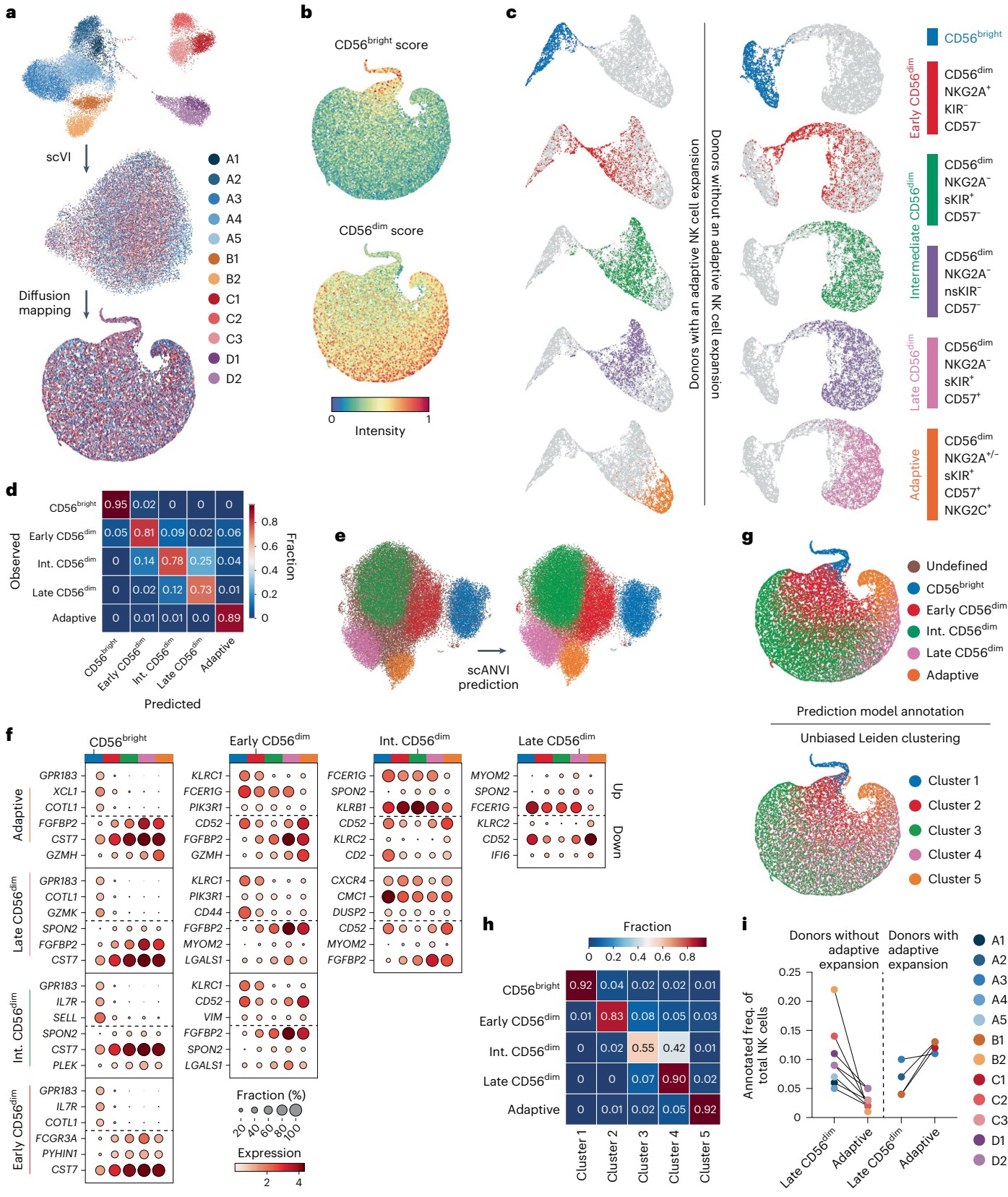

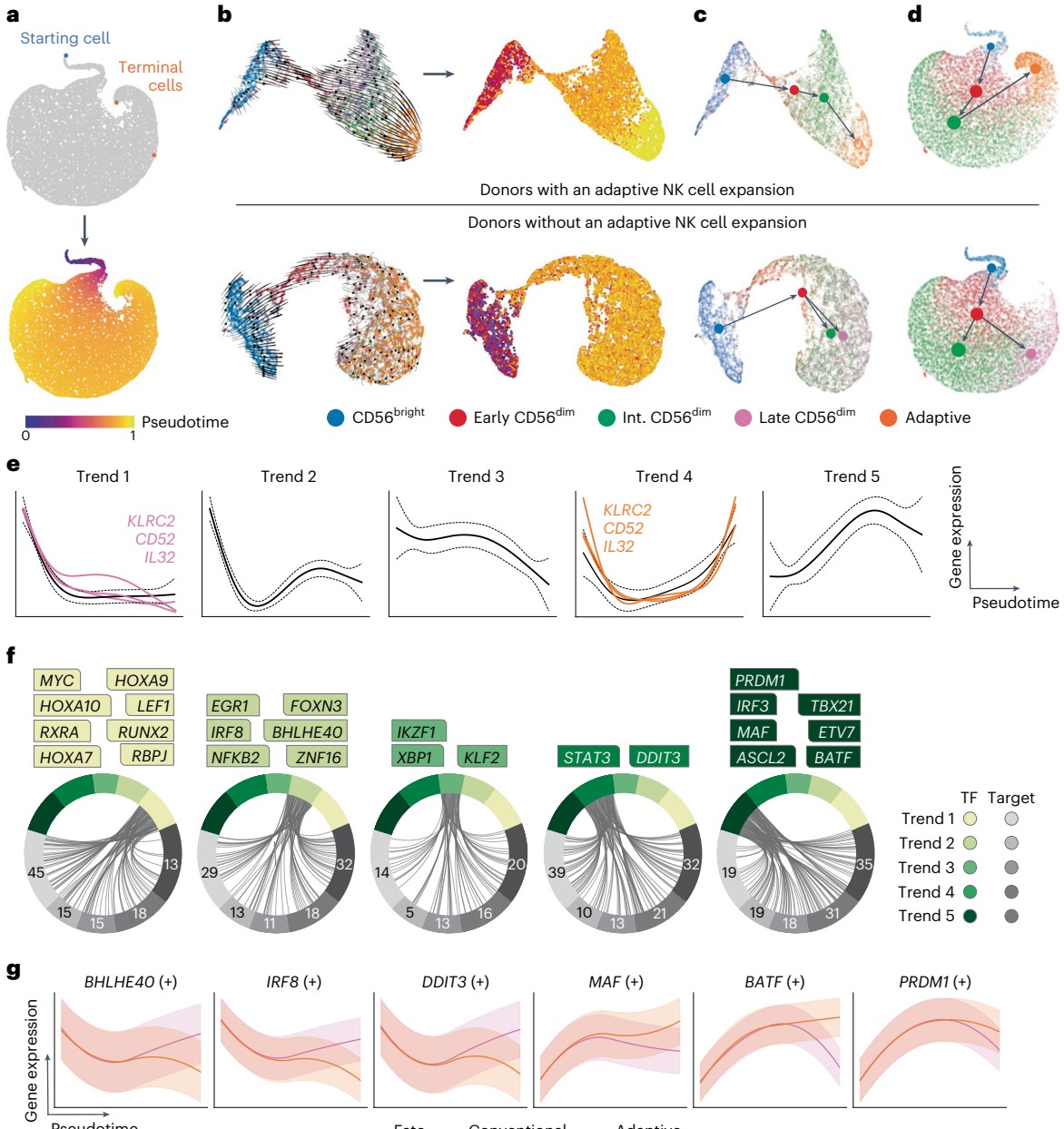

**Fig. 2 | GRNs defining conventional and adaptive NK cell fates. a**, UMAP representation highlighting the starting cell (blue) with the lowest value CD56[dim] signature score and the two terminal cells (orange) as predicted by Palantir. **b**, UMAP representation of the data from the sorted subsets (two donors) showing the RNA velocity vector field as a stream plot and the inferred pseudotime. **c,d**, PAGA graph with directionality and transitions from RNA velocity analysis for the sorted subsets (2 donors) (**c**) and subset-inferred bulk donors (12 donors) stratified based on the presence or absence of adaptive expansion (**d**). **e**, Gene trends clustered into five overall trends of expression along pseudotime, showing expression of *KLRC2*, *CD52* and *IL32* in both terminal fates (pink, conventional fate; orange, adaptive fate). **f**, Inferred GRNs where dominant TFs for each trend are highlighted. **g**, Selection of regulons showing differential expression over pseudotime within the conventional and adaptive fate.

on top of the natural transcriptional changes with NK cell subsets at different stages of differentiation.

## Fate-specific gene-regulatory networks

To decipher the regulatory gene pathways driving NK cell differentiation at the transcriptional level, we used Palantir[26] and RNA velocity to calculate pseudotime[27,28]. Palantir identifies terminal cells based on a chosen starting cell, placing the remaining cells along a timeline (pseudotime). Defining the starting cell (blue) based on the lowest CD56[dim] score[2] (Fig. 1b) identified two terminal cells (orange), predicted to be part of the late CD56[dim] and adaptive population, respectively (Fig. 2a). To validate this trajectory, we utilized the dynamic model

implemented in scVelo[27] to compute RNA velocity (spliced versus unspliced transcripts), inferring pseudotime without a predefined starting cell (Extended Data Fig. 2a,b). The resulting vector field and extrapolated pseudotime confirmed a trajectory starting within the CD56[bright] NK cell subset and terminating in the adaptive subset (Fig. 2b). Last, to infer developmental relationships at the resolution of the five subsets, representing functionally distinct subsets and proposed stages of NK cell differentiation[7], we applied partition-based graph abstraction (PAGA)[29] to quantify their connectivity and estimate transitions. In line with the two terminal fates (late CD56[dim], adaptive) identified by Palantir, we analyzed donors with conventional and adaptive NK cells separately (Fig. 1i). In both types of donors, early CD56[dim] NK

cells formed the connecting link between CD56[bright] and the remaining CD56[dim] populations (Fig. 2c,d). However, although adaptive donor NK cells continued their progression to intermediate CD56[dim] cells, terminating in the transcriptionally distinct adaptive population, conventional donors instead progressed toward intermediate/late CD56[dim] populations (Fig. 2c,d).

Having established a temporal axis to NK cell differentiation, we utilized generalized additive models (GAMs) to compute gene expression trends as a function of time for each gene[26], which clustered into five distinct trends (Fig. 2e). Genes varying in expression across the two terminal fates were depicted in their trends for each fate, exemplified by *KLRC2*, *CD52* (refs. 15,18) and *IL32* clustering into trend 1 in the conventional late CD56[dim] fate and trend 4 in the adaptive fate (Fig. 2e). Based on the two-fate model, we constructed gene-regulatory networks (GRNs)[21] stratified by the five gene trends and identified the dominant TFs across pseudotime and their known downstream target genes (Fig. 2f). Trend 1 is dominated by genes that are downregulated with differentiation from CD56[bright] to CD56[dim] cells, including previously reported TFs (*MYC*, *LEF1*, *RUNX2*)[11], *RBPJ*[30] involved in Notch signaling, the retinoic acid receptor (*RXRA*) and TFs regulating ID2 expression (*HOXA9*, *HOXA10*)[31] (Fig. 2e,f). Trend 2 genes, compared with trend 1, are upregulated during differentiation from early to intermediate CD56[dim] cells and include, among others, *EGR1* (ref. 32) (cell survival, proliferation, apoptosis, regulation of TRAIL expression), *BHLHE40* (refs. 33,34) (associated with NK cell activation and repression of *RXRA*) and *IRF8* (refs. 35,36) (role in orchestrating adaptive response, essential NK cell gene) (Fig. 2e,f). TFs exhibiting less dynamic changes across pseudotime are clustered in trend 3, such as *IKZF1* (ref. 37), *XBP1* and *KLF2*, which play a role in regulating homeostatic proliferation, effector function and cytokine responsiveness[38,39]. TFs exhibiting higher expression at the start and end of pseudotime fall into trend 4, including *STAT3* (cell survival, IFN-γ production) and *DDIT3* (ref. 40) (stress response, metabolism). Last, expression of trend 5 genes steadily increases with differentiation, decreasing only during late differentiation, and includes previously reported TFs associated with CD56[dim] NK cells (*MAF*, *PRDM1*, *TBX21*)[11], the AP-1 family member *BATF*, the ETS family member *ETV7* and the Wnt target gene *ASCL2* (Fig. 2e,f). The TF-based GRNs were further curated to only retain direct targets with significant motif enrichment, referred to as 'regulons' (denoted by '(+)'), expression of which was confirmed in an independent bulk RNA-seq dataset on sorted NK cell subsets. Regulon expression substantially differing between the conventional and adaptive fate includes conventional fate-associated *BHLHE40* (ref. 34), *IRF8* (refs. 35,36) and *DDIT3* (ref. 40) and adaptive fate-associated *MAF*[11], *BATF* and *PRDM1* (ref. 41) regulons (Fig. 2g). Clustering of dominant TFs according to their temporal expression during NK cell differentiation revealed a set of highly connected regulatory circuits, expression of which diverged during terminal differentiation into one of the two cell fates: conventional or adaptive.

### Transfer learning to generate pan-cancer atlas

Having transcriptionally defined NK cell differentiation in peripheral blood (PB), we proceeded to train a second model (M2) with publicly available scRNA-seq datasets encompassing 6 healthy tissues (prostate, lung, pancreas, skin, breast, brain) from a total of 136 donors using scVI[19] to generate a healthy reference map (PB-NK + TrNK) (Fig. 3a and Supplementary Table 2). The tissue-specific datasets were integrated and annotated using scANVI and CellTypist[42] was used to identify immune subsets of interest at the pan-tissue level (Fig. 3b and Extended Data Fig. 3a) and within individual tissues (Extended Data Fig. 3b–f). The annotation and integration steps were repeated for the scRNA-seq datasets from 7 solid tumors (prostate (PRAD), lung (NSCLC), melanoma (SKCM), pancreas (PAAD), breast cancer (BRAC), glioblastoma (GBM) and osteosarcoma (SARC)) from a total of 427 patients (Supplementary Tables 3 and 4), at the pan-cancer level (Fig. 3c,d) and within individual

tumor types (Extended Data Fig. 4a–g). CellTypist-annotated innate lymphoid cells (ILCs) (Extended Data Fig. 5a, b) were further stratified into ILC1/2/3 based on previously described scRNA-seq signatures[43]. We could not identify ILC1s in both the tissue and the tumor datasets, but, importantly, ILC2- and ILC3-annotated cells scored highly for *IL7R* expression compared with CD56[bright]- and CD56[dim]-annotated NK cells, excluding contamination by ILC1s (Extended Data Fig. 5c,d).

To assess tissue-residency status in our annotated NK cells in the tissue- and tumor-derived datasets (Extended Data Fig. 5a,b), we utilized a literature-derived TR signature as well as our own atlas-derived TR (atlas-TR) signature (Fig. 3e). The atlas-TR signature is based on the top six genes differentially expressed by both CD56[bright] and CD56[dim] NK cells across tissue types when comparing with the corresponding subset in the blood-derived NK cells (Extended Data Fig. 5e,f). CD56[bright] NK cells scored generally higher for a TR signature compared with CD56[dim] NK cells in both normal tissue and tumors, with a more distinct TR signal (compared with PB-NK) achieved with the atlas-TR signature (Fig. 3e and Extended Data Fig. 5g). NK cells annotated in a healthy brain scored very low for tissue residency and thus we cannot exclude blood contamination in these samples (Extended Data Fig. 5g).

CD56[bright]- and CD56[dim]-annotated TiNK cells were mapped on to the reference map (PB-NK, TrNK) using transfer learning (scArches[44]) to generate the final model (M3), our pan-cancer NK atlas (Fig. 3f). CD56[bright] and CD56[dim] subsets from PB, tissues and tumors clustered together (Fig. 3g,h) and were more tightly connected than to their respective tissues/tumor origin, apart from skin-/SKCM-derived NK cells (Fig. 3g,h). Thus, differentiation stage had a greater influence on the NK cell transcriptome compared with tissue origin. Transfer learning facilitated incorporation of TiNK cells on to our healthy reference map of PB and TrNK cells, allowing for downstream systematic interrogation of cellular states within solid TiNK cells.

### Altered NK cell subset frequencies across tissues and tumors

The TME is shaped by its cellular composition, in particular by the infiltrating immune cells, which in turn can be modulated by their surroundings. A pan-cancer comparison of the healthy tissue and tumor-annotated immune subtypes (Fig. 3b,d) identified an increased proportion of plasma cells and naive B cells, as well as a decreased proportion of CD56[dim] NK cells, classic monocytes, dendritic cells, NK T cells, and effector memory/effector T helper cells (helper T[EM/EFF]), effector memory/effector memory re-expressing CD45RA cytotoxic T cells (cytotoxic T[EM/EMRA]) and resident memory cytotoxic T cells (cytotoxic T[RM]) in the pan-cancer datasets (Fig. 4a). The fraction of CD56[bright] NK cells out of total immune cells was enriched in BRAC, whereas CD56[dim] NK cells were enriched in SKCM, but decreased in NSCLC and BRAC (Fig. 4a–c). We further annotated the CellTypist-identified NK cells at the subset level using our subset-trained model (M1) (Fig. 4d,e). Skewing of the CD56[bright]:CD56[dim] ratio between healthy blood or tissue and tumor was observed for most tumor types (Fig. 4d,e), including non-small cell lung cancer (NSCLC), which was independently validated by flow cytometry in an NSCLC cohort (Fig. 4f and Extended Data Fig. 6a). In line with this, we observed a general decrease in the intermediate CD56[dim] population within the TiNK cells (Fig. 4d,e). Protein-based annotation of the CD56[dim] population in the NSCLC cohort also identified a decrease of the early and intermediate CD56[dim] subset and a modest increase of the late CD56[dim] subsets in the NSCLC cohort compared with healthy blood controls (Fig. 4g and Extended Data Fig. 6b–e). Solid TiNK cells were enriched for a CD56[bright] transcriptional phenotype whereas intermediate CD56[dim] NK cells were reduced within the CD56[dim] compartment in solid tumors, findings that were verified at the protein level in an NSCLC cohort[45].

### Six functionally distinct cellular states of NK cells

TMEs of solid tumors are hostile and often immunosuppressive environments for immune cells to infiltrate[46]. Understanding how the TME can

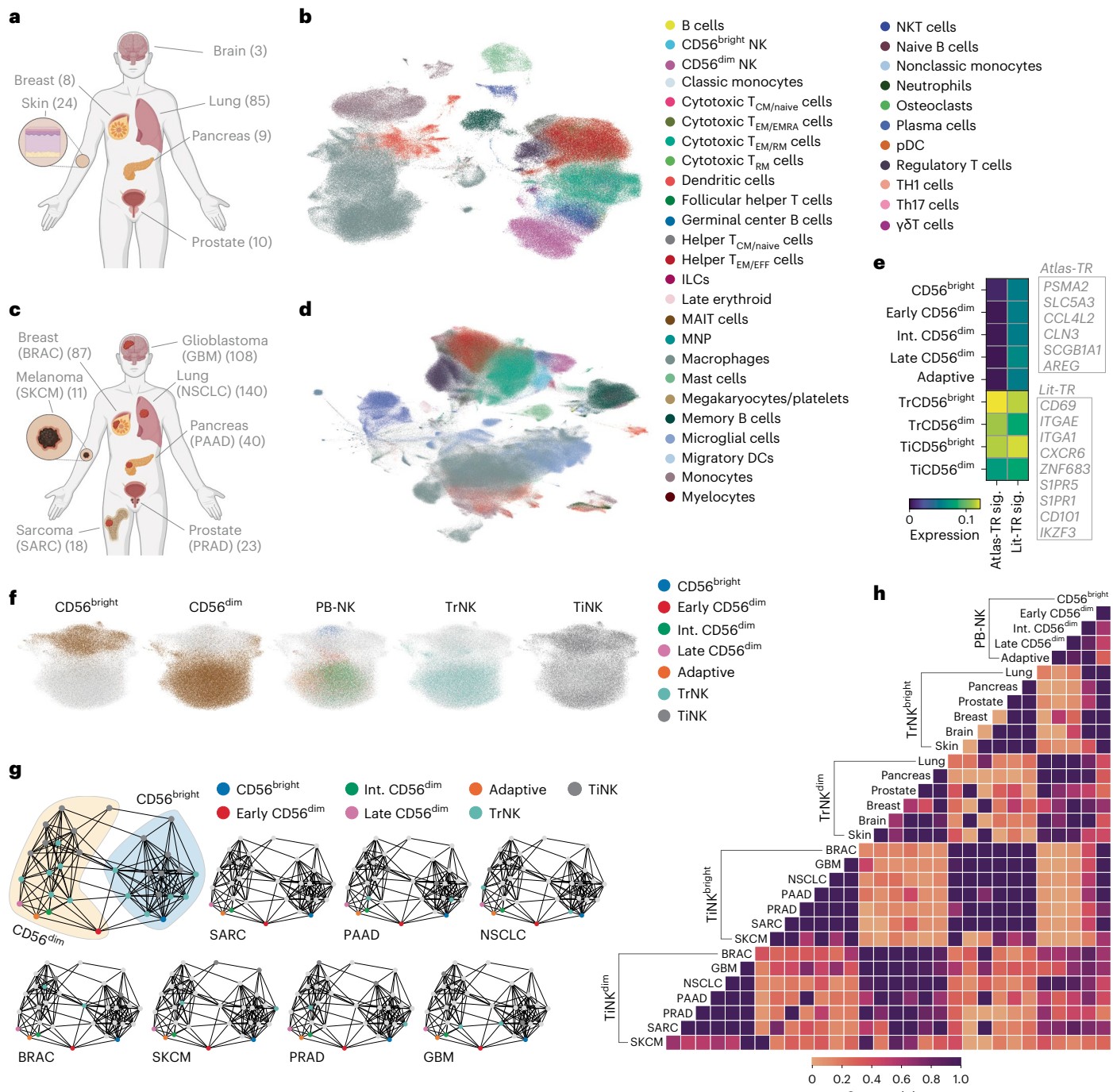

**Fig. 3 | Pan-cancer atlas of healthy tissue-resident and solid TiNK cells.**
**a**, Graphic overview of healthy tissue datasets included in the analysis, with the number of donors denoted in brackets. **b**, UMAP representation showing integration of all healthy tissue datasets. **c**, Graphic overview of solid tumor datasets included in the analysis, with the number of donors denoted in brackets. **d**, UMAP representation showing integration of all solid tumor datasets. **e**, Scoring of tissue-residency signatures (sig.) in PB-NK cell subsets,

as well as CD56^bright- and CD56^dim-annotated TrNK and TiNK subsets. **f**, UMAP representation showing integration of subset-annotated PB-NK, TrNK and TiNK cells. **g,h**, PAGA graphs (**g**) and connectivity heatmap (**h**) showing connectivity of PB-NK, TrNK and TiNK subsets across all tissues/tumor types, with individual tissues/tumor types highlighted (**g**). The scale represents gene set activity computed by AUCell (**e**). Panels **a** and **c** created with BioRender.com.

modulate NK cells at the transcriptional level can provide important insights into understanding the tumor-mediated immunosuppressive mechanisms and how to overcome them.

We implemented an unbiased approach (Milo[47]) to ascertain cellular states in our pan-cancer NK cell atlas by identifying 6,932 individual neighborhoods without pre-clustering based on cellular origin. Annotating individual neighborhoods as subset specific (>70% of cells

in the neighborhood) identified TiCD56^bright NK cells as having the most frequent, but also the most unique (differentially abundant), specific neighborhoods (Extended Data Fig. 7a). Notably, most neighborhoods were annotated as 'mixed', highlighting transcriptional similarities among NK cells found in PB, tissues and tumors (Extended Data Fig. 7a). The 6,932 neighborhoods were grouped into 6 distinctive neighborhood groups and tested for differential abundance of neighborhoods

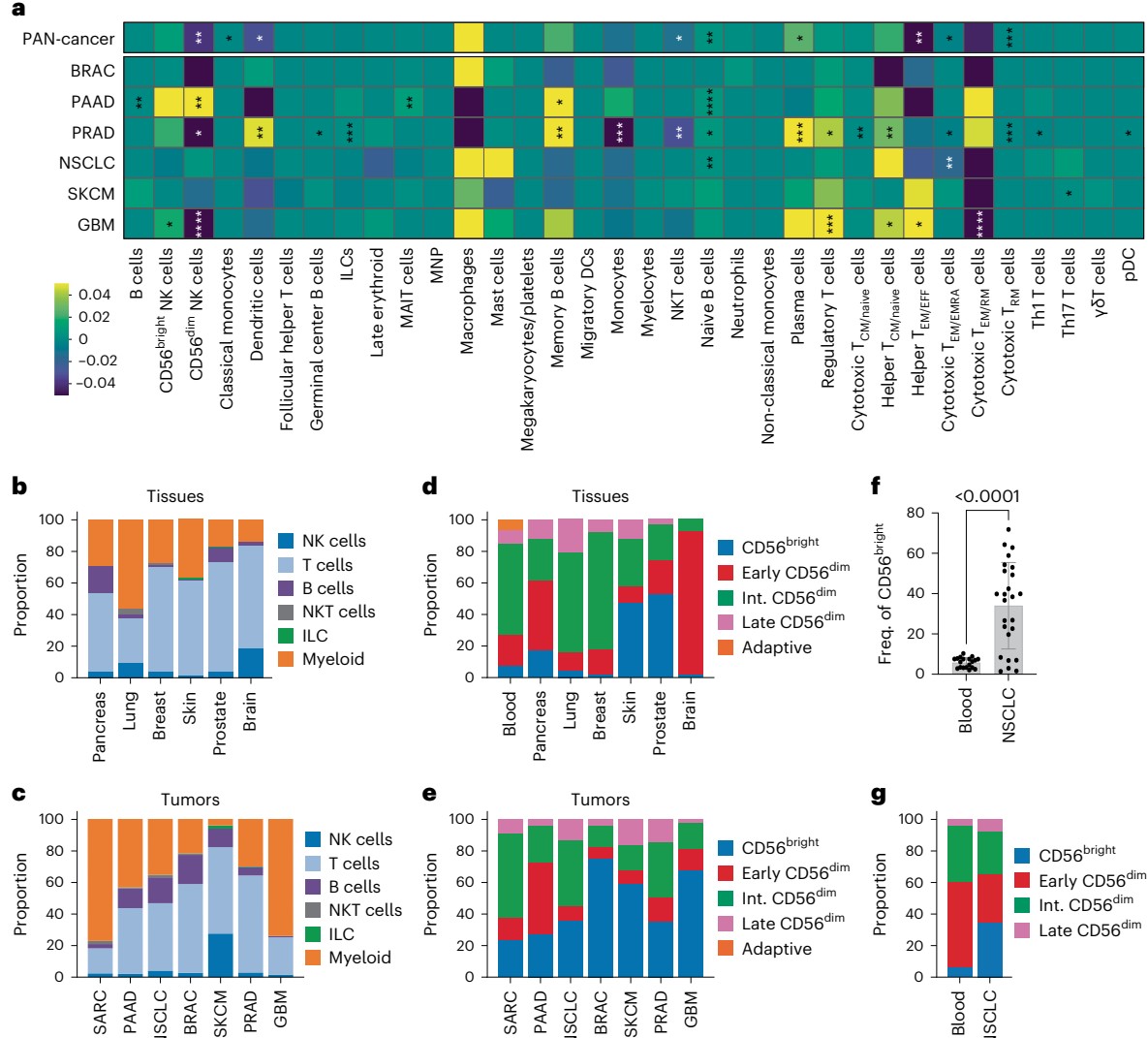

**Fig. 4 | Cellular composition of pan-cancer cell atlas and subset distribution of TiNK cells. a**, Heatmap depicting changes in immune subset proportion in tumor samples compared with healthy tissue samples at the pan-cancer level and within individual tumor types. **b,c**, Proportions of major immune subsets within healthy tissue (**b**) and tumor samples (**c**). **d,e**, Predicted subset annotations of CellTypist-identified NK cells in healthy tissue (**d**) and tumor samples (**e**) compared with annotated PB-NK cells. **f,g**, Frequency of CD56^bright NK cells (**f**) and relative frequency of subsets (**g**) identified by flow cytometry in a cohort of healthy blood donors (*n* = 19) and central tumor samples from patients with NSCLC (*n* = 25), from 23 independent experiments. Data were analyzed using two-sample Student's *t*-test with Bonferroni's correction (**a**) and a two-tailed Mann–Whitney *U*-test (**f**): **P* < 0.05, ***P* < 0.01, ****P* < 0.001, *****P* < 0.0001. The bar graph in **f** represents the mean ± s.d, with the actual *P* value indicated (*P* < 0.0001).

between TiNK cells and Ref-NK cells (Fig. 5a and Extended Data Fig. 7b). Neighborhood groups 1 and 2 consisted of neighborhoods significantly enriched for TiNK cells and group 6 included neighborhoods enriched for Ref-NK cells (Fig. 5b and Extended Data Fig. 7b).

Next, we visualized the distribution of NK cell subsets within each group using our annotation model (M1). Groups 1 and 2 were enriched for, but not exclusive to, CD56^bright cells, whereas groups 3–6 were dominated by CD56^dim NK cell subsets (Fig. 5c). The dominant TF regulons of PB-NK cell differentiation previously identified (Fig. 2f) confirmed groups 1 and 2 as two CD56^bright states and groups 3–6 as four CD56^dim NK cell states (Fig. 5d).

Cell-state-specific GRNs, DEGs, gene set enrichment analysis (GSEA) and signature scoring informed our annotation of the states as stressed CD56^bright (group 1), typical CD56^bright (group 2), effector CD56^dim (group 3), adaptive CD56^dim (group 4), activated CD56^dim (group 5) and typical CD56^dim (group 6) (Fig. 5e–n and Extended Data Fig. 7c–i). Comparing the stressed with the typical CD56^bright state

(group 1 versus group 2) identified increased expression of the cellular stress response *ATF3* regulon, the hypoxia-induced *MAFF* regulon and numerous heat shock proteins (Fig. 5e,g and Extended Data Fig. 7f). The stressed CD56^bright cell state scored highly for immunosuppressive pathways (transforming growth factor (TGF)-β signaling, hypoxia, reactive oxygen species (ROS)) and exhibited increased metabolic activation (glycolysis, cholesterol homeostasis, fatty acid metabolism and mTORC1 (mammalian target of rapamycin complex 1)) (Fig. 5g,j–l). Furthermore, a low NK cell cytotoxicity score, exemplified by reduced effector and activating signaling molecules, was suggestive of reduced functionality in this stressed CD56^bright cellular state, which was uniquely enriched across all seven tumor types (Fig. 5i,m,o). In line with increased infiltration of CD56^bright cells in the TME, the typical CD56^bright cellular state was also enriched in five of seven tumor types compared with healthy tissue, with both CD56^bright groups exhibiting higher expression of immunomodulatory molecules, including *XCL1*, *XCL2* and *IFNG* (Fig. 5n–o).

Of the CD56$^{dim}$ states, the effector state was most frequently enriched across tumor types (SARC, PAAD), characterized by an enrichment for apical junction, actin and cytoskeleton-related genes as well as effector molecules (Fig. 5h and Extended Data Fig. 7g). This state, phenotypically enriched for intermediate and late CD56$^{dim}$ NK cell subsets, scored highly for NK cytotoxicity and oxidative phosphorylation and, importantly, low for immune suppression (Fig. 5i,k–m). The adaptive CD56$^{dim}$ state was uniquely enriched for adaptive NK cells, in line with adaptive-associated genes (*CD52, IL32, GZMH, CD3E*) being upregulated in this state (Fig. 5c and Extended Data Fig. 7c). The activated CD56$^{dim}$ state was distinguished by increased hypoxia, upregulated nutrient transporters and the mTORC1–Myc axis (Fig. 5i,k and Extended Data Fig. 7d,h). Last, the PB-enriched typical CD56$^{dim}$ state exhibited a low stress score and a high cytotoxicity score and was associated with IFN, tumor necrosis factor (TNF) and JAK/STAT signaling (Fig. 5i–j,m and Extended Data Fig. 7e,i). Notably, although we observed enrichment of individual cellular states in the TME, including the two CD56$^{bright}$ and the effector CD56$^{dim}$ states, all states were represented in healthy blood and tissue samples, albeit at different frequencies.

## State-specific signaling in the TME links to functionality
To elucidate any TME-based influence on the six functional states identified, we employed CellChat[48] to infer intercellular communication, focusing on commonly enriched signaling pathways across all seven tumor types. Group 1 and 2 NK cell states were enriched for incoming signaling across tumor type from four dominant communication pathways (Fig. 6a). Increased expression of *CD44*, *CXCR4* and *CD74* on group 1 and 2 NK cells, on which numerous signals from fibroblasts, endothelial cells, tumor cells and macrophages converged (COLLAGEN, MIF, LAMININ), facilitated the augmented incoming signaling in NSCLC (Fig. 6b,c). Notably, the fibroblasts, endothelial cells, tumor cells and cancer-associated fibroblasts (CAFs) also exhibited the strongest outgoing interaction strength across tumor types (Extended Data Fig. 8a–g). Furthermore, group 1 and 2 NK cells preferentially received inhibitory input via the major histocompatibility complex I (MHC-I) (*HLA-E/KLRC1*) pathway owing to high *KLRC1* expression in these cellular states (Fig. 6a,d). Hence, group 1 and 2 cellular states were more receptive to TME-induced immunosuppressive signals via upregulated expression of *CD44*, *CXCR4*, *CD74* and *KLRC1*.

To understand how NK cells contribute to shaping the TME via an immunomodulatory role, we focused our analysis on outgoing signaling largely restricted to NK cells. We identified three signaling pathways (CC chemokine ligand (CCL), protease-activated inhibitors (PARs), IFN-II) through which NK cells predominantly communicated with dendritic cells, macrophages, fibroblasts and endothelial cells (Fig. 6e,f). CCL3 and CCL5, expressed across all states, can lead to the recruitment of cells expressing ACKR1, CCR1 and CCR4 (Extended Data Fig. 6h). Release of granzyme A, highly expressed at the transcriptional level by the effector NK cell state (group 3), can induce apoptosis of F2R-expressing cells in the TME, such as fibroblasts (Fig. 6g). Granzyme A expression was reduced in both frequency and intensity in CD56$^{dim}$ NK cells from central tumor samples from patients with NSCLC compared with healthy blood controls, hinting at a release of granzyme A by NK

cells within the tumor (Fig. 6h,i). Release of IFN-γ, predominantly by the stressed CD56$^{bright}$ (group 1) state, can induce surrounding cells to upregulate MHC-I expression, including HLA-E (Fig. 6g and Extended Data Fig. 9a–d). Inhibitory signaling via the HLA-E axis significantly inhibits degranulation and granzyme B release of both CD56$^{bright}$ and CD56$^{dim}$ NK cells, as demonstrated by co-culturing NK cells with A549 (NSCLC) targets cells pre-stimulated with IFN-γ to upregulate HLA-E expression (Fig. 6j,k and Extended Data Fig. 9a–e). Blockade of the NKG2A–HLA-E axis, using an anti-NKG2A antibody, resulted in significant recovery of function, both degranulation and granzyme B release (Fig. 6j,k and Extended Data Fig. 9e). CD56$^{bright}$ cellular states exhibited increased inhibitory signaling (MHC-I) and augmented susceptibility to TME-induced suppression (MIF, COLLAGEN, LAMININ) whereas CD56$^{dim}$ states, particularly the effector state, exhibited high *GZMA* signaling, which was confirmed in samples of CD56$^{dim}$ from patients with NSCLC.

## Ratio of cellular states is predictive of patient outcome
Having identified 6 functionally distinct cellular states of NK cells within our pan-cancer NK cell atlas comprising 89,850 scRNA-seq transcriptomes, we validated our findings in spatial RNA-seq datasets (Supplementary Table 5). Spatial RNA-seq tissue sections from SKCM, NSCLC and GBM were deconvoluted using Tangram[49] combined with our established scRNA-seq references for the tumor types being analyzed to identify the cell types in these datasets (Fig. 7a). Compositional analysis of the main immune subtypes in SKCM, NSCLC and GBM varied greatly across tumor type, but was highly consistent across sequencing technique (scRNA-seq versus spatial-seq) (Fig. 7b). Focusing on SKCM, harboring the highest proportion of NK cells (Fig. 7b), we could further stratify the annotated NK cells into CD56$^{bright}$ and CD56$^{dim}$ subsets (Fig. 7c) and cellular states (Fig. 7d). Importantly, confirming previous results (Fig. 5i,m), the effector (group 3) and typical (group 6) CD56$^{dim}$ states scored highly for genes associated with NK cell cytotoxicity. Similarly, stress response-related genes, as well as immunosuppressive-related genes (ROS, hypoxia) scored highest in the stressed CD56$^{bright}$ (group 1) state (Fig. 7f,g), in line with results in the scRNA-seq data (Fig. 5i–k).

The clinical benefit of NK cell infiltration in solid tumors has previously been assessed through a general NK cell signature score[50,51]. Having identified six functional states of NK cells in blood, tissue and solid tumors, in both scRNA-seq and spatial-seq datasets, we proceeded to test clinical relevance of these cellular states by using BayesPrism[52] to deconvoluted TCGA (The Cancer Genome Atlas) RNA-seq data where we also had survival data[53,54] (Extended Data Fig. 10). A higher ratio of effector CD56$^{dim}$:stressed CD56$^{bright}$ NK state signatures was predictive of improved survival in SARC and SKCM (Fig. 7h). We hereby confirm that the six functional states identified in our pan-cancer NK cell atlas, and confirmed in spatial RNA-seq datasets, are also predictive of outcome in patients with osteosarcoma and melanoma.

## Discussion
In the present study, we report a compact description of the transcriptional diversification encompassing human NK cell differentiation at

---

**Fig. 5 | Distinct cellular states of NK cells identified in pan-cancer atlas.** **a**, UMAP depicting neighborhood (Nhood) groups identified by Milo and computed using the scVI representation. **b**, Beeswarm plot depicting differential abundance of neighborhoods (TiNK versus Ref-NK enriched). Colored neighborhoods are differentially abundant at a false recovery rate (FDR) of 0.1. **c**, Pie charts showing distribution of NK subsets across neighborhood groups annotated using our annotation model (Fig. 1). **d**, Expression of dominant TF regulons of NK cell differentiation across NK cell states (neighborhood groups). **e**, Expression of TF regulons uniquely expressed across cellular states. **f**, Graphic representation of cellular states. **g,h**, Volcano plots depicting DEGs between group 1 versus group 2 (**g**) and group 3 versus group 4/5/6

(**h**) cellular states. Differential expression analysis was performed using the findNhoodGroupMarkers method within the miloR package. Counts were aggregated per sample; groups were compared using edgeR and the adjusted *P* values were used for the plots. **i**, Scoring of pathway gene signatures in NK cell states. Func., function; homeo., homeostasis. **j–n**, Dot plots depicting selected genes belonging to stress response (**j**), immune suppression (**k**), metabolism (**l**), cytotoxicity (**m**) and chemokine/cytokine secretion (**n**). **o**, Pie charts depicting distribution of NK cell states in blood, tissues and tumors. Volcano plots: log(fold-change) cutoff at 0.5, *P* < 0.05. The scale represents regulon activity (**d** and **e**) or gene set activity (**i**) computed by AUCell.

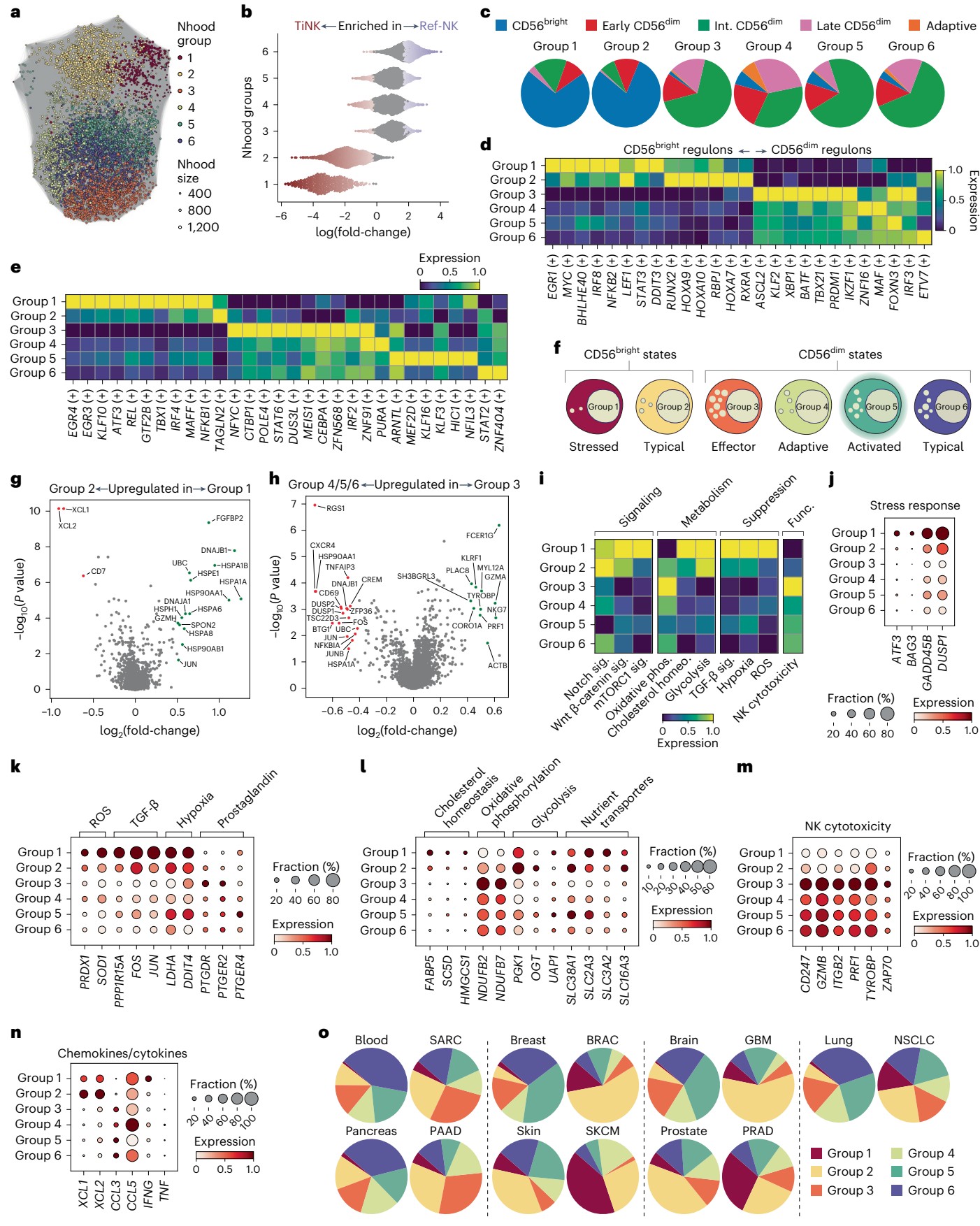

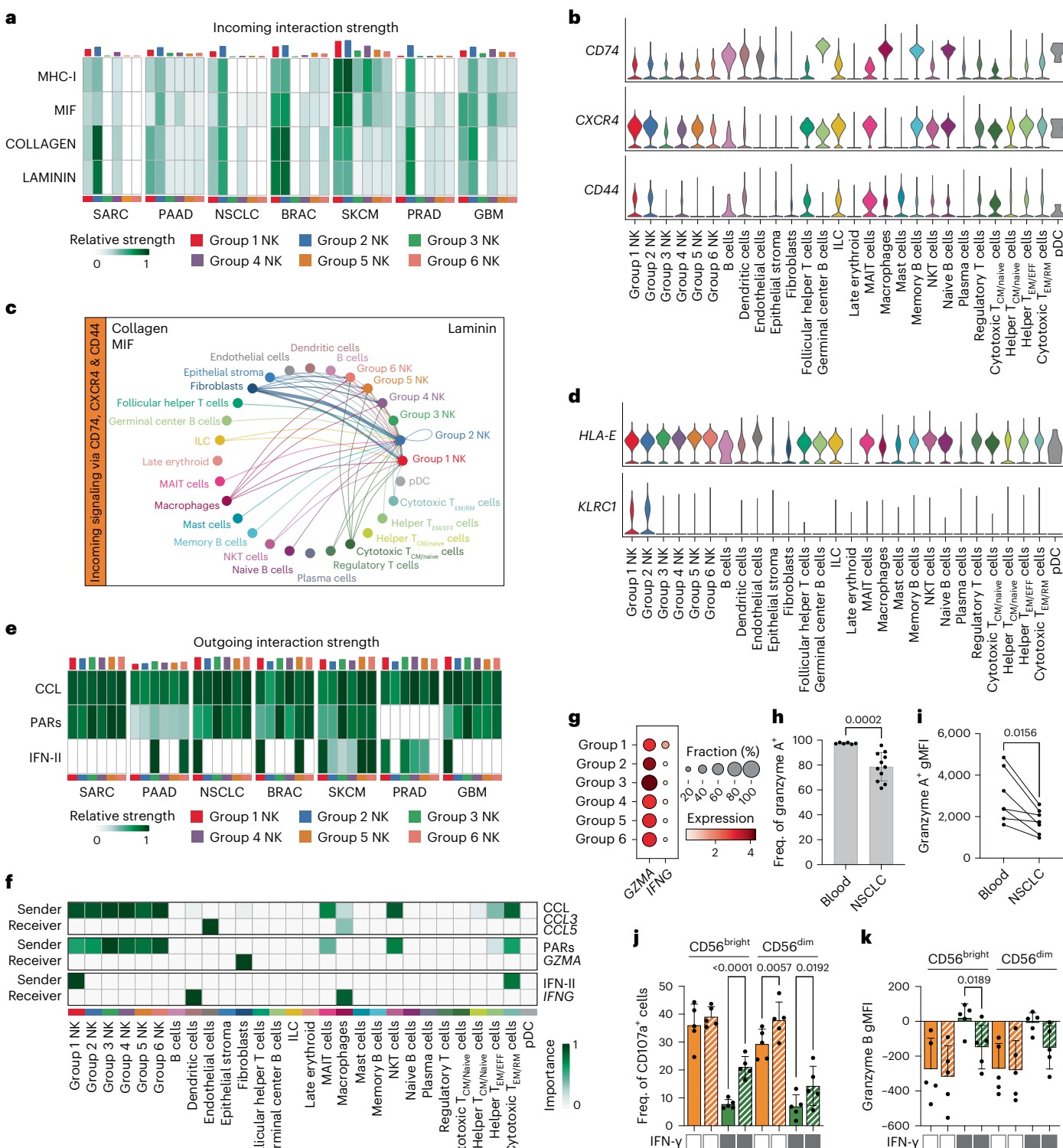

**Fig. 6 | Intercellular communication of distinct cellular states in the TME.**
**a**, Selected predicted incoming signaling pathways involving TiNK cells common across tumor type, identified by CellChat. **b**, Violin plots showing expression of receptors for the MIF, COLLAGEN and LAMININ communication pathway in NSCLC. **c**, Circle plot depicting predicted incoming signaling via *CD74*, *CXCR4* and *CD44* expression (NSCLC). **d**, Violin plots showing expression of receptor and ligand for the MHC-I communication pathway in NSCLC. **e**, Selected predicted outgoing signaling pathways involving TiNK cells across tumor type. **f**, Heatmap depicting interaction role of individual cell populations in CCL, PARs and IFN-II signaling pathways in NSCLC based on network centrality analysis. **g**, Dot plot depicting *GZMA* expression (ligand for PARs) and *IFNG* expression (ligand for IFN-II) in NK cell states across tumor type. **h**, Frequency of granzyme A⁺CD56^dim

NK cells in healthy blood donors (*n* = 6) and patients with NSCLC (*n* = 11) from 9 independent experiments. **i**, Geometric mean fluorescence intensity (gMFI) of granzyme A⁺CD56^dim NK cells in healthy blood donors and patients with NSCLC (*n* = 7) from six independent experiments. **j,k**, Degranulation (CD107a) (**j**) and granzyme B release (**k**) of CD56^bright and CD56^dim NK cells against A549 target cells pre-treated with (black box) and without (white box) IFN-γ (24 h) in the presence (black box) or absence (white box) of α-NKG2A antibody (E:T 1:1, 4 h, *n* = 5 biologically independent replicates from one experiment). Data were analyzed using two-way analysis of variance followed by Šidák's multiple-comparison test (**j** and **k**), two-tailed Mann–Whitney *U*-test (**h**) or two-tailed Wilcoxon's test (**i**). All bar graphs represent the mean ± s.d, with the actual *P* values indicated.

the single-cell level. By enriching for less frequent, but phenotypically well-defined, functionally distinct NK cell subsets, we could first train a model to correctly annotate five transcriptional subsets from bulk NK cell populations. By applying probabilistic models implemented in scvi-tools, we created a transcriptional reference map of human blood and TrNK cells from normal tissues, including blood, pancreas, lung, breast, skin, prostate and brain. Transfer learning using scArches facilitated integration of query datasets comprising a total of 2,176,214 transcriptomes from 427 patients spanning 7 solid tumor types. By extracting, annotating and mapping the TiNK cells on to our reference map of healthy donors, we could systematically interrogate TME-induced perturbations of GRNs and functional states of TiNK cells (Supplementary Fig. 1). Our pan-cancer atlas revealed six functionally distinct NK cell states with varying abundance across blood, tissues and tumor types, which we could confirm in spatial RNA-seq datasets (SKCM, NSCLC, GBM). Two states commonly enriched for across tumor types included a dysfunctional CD56[bright] cellular state susceptible to TME-induced immunosuppression and a cytotoxic TME-resistant CD56[dim] state, the ratio of which was predictive of patient outcome.

The view that NK cells, like T cells and other immune cells, undergo a continuous process of NK cell differentiation is relatively recent and was originally based on phenotypic and functional classification of discrete subsets[7,55]. There is abundant evidence to suggest that the CD56[bright] NK cell subset is the most naive, giving rise to the more differentiated CD56[dim] NK cells which can further differentiate toward terminal stages, a process accelerated by CMV infection[8,56,57]. Instead of forcing individual NK cells into arbitrary clusters representing a snapshot of a given time point of differentiation, we clustered TFs and their target genes into five distinct gene expression trends as a function of pseudotime, reflecting continuous differentiation. The dominant TF regulons within these five gene trends correlated with functional traits of NK cells along the differentiation axis, such as cytokine responsiveness, as well as proliferative and cytotoxic capacity. By retaining fate-specific expression profiles, conventional versus adaptive fate in donors with CMV-induced clonal NK cell expansions, we could observe clear divergence of regulon expression (for example, BATF, MAF) during terminal differentiation. BATF belongs to the AP-1 TF family which have been identified as potential drivers in shaping adaptive NK cell chromatin accessibility and thus dictating the unique functional features of this subset, including enhanced IFN-γ response to receptor stimulation[15]. Establishing dominant regulons defining NK cell differentiation in PB provided a vital reference for downstream interrogation of both TrNK and solid TiNK cells.

Utilizing CellTypist, we harmonized annotations of individual cell subtypes across multiple datasets from six different healthy tissues, extracting and integrating CD56[bright] and CD56[dim] NK cells using scVI[19] to expand our transcriptional reference map. Importantly, tissue-, as well as tumor-annotated, NK cells, did not express human ILC signature genes (IL7R), instead expressing both EOMES and TBX21. Literature-derived tissue-residency genes (for example, CD69, ITGAE, ITGA1, CXCR6, ZNF683 and IKZF3), originally extrapolated from tissue-resident T cell signatures[58–61], were more highly expressed in tissue-derived NK cells, particularly in CD56[bright] NK cells[62]. Using our extensive pan-cancer NK cell atlas, we were able to generate a solely NK cell-derived, tissue-residency signature (atlas-TR: PSMA2, SLC5A3, CCL4L2, CLN3, SCGB1A1, AREG), which outperformed the conventional literature-derived TR signature across tissue and tumor type. CD56[bright] and CD56[dim] NK cells from healthy brain tissue exhibited a low TR-score, indicative of potential blood contamination in this specific dataset. Importantly, GBM-derived CD56[bright] and CD56[dim] NK cells scored highly for tissue residency, supporting their infiltration into the tumor. Expression of CCL4L2, encoding a chemokine that induces chemotaxis of CCR5- and CCR1-expressing cells, such as T cells, dendritic cells and macrophages, has previously been described in NK cells isolated from melanoma samples[63]. This represents an

independent verification, because this dataset was not included in our study. These melanoma-infiltrating NK cells also exhibited high AREG expression, an epidermal growth factor (EGF) receptor ligand. Notably, upregulation of AREG has also been described in the setting of healthy and cirrhotic liver-resident NK cells[64], a tissue type not included in our pan-cancer atlas. Intriguingly, SCGB1A1, a member of the secretoglobin family, functions as a potent inhibitor of phospholipase $A_2$ (ref. 65), a well-described immunosuppressive molecule contributing to the development of the TME. Hence, it is tempting to speculate that secretion of the SCGB1A1-encoded protein could be another effector mechanism through which TiNK cells can positively contribute to remodeling of the TME.

The presence and abundance of NK cells that reside in the tumor bed vary across tumor types and treatments and between patients, and appears to be associated with the chemokine profiles in the different tissues/TMEs[66–69]. In agreement with previous studies[45,67,70], we observed a predominance of CD56[bright] NK cells in tumors compared with the corresponding normal tissue. TrNK cells are probably a mixed population including naturally residing TrNK cells and TiNK cells. Compositional differences between normal and tumor tissues suggests some degree of active recruitment, particularly in SKCM where NK cell frequencies starkly increased, albeit expansion from tissue-resident pools cannot be excluded. Migration into the TME is regulated by a broad family of integrins, selectins and chemokine receptors that are differentially expressed during NK cell differentiation. CXCR3, primarily expressed on CD56[bright] NK cells, has been implicated in homing to several solid tumors based on CXCL10 gradients[71,72], and thus may contribute to the predominance of this subset in tumors. CCL2, CCL3, CCL5, CXCL8, CXCL9, CXCL10 and CXCL12 have similarly been implicated in mediating predominantly CD56[bright] NK cell trafficking into solid tumors based on chemokine receptor expression[69]. Release of CCL3 and CCL5 by NK cells can also recruit CCR1-expressing immune cells, such as macrophages. We observed increased CXCR4 expression in group 1 and 2 cellular states, corresponding to CD56[bright] TrNK and TiNK cells. Previous reports[73,74] have demonstrated CD44-induced CXCR4 upregulation resulting in increased migration and invasiveness of malignant cells. Notably, CD44 was highly expressed on the tumor-enriched stressed CD56[bright] state, alongside CXCR4 and CD74, possibly sensitizing this population to TME-mediated immunosuppression from CAFs, fibroblasts, endothelial and tumor cells, as noted by high scores for TGF-β signaling, hypoxia and ROS. High immunosuppression of this state is in line with the increased stressed response noted, as exemplified by high expression of the cellular stress response-associated TF ATF3, the HSP70 co-chaperone BAG3, the stressful growth arrest gene GADD45B and DUSP1, which is associated with cellular response to environmental stress.

Transcriptional stress response programs, including heat shock proteins, have previously been reported as a potential artefact downstream of digestion of tissues[75]. We therefore took several measures to rule out digestion artefacts when compiling the present resource. In addition to implementing upstream data-processing steps, including removal of ambient RNA using decontX[76], we found no evidence for systematic artefactual stress signal coming from a particular study or tumor type. Perhaps most importantly, the stress signature defining the group 1 NK cell state was also found in spatial transcriptomics data directly on tissue sample sections that have not undergone any upstream tissue dissociation/digestion.

We also found high KLRC1 expression on the group 1 and 2 states, which, alongside high IFNG expression, can induce an inhibitory feedback loop, whereby local IFN-γ secretion leads to HLA-E upregulation resulting in inhibitory input through CD94/NKG2A. Conversely, the effector CD56[dim] state, associated with improved patient outcome, lacked CD44 expression and highly expressed GZMA. Notably, this state exhibited high expression of the KLF2, PRDM1, BATF, TBX21 and IKZF1 regulons, indicative of high effector function, regulation of

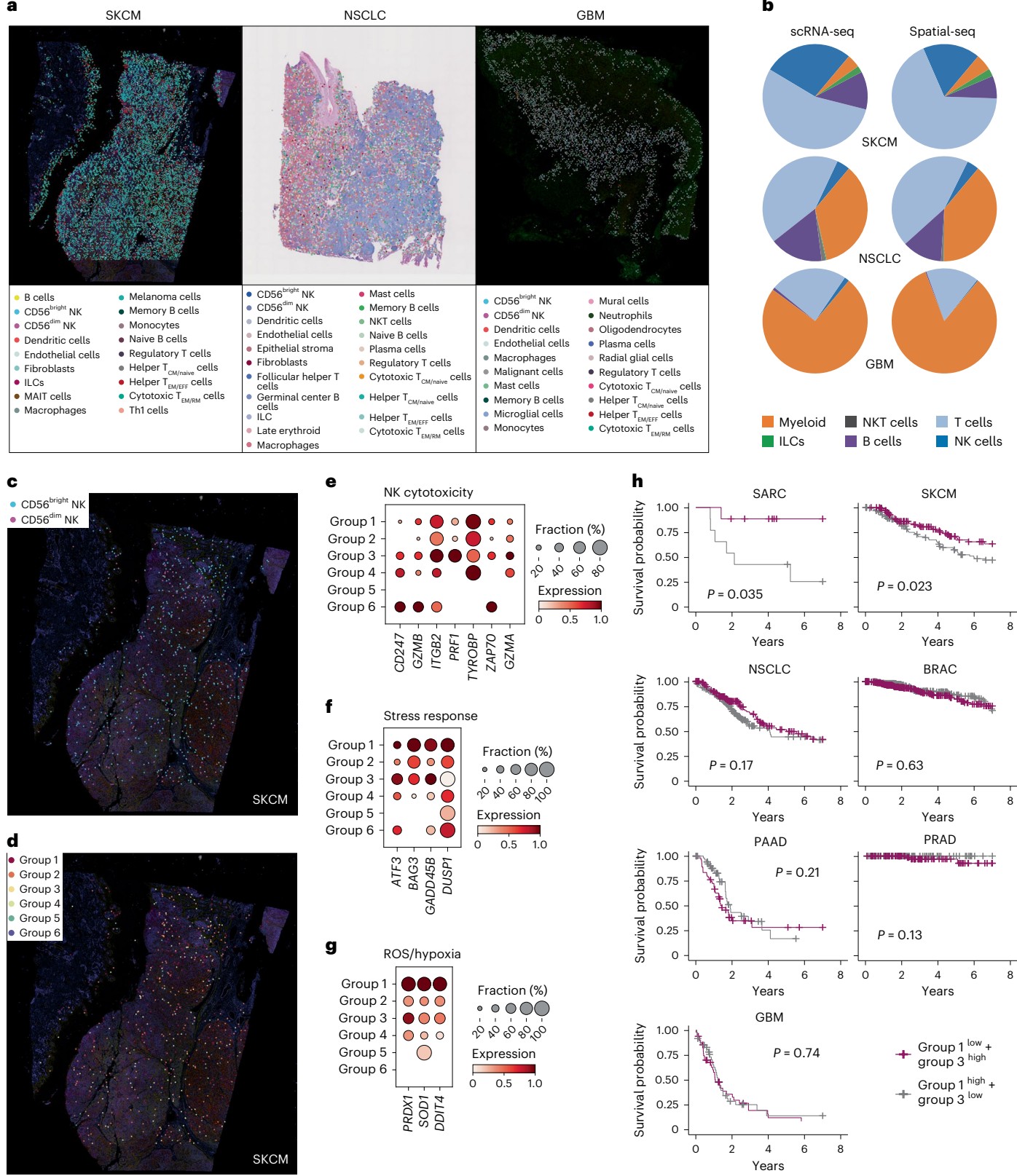

**Fig. 7 | Distinct cellular states in spatial RNA-seq and association with patient outcome. a**, Deconvoluted spatial RNA-seq images from SKCM, NSCLC and GBM at the level of immune populations. **b**, Pie charts depicting compositional analysis of major immune populations from scRNA-seq datasets and spatial-seq datasets for SKCM, NSCLC and GBM samples. **c,d**, Annotation of CD56^bright and CD56^dim NK cell subsets (**c**) and the six cellular states of NK cells (**d**) in SKCM.

**e–g**, Dot plots depicting selected genes belonging to NK cytotoxicity (**e**), stress response (**f**) and immunosuppression (**g**) scored across NK cell states in spatial-seq data from SKCM. **h**, Kaplan–Meier survival curves showing association of high/low group 1/3 gene signatures with patient outcome across tumor types. Survival analysis was performed using Cox's proportional hazards model; P values were computed using the log(rank) test.

homeostatic proliferation and survival, but also cell migration and tissue residency. Unique TiNK cell-specific regulons in this state consisted of *NFYC*, *CTBP1*, *POLE4* and *CEBPA*, which are involved in DNA repair, monitoring of proliferation, regulating MHC expression and maintaining structural homeostasis in the Golgi complex[77–80]. Conversely, TiNK cell-specific regulons in the stressed CD56^bright state included hypoxia-induced *MAFF*, cellular stress response regulon *ATF3* and *EGR3* (ref. 81) which induce negative regulators in response to activation. Metabolically, the effector CD56^dim state scored highly for oxidative phosphorylation, compared with the stressed CD56^bright state which favored glycolysis, mTORC1 activation and exhibited upregulated nutrient transporters and genes associated with cholesterol homeostasis.

Contrary to Tang et al.[16], increased gene signature scoring of the tumor-enriched states stressed that the CD56^bright state did not consistently associate with reduced survival across tumor types. Instead, we observed increased survival in patients exhibiting a high effector CD56^dim state, which was further augmented with a low signature for the stressed CD56^bright state. Of the four CD56^dim states, the effector CD56^dim state was enriched across two tumor types, painting a promising picture for the role of solid TiNK cells.

This resource provides a transcriptional reference map of human NK cells across healthy blood and tissues with harmonized annotations of transcriptional NK cell subsets. Uncovering the dominant gene-regulatory circuits during NK cell differentiation enabled identification of TME-induced perturbations in solid TiNK cells across tumor type. We identified functionally distinct NK cell states across healthy and malignant tissues, including tumor-enriched states predictive of patient outcome. Modeling of the intercellular communication pathways of outcome predicting NK cell states with the surrounding TME identified potential pathways of TME-induced NK cell suppression. Thus, our analysis has the potential to design more potent NK cell therapy products able to resist suppressive factors operating within the TME of solid tumors. Ultimately, this resource can be extended endlessly through transfer learning to interrogate new datasets from experimental perturbations or different tumor types.

## Online content

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

[1]Department of Cancer Immunology, Institute for Cancer Research, Oslo University Hospital, Oslo, Norway. [2]Precision Immunotherapy Alliance, University of Oslo, Oslo, Norway. [3]Center for Infectious Medicine, Department of Medicine Huddinge, Karolinska Institutet, Stockholm, Sweden. [4]Fate Therapeutics, San Diego, CA, USA. [5]Wellcome Sanger Institute, Wellcome Genome Campus, Hinxton, Cambridge, UK. [6]Wellcome-MRC Cambridge Stem Cell Institute, Jeffrey Cheah Biomedical Centre, Cambridge Biomedical Campus, University of Cambridge, Cambridge, UK. [7]Department of Medicine, University of Cambridge, Cambridge, UK. [8]Center for Hematology and Regenerative Medicine, Department of Medicine Huddinge, Karolinska Institutet, Huddinge, Sweden. [9]Oslo Cancer Cluster, NEC OncoImmunity AS, Oslo, Norway. [10]Department of Vaccine Informatics, Institute for Tropical Medicine, Nagasaki University, Nagasaki, Japan. [11]Department of Immunology & Immunotherapy, Lipschultz Precision Immunology Institute, Icahn School of Medicine at Mount Sinai, New York, NY, USA. [12]Department of Oncological Sciences, Tisch Cancer Institute, Icahn School of Medicine at Mount Sinai, New York, NY, USA. [13]These authors contributed equally: Herman Netskar, Aline Pfefferle. [14]These authors jointly supervised this work: Amir Horowitz, Karl-Johan Malmberg. ✉e-mail: aline.pfefferle@ki.se; k.j.malmberg@medisin.uio.no

## Methods

### Cell processing
Peripheral mononuclear cells (PBMCs) were isolated using density gradient centrifugation from anonymized healthy blood donors (Oslo University Hospital; Karolinska University Hospital) with informed consent. The study was approved by the regional ethics committee in Norway (Regional etisk komité (REK): protocol no. 2018/2482) and Sweden (Regionala etikprövningsnämnden i Stockholm: protocol no. 2016/1415-32; Etikprövningsmyndigheten: protocol no. 2020-05289). Donor-derived PBMCs were screened for KIR education and adaptive status using flow cytometry. NK cells were purified using an AutoMACS (DepleteS program, Miltenyi Biotec) and before overnight resting in complete Roswell Park Memorial Institute (RPMI) 1640 (Cytiva) (10% fetal bovine serum (FBS; GE Healthcare), 2 mM L-glutamine (GE Healthcare)) at 37 °C and 5% $CO_2$.

### Flow cytometry screening
PBMCs were stained for surface antigens and viability in a 96 V-bottomed plate, followed by fixation/permeabilization and intracellular staining at 4 °C. The following antibodies were used in the screening panel: CD3-V500 (clone UCHT1), CD14-V500 (clone MφP9), CD19-V500 (clone HIB19) and Granzyme B-AF700 (clone GB11) from Beckton Dickinson; CD57-FITC (clone HNK-1), CD38-BV650 (clone HB-7) and CD158e1-BV421 (clone DX9) from BioLegend; CD158a-APC-Vio770 (clone REA284) and CD158a/h-PE-Vio770 (clone 11PB6) from Miltenyi Biotec; and CD158b1/b2,j-PE-Cy5.5 (clone GL183), CD159a-APC (clone Z199) and CD56-ECD (clone N901) from Beckman Coulter. LIVE/DEAD Fixable Aqua Dead Stain kit for 405-nM excitation (Life Technologies) was used to determine viability. Samples were acquired on an LSR-Fortessa equipped with a blue, red and violet laser and analyzed in FlowJo v.9 (TreeStar, Inc.).

### FACS sorting
Cells were harvested and surface stained with the following antibodies: CD57-FITC (HNK-1) from BioLegend; CD158e1/e2-APC (clone Z27.3.7), CD56-ECD (clone N901) and CD158b1/b2,j-PE-Cy5.5 (clone GL183) from Beckman Coulter; and CD158a-APC-Vio770 (clone REA284), CD159c-PE (clone REA205) and CD159a-PE Vio770 (clone REA110) from Miltenyi Biotec. Cells, 12,000, were directly sorted into Eppendorf tubes at 4 °C for each sample using a FACSAriaII (Beckton Dickinson). Sorting strategies for scRNA-seq for the donor with and without an adaptive NK cell expansion are depicted in Extended Data Fig. 1c,d.

### ScRNA-seq
After sorting, cells were kept on ice during the washing (phosphate-buffered saline (PBS) + 0.05% bovine serum albumin (BSA)) and counting steps. Cells, 10,000, were resuspended in 35 µl of PBS + 0.05% BSA and immediately processed at the Genomics Core Facility (Oslo University Hospital) using the Chromium Single Cell 3′ Library & Gel Bead Kit v.2 (Chromium Controller System, 10x Genomics). The recommended 10x Genomics protocol was used to generate the sequencing libraries, which was performed on a NextSeq500 (Illumina) with ~5% PhiX as spike-in. Sequencing raw data were converted into fastq files by running Illumina's bcl2fastq v.2.

### ScRNA-seq data collection and processing
Previously published scRNA-seq data were collected mostly in the form of count matrices already aligned to GRCh38; the rest were collected as fastq files. For the datasets where we collected fastq files, the data were aligned to GRCh38 using Cell Ranger (10x Genomics Cell Ranger 7.0.0).

### Quality control and normalization of scRNA-seq data
Data-cleaning steps were first carried out whereby cells not expressing a minimum of 1,000 molecules and genes expressed by <10 cells were filtered out. Doublets were removed using the SOLO algorithm[82].

The count matrices for all the tumor and tissue types were corrected for ambient RNA using decontX[76]. The data were normalized using log(transformation) for some of the downstream analysis as well as for visualization of gene expression-like dot plots. Quality control, transformation and most of the visualization of the gene expression data were performed using Scanpy[83]. For analysis using scVI and scANVI, the raw count data were used.

### Integration of scRNA-seq data
The probabilistic models scVI and scANVI, as implemented in scvi-tools[19], were used for integration of scRNA-seq data. These methods have been shown to perform well for integration of scRNA-seq data, especially when dealing with complex batch effects and integrating atlas-level data[84]. For cell-type and -subset annotations and prediction, scANVI was used to capture annotation of single-cell profiles. For the analysis of PB-NK subsets, the sorted subsets provided labels for training the scANVI model. The subset prediction provided by the model was tested on a held-out set of cells (15%) from the sorted subset data, giving us a confusion matrix summarizing the performance of the prediction.

### Dimensionality reduction, clustering and visualization of scRNA-seq data
We computed the Uniform Manifold Approximation and Projection (UMAP) embeddings for visualization using the embedding learned from scVI and scANVI. Unsupervised clustering was also carried out using this learned embedding with Phenograph and the Leiden algorithm as implemented in Scanpy. PAGA[29] was used to quantify the connectivity of different groups of cells, thereby providing a representation of the data as a simpler graph. The various plots were mostly generated using the plotting functions in Scanpy.

### Cell-type annotations and harmonization
For many of the publicly available datasets, cell-type annotations were readily available and used as seed labels when training the scANVI model for that particular tissue/tumor type to annotate the nonimmune cells. The scANVI model allowed us to harmonize annotations that were needed for analysis across datasets. All immune cells for all tissue types were integrated using scVI and annotated using CellTypist[42]. The same was done for all immune cells across all tumor types. The CD16$^-$ and CD16$^+$ NK cells identified by CellTypist were annotated as CD56$^{bright}$ and CD56$^{dim}$, respectively. Where CITE-seq data were available, the surface expression of key markers also helped validate the cell-type annotations. For the identified NK cells, the cells were also scored using NK1/NK2 (CD56$^{bright}$/CD56$^{dim}$) signatures to validate the annotation of CD56$^{bright}$ and CD56$^{dim}$ NK cells. We also performed our own unsupervised Leiden clustering, which identified two dominating clusters corresponding to CD56$^{bright}$ and CD56$^{dim}$ NK cells.

### Calculation of signature scores
Signature scores were computed using AUCell[21], allowing for exploration of the relative expression of the signatures of interest in the datasets. Various gene sets were taken from the MSigDB Hallmark gene set collection[85].

### Pseudotime and RNA velocity analysis
Pseudotime was computed using Palantir[26], which captures the continuous nature of differentiation, and cell fate, which allowed us to explore two terminal states and the gene expression changes seen along these trajectories. For this analysis, the starting cell was defined as the cell that was the least CD56$^{dim}$ (the lowest score for the NK1 signature). GAMs fitted on cells ordered by pseudotime were used to calculate gene trends, where the contribution of cells was weighted by their probability to end up in the given terminal state as calculated by Palantir. The gene trends indicate how gene expression levels develop

over the differentiation timeline. These trends were clustered using the Leiden clustering algorithm to give us five clusters of gene trends. RNA velocity[28] was also used to take advantage of splicing kinetics to identify directed dynamic information. We used velocyto[28] and scVelo[27] for this analysis, specifically the dynamic model implemented in the scVelo toolkit. The RNA velocity analysis was run on the 2 donors where sorted subsets were sequenced separately, as well as on the integrated data from 12 blood donors.

## GRN analysis

SCENIC[21] was used to infer TFs and GRNs from the scRNA-seq data. The SCENIC workflow[86] was followed and the pySCENIC implementation was used. TF–gene associations were inferred by GRNBoost[87] and motif–TF associations were downloaded from Aerts's lab website and used for pruning the inferred associations. The inferred regulatory networks were also further pruned by removing lowly expressed TFs based on the bulk RNA-seq data. AUCell was used to compute the activity of the final regulons. The regulon activity was visualized using matrix plots, as implemented in Scanpy, to look at the activity across different groups of cells.

## Bulk RNA-seq for TF and target validation

For validation of the TF and targets, we checked their expression in bulk RNA-seq data from four sorted NK cell populations (CD56$^{bright}$, NKG2A$^-$KIR$^-$CD56$^{dim}$, NKG2A$^-$KIR$^+$CD56$^{dim}$ and NKG2A$^-$KIR$^+$NKG2C$^+$CD56$^{dim}$). Sequencing was performed using single-cell tagged reverse transcription[88].

## Reference mapping

The TiNK cells were added after the model for a healthy NK cell reference was trained. Then, scArches[44] as implemented in scvi-tools[19] was used to map these new data on to the established reference.

## Cell–cell communication inference using CellChat

To infer the communication between the various cell types in the tumor datasets we used CellChat[48]. Based on gene expression of receptors and ligands in the data and a curated database of pathways, CellChat computes the communication probability between various receptor–ligand pairs. CellChat also provided ways to aggregate this information and for us to visualize the inferred cell–cell communication networks. CellChat was computed separately for each of the tumor types included in the analysis.

## Differential gene expression analysis

To perform differential gene expression analysis we used pseudobulk because this has shown good results when analyzing scRNA-seq data in various studies[89]. This allowed us to aggregate up counts for each sample and consider the samples instead of the cells as replicates. We then used edgeR[90] on the pseudobulk data. We could then identify DEGs between healthy reference NK cells and TiNK cells within and across subsets.

## Differential abundance analysis using Milo

We used Milo[47] to assign cells to neighborhoods on the $k$-nearest neighbors graph ($k$-NNG). The scVI representation of the cells was used for building the $k$-NNG. This allowed us to have a batch-corrected representation of the cells as input to this analysis. The differential abundance of the neighborhoods between the healthy reference and the TiNK cells was then computed. The neighborhoods were grouped into six groups using the groupNhoods function in Milo. These groups were considered as different NK cell states and further characterized using the functions in Milo for identification of DEGs. The differential expression analysis was done using pseudobulk by aggregating gene expression per sample. The single cells were then annotated using these groups for downstream analysis.

## GSEA

GSEA was performed using the GSEA software[91] and the MSigDB collection of gene sets. Genes were first ordered based on the differential expression analysis based on either the pseudobulk approach or the Milo analysis.

## Spatial transcriptomics

Spatial transcriptomics datasets from lung tumor, glioblastoma and melanoma were collected from the 10x Genomics website (https://www.10xgenomics.com/datasets). Squidpy[92] was used for preprocessing and segmentation and Tangram[49] was used for deconvolution using our annotated scRNA-seq data for each of the tumor types as reference. The deconvolution was performed with the NK cells annotated as CD56$^{bright}$ and CD56$^{dim}$, as well using the group annotations established in this paper.

## Clinical and bulk RNA-seq data from TCGA and TARGET

Bulk RNA-seq data and clinical data were downloaded from TCGA and TARGET using TCGAbiolinks[53] and curated survival data were downloaded from Xena[54].

## Deconvolution of bulk RNA-seq

Deconvolution of the bulk RNA-seq data was performed for each of the tumor types using BayesPrism[52]. BayesPrism has been shown to work well for deconvolution of data from tumors and especially well in dealing with high cell-type granularity[93]. The annotated reference datasets for each of the tumor types were used as prior information in the deconvolution. BayesPrism then computed both an expression matrix for each cell type and the cell-type fraction for each sample.

## Survival analysis

The NK expression matrix inferred by BayesPrism for the various tumor types was used to score the signature genes for each of the identified NK cell states. The patients were then assigned as high and low for a group/state based on belonging to the highest or lowest half in terms of expression of these signature genes within the group of patients with a specific tumor type. The high and low designations could then be combined in an approach where a patient could be assigned as high or low in multiple groups. Survival analysis was conducted using Cox's proportional hazards model from the R package survival[94], adjusting for confounding clinical factors such as tumor stage, gender and age. Subsequently, survival curves were derived using the Kaplan–Meier method within the same package. For visualization, the ggsurvplot function of the survminer package in R was utilized.

## Samples from patients with primary NSCLC

The patient cohort, processing of tissue specimens and flow cytometry staining were collected and performed as previously described[45].

## Functional assay using A549 cells

A549 cells were cultured in Dulbecco's modified Eagle's medium/high glucose with L-glutamine, sodium pyruvate (Cytiva) + 10% heat-inactivated FBS (Sigma-Aldrich) at 37 °C in 5% $CO_2$. A549 cells, 20,000, were seeded per well in a 96-well F-bottom plate and pre-treated with and without 50 ng ml$^{-1}$ of IFN-γ (PeptroTech) for 24 h before addition of NK cells. HLA-E expression after IFN-γ stimulation was evaluated using HLA-E–PE antibody (BioLegend, clone 3D12). NK cells were isolated using negative selection (NK cell isolation kit, Miltenyi Biotec) from previously cryopreserved PBMCs from healthy individuals. Cells were activated overnight with 5 ng ml$^{-1}$ of IL-15 (R&D) in RPMI 1640 (Cytiva) + 10% heat-inactivated FBS at 37 °C in 5% $CO_2$. NK cells were washed, resuspended in RPMI 1640 + 10% FBS and pre-incubated with and without α-NKG2A (a monalizumab biosimilar: immunoglobulin (Ig)G1 with PGLALA mutation, Merck) for 20 min prior. Target cells were washed in PBS before the addition of

NK cells at a 1:1 effector:target (E:T) ratio in the presence of brefeldin A (GolgiPlug, 1:1,000, BD Biosciences), monensin (GolgiStop, 1:1,500, BD Biosciences) and anti-CD107a-BUV394 (BD Horizon, clone H4A3). After a 4-h incubation, the cells were stained with anti-IgG Fc−PE (Invitrogen), followed by surface, fixation and permeabilization (Cytofix/Cytoperm, BD) and finally intracellular staining using the following antibodies: CD159a-VioBright FITC (Miltenyi Biotec, clone REA110), Granzyme B-AF700 (BD, clone GB11), CD16-Pacific Blue (BD, clone 3G8), CD3-V500 (BD, clone UCHT1), TNF-α-BV650 (BioLegend, clone Mab11), IFN-γ-BV785 (BioLegend, clone 4S.B3), CD56-ECD (Beckman Coulter, clone N901) and perforin−PE-Cy7 (eBioscience, clone dG9), LIVE/DEAD Fixable Aqua Dead Cell Stain kit (Thermo Fisher Scientific).

### Reagents and antibodies
A full list containing company information, catalog nos and antibody clones for all reagents can be found in Supplementary Data.

### Reporting summary
Further information on research design is available in the Nature Portfolio Reporting Summary linked to this article.

### Data availability
The gene expression data generated for this paper are available at the National Center for Biotechnology Information's Gene Expression Omnibus with accession no. GSE245690 and raw sequencing data are available at the European Genome−Phenome Archive with accession no. EGAS50000000014. The details about the publicly available data included in the analysis are available in Supplementary Tables 1, 2 and 3. For GSEA the Molecular Signature Database (v.2023.2.Hs), available at https://www.gsea-msigdb.org/gsea/msigdb, was used. Relevant gene sets for scoring were also retrieved from this database. Bulk RNA-seq data were downloaded from TCGA and TARGET. Curated survival data were downloaded from Xena. Processed data and models have also been made available via Zenodo at https://doi.org/10.5281/zenodo.8434223 (ref. 95) and as an online resource at http://nk-scrna.malmberglab.com. Source data are provided with this paper.

### Code availability
The code generated for our analysis is available on GitHub at https://github.com/hernet/transcriptional-map-nk.

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

### Acknowledgements
Large parts of the analyses were run using the Machine learning infrastructure (ML Nodes), University Centre for Information Technology, University of Oslo, Norway. This publication is part of the Human Cell Atlas (www.humancellatlas.org/publications), HCA-106. We thank the flow cytometry and genetics core facilities at Oslo University Hospital. We thank Merck KGaA for providing tool reagents. This work was supported by the Swedish Research Council (grant nos 223310 to K.-J.M. and 2021-03069 and 2021-01039 to N.M.), the Swedish Children's Cancer Society (grant no. PR2020-1059 to K.-J.M.), the Swedish Cancer Society (grant nos 21-1793Pj to K.-J.M., 22-2319Pj to N.M. and 23-2946Pj to J.M.), Sweden's Innovation Agency (K.-J.M.), the Center for Innovative Medicine (CIMED, grant no. 20200680 to N.M.), the Tornspiran Foundation (N.M.), the Karolinska Institutet (K.-J.M.), the Research Council of Norway (grant nos 275469 and 237579 to K.-J.M.), Center of Excellence: Precision Immunotherapy Alliance (grant no. 332727 to K.-J.M.), the Norwegian Cancer Society (grant nos 190386 and 223310 to K.-J.M.), the South-Eastern Norway Regional Health Authority (grant nos 2021-073 and 2024-053 to K.-J.M.), EU H2020-MSCA Research and Innovation program (grant no. 801133 to K.-J.M.), the Knut and Alice Wallenberg Foundation (grant no. 2018.0106 to K.-J.M.), the Swedish Foundation for Strategic Research (K.-J.M.) and the US National Cancer Institute (grant nos P01 CA111412 and P009500901 to K.-J.M. and R21AI130760 to A.H.).

### Author contributions

J.P.G., A.H. and A.P. performed the scRNA-seq experiments. A.P. performed the in vitro experiments and analysis. N.M., D.B. and J.M. provided the data on patients with NSCLC (flow cytometry). H.N. performed the computations. H.N. and A.P. performed the bioinformatic analysis. E.S., T.C., O.D., S.A.T., A.H. and K.-J.M. provided scientific input. A.P. wrote the manuscript with support from H.N., A.H., and K.-J.M. All authors edited the manuscript.

### Funding

### Competing interests

J.P.G. is an employee at Fate Therapeutics. T.C. is an employee at NEC OncoImmunity AS. K.-J.M. is a consultant at Fate Therapeutics and Vycellix and has research support from Fate Therapeutics, Oncopeptides for studies unrelated to this work. O.D. has received research funding from Gilead Sciences and Incyte, unrelated to this work, and personal fees from Sanofi, unrelated to this work. S.A.T. is a scientific advisory board member of ForeSite Labs, QIAGEN and Element Biosciences, and a co-founder and equity holder of TransitionBio and EnsoCell Therapeutics, and a part-time employee

of GlaxoSmithKline. A.H. has received funding from Astra Zeneca/ MedImmune, unrelated to this work. A.H. is a consultant at Purple BioTech. The other authors declare no competing interests.

## Additional information

**Extended data** is available for this paper at https://doi.org/10.1038/s41590-024-01884-z.

**Correspondence and requests for materials** should be addressed to Aline Pfefferle or Karl-Johan Malmberg.

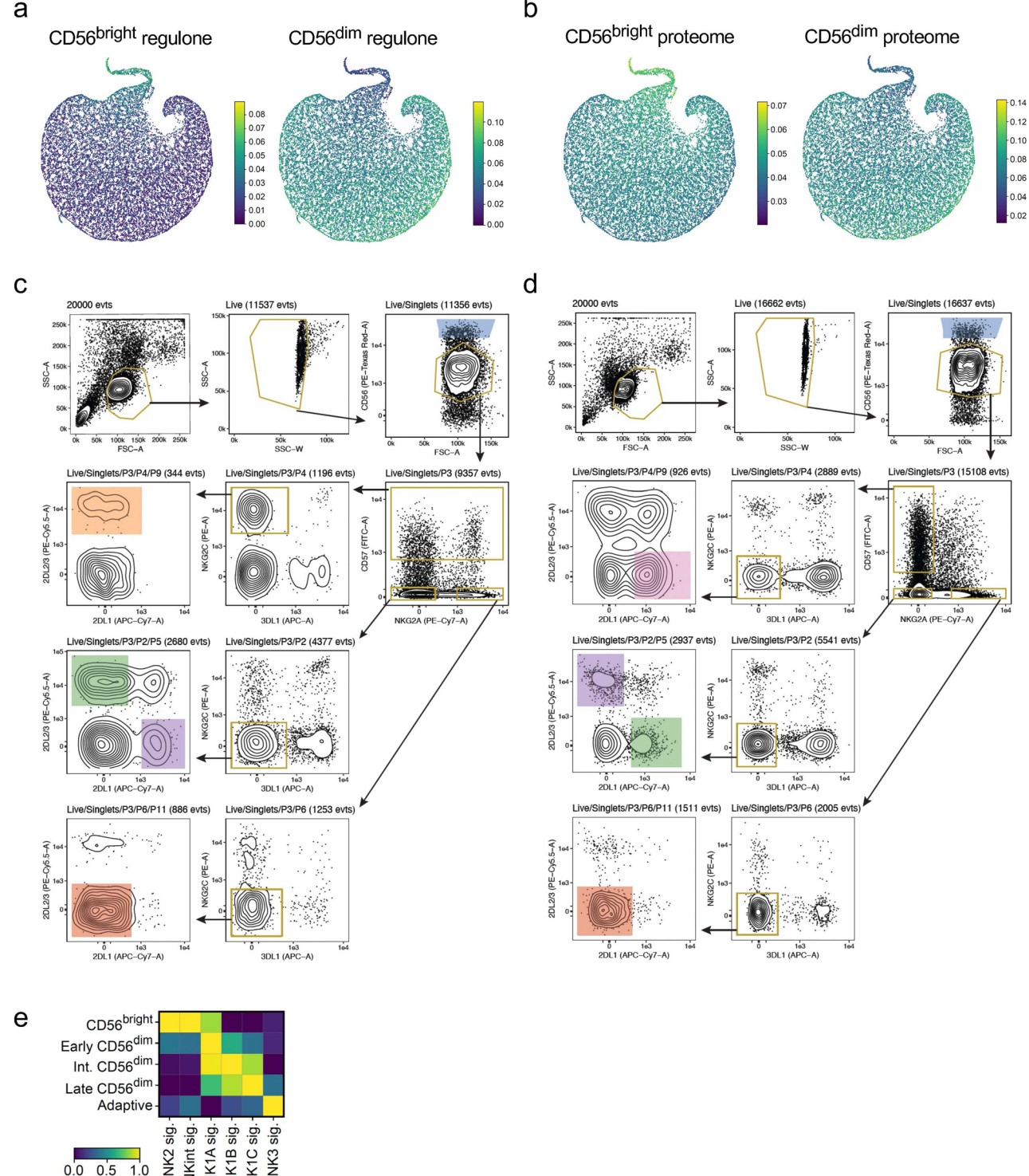

**Extended Data Fig. 1 | Peripheral blood NK cell subsets and sorting strategy.**
(**a**, **b**) AUCell scores of gene signatures for CD56^bright and CD56^dim NK regulones (**a**) and proteomes (**b**). (**c**, **d**) Sorting strategy for phenotypically defined functional PB-NK cell subsets sequenced in one donor with (**c**) and one without (**d**) an adaptive expansion. (**e**) Heatmap depicting similarity between our five annotated transcriptional NK cell subsets (y-axis) and the Meta-NK defined NK subsets (x-axis). The scale represent gene set activity calculated by AUCell (**a-b**, **e**).

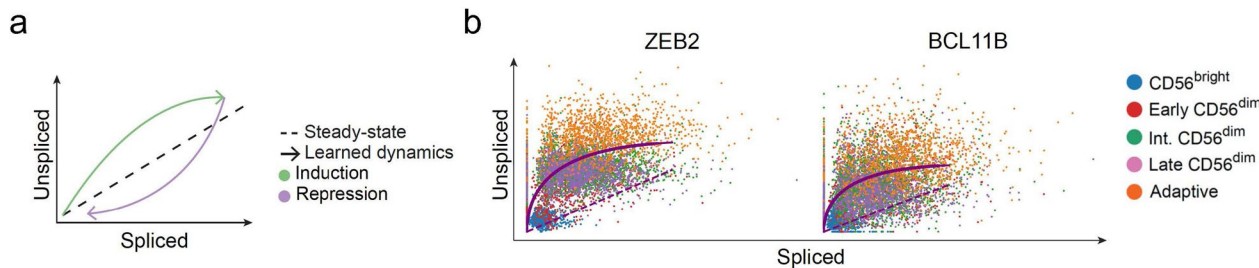

**Extended Data Fig. 2 | RNA velocity.** (**a**) Graphical depiction of inferring RNA velocity based on spliced vs unspliced transcripts. (**b**) RNA velocity plots for ZEB2 and BCL11B transcripts stratified by subset annotation in donors with an adaptive expansion.

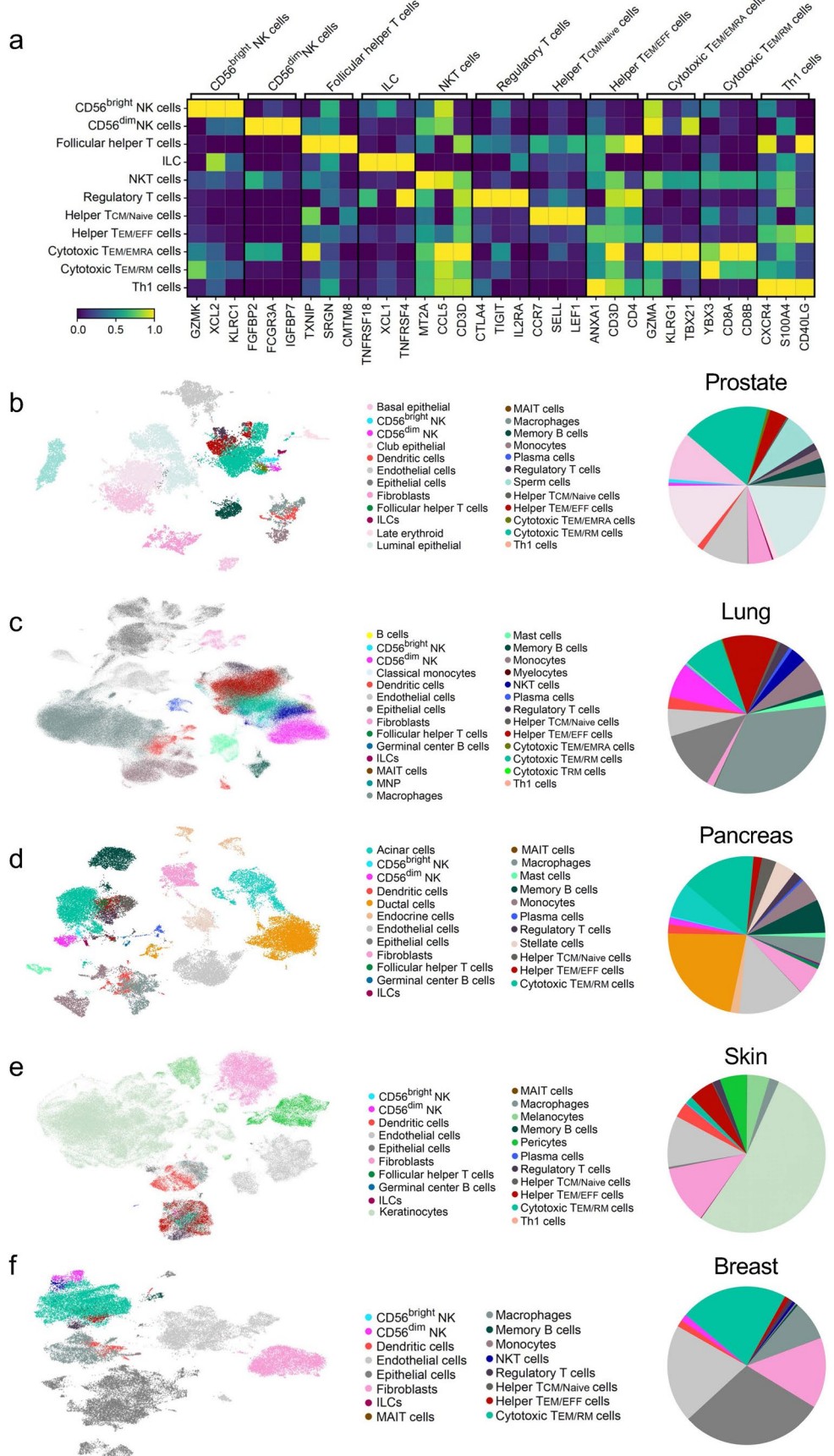

**Extended Data Fig. 3 | Healthy tissue dataset annotation using CellTypist.**
(**a**) Heatmap depicting expression of signature genes of the main immune populations annotated by CellTypist across all tissue samples. (**b-f**) UMAP representation showing integration of all healthy tissue datasets, prostate (**b**), lung (**c**), pancreas (**d**), skin (**e**), breast (**f**), with individual cell subtypes annotated using CellTypist.

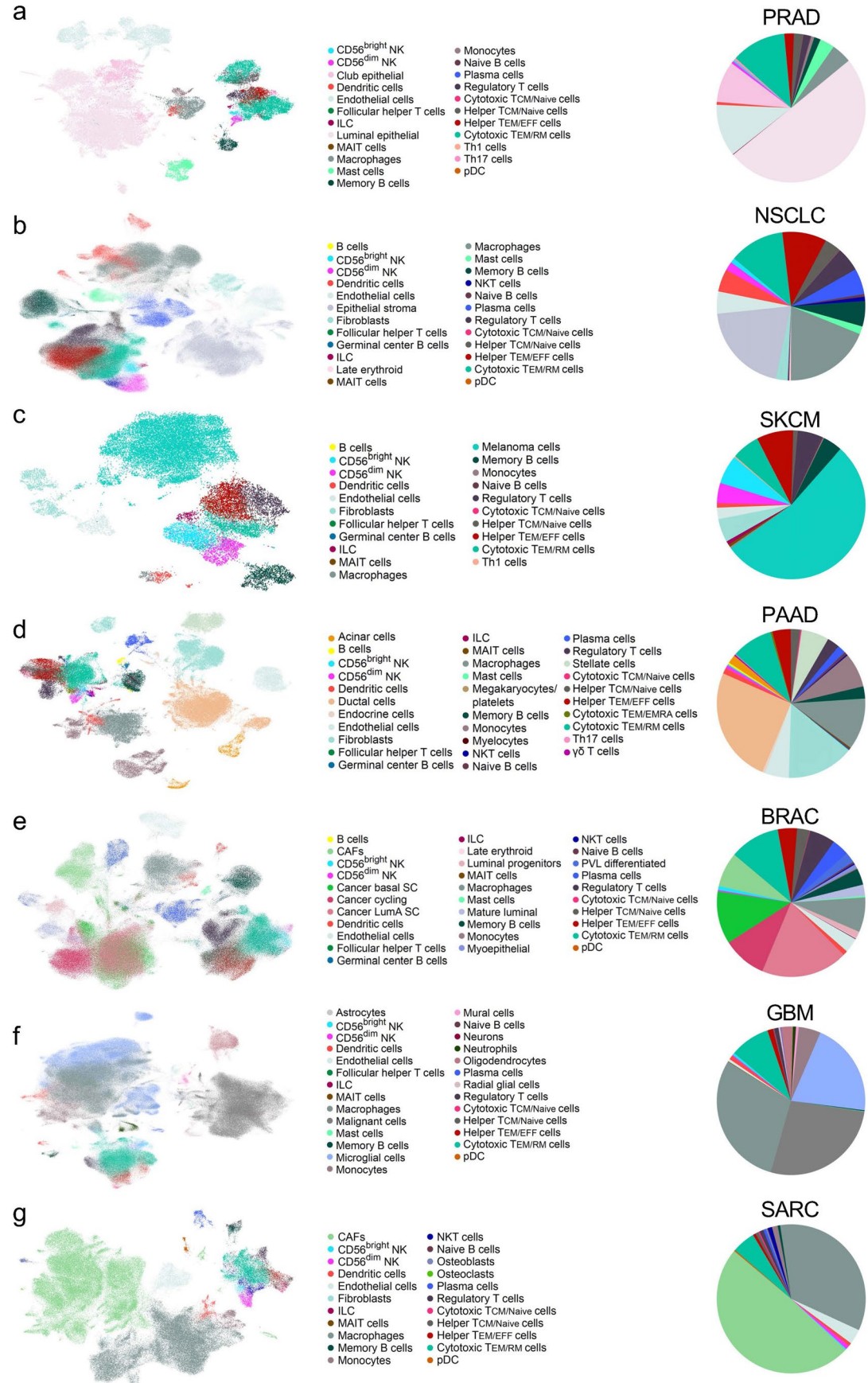

**Extended Data Fig. 4 | Solid tumor dataset annotation using CellTypist.** (**a-g**) UMAP representation showing integration of all solid tumor datasets, PRAD (**a**), NSCLC (**b**), SKCM (**c**), PAAD (**d**), BRAC (**e**), GBM (**f**), SARC (**g**), with individual cell subtypes annotated using CellTypist.

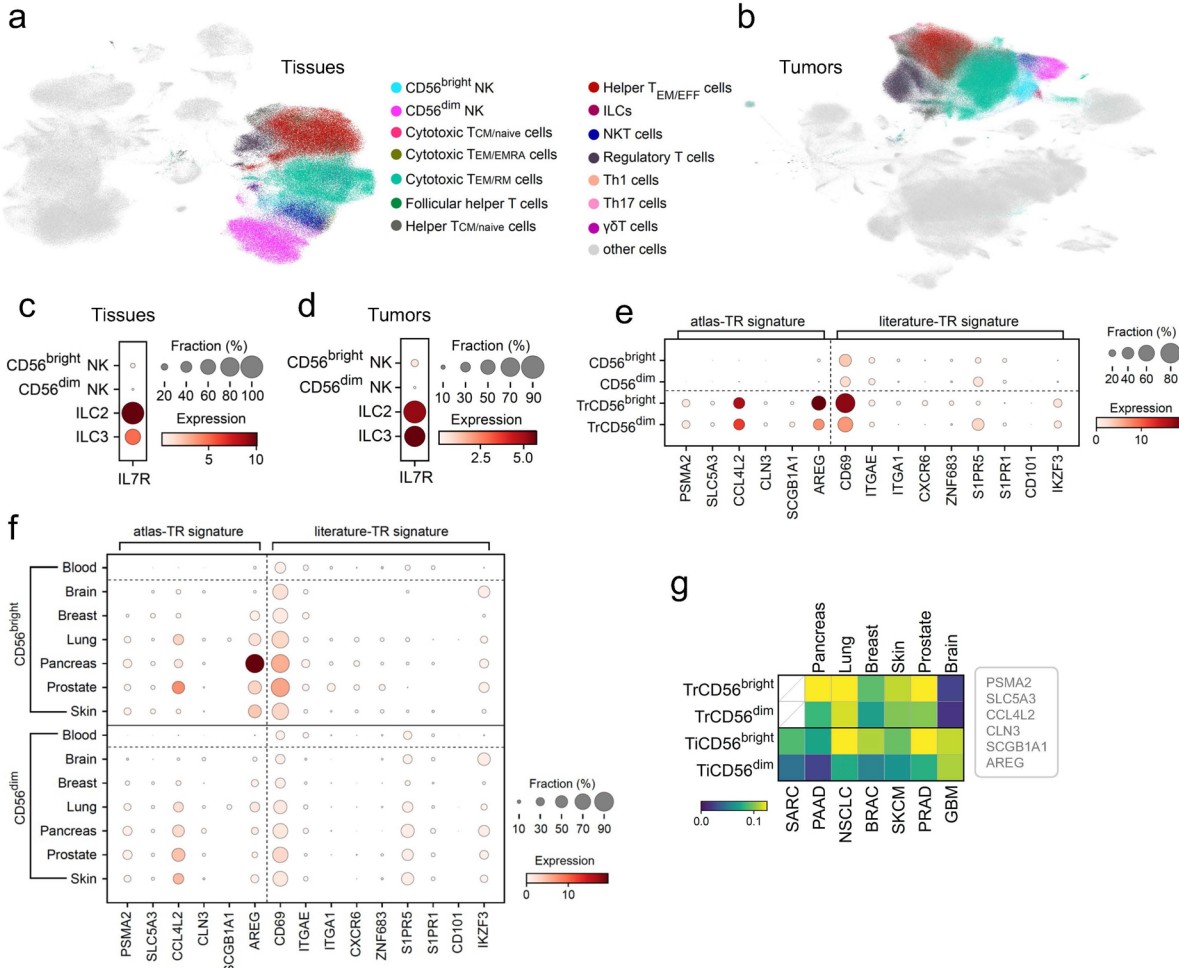

**Extended Data Fig. 5 | Tissue-residency scoring of NK cells. (a, b)** UMAP representation showing integration of all healthy tissue (**a**) and solid tumor (**b**) datasets, with lymphocytes populations visualized. (**c, d**) IL7R expression in annotated NK cells (CD56bright, CD56dim) and ILCs (ILC2, ILC3) in tissues (**c**) and tumors (**d**). (**e, f**) Dotplots depicting expression of genes defining the literature-TR and atlas-TR signatures in CD56bright and CD56dim subsets in healthy blood and across all tissue types (**e**) and stratified by individual tissues (**f**). (**g**) Tissue-residency scoring (atlas-TR) of CD56bright and CD56dim annotated NK cells in individual tissue and tumor types. The scale represents gene set activity calculated by AUCell (**g**).

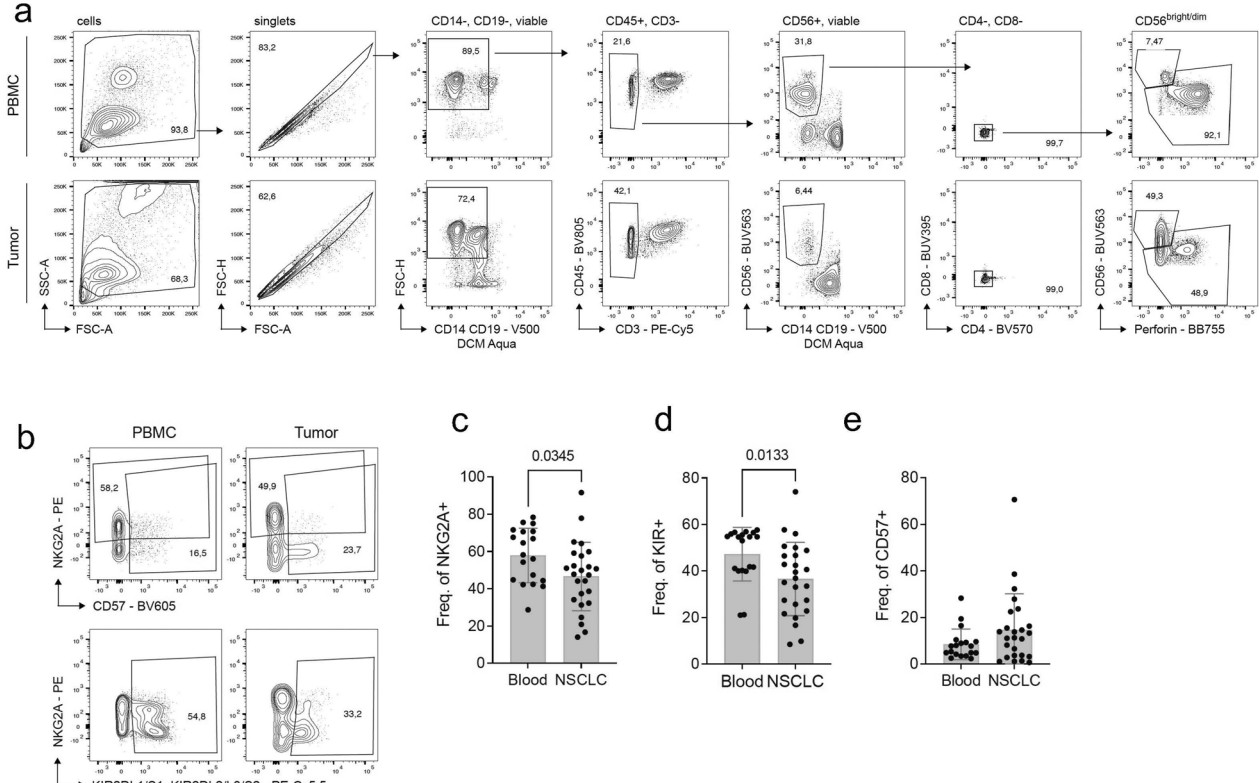

**Extended Data Fig. 6 | Phenotyping of NK cells in NSCLC patient samples.**
(**a**) Gating strategy for CD56^bright and CD56^dim NK cells in healthy donor (PBMC) and NSCLC samples (Tumor). (**b-e**) Representative plots of CD56^dim NK cells (**b**) and quantification of NGK2A (**c**), KIR (**d**) and CD57 (**e**) expression in PBMC (n = 19) and NSCLC samples (n = 25), from 23 independent experiments. Data were analyzed using two-tailed Mann-Whitney test (**c-e**). All bar graphs represent the mean ± s.d. Actual p values are indicated.

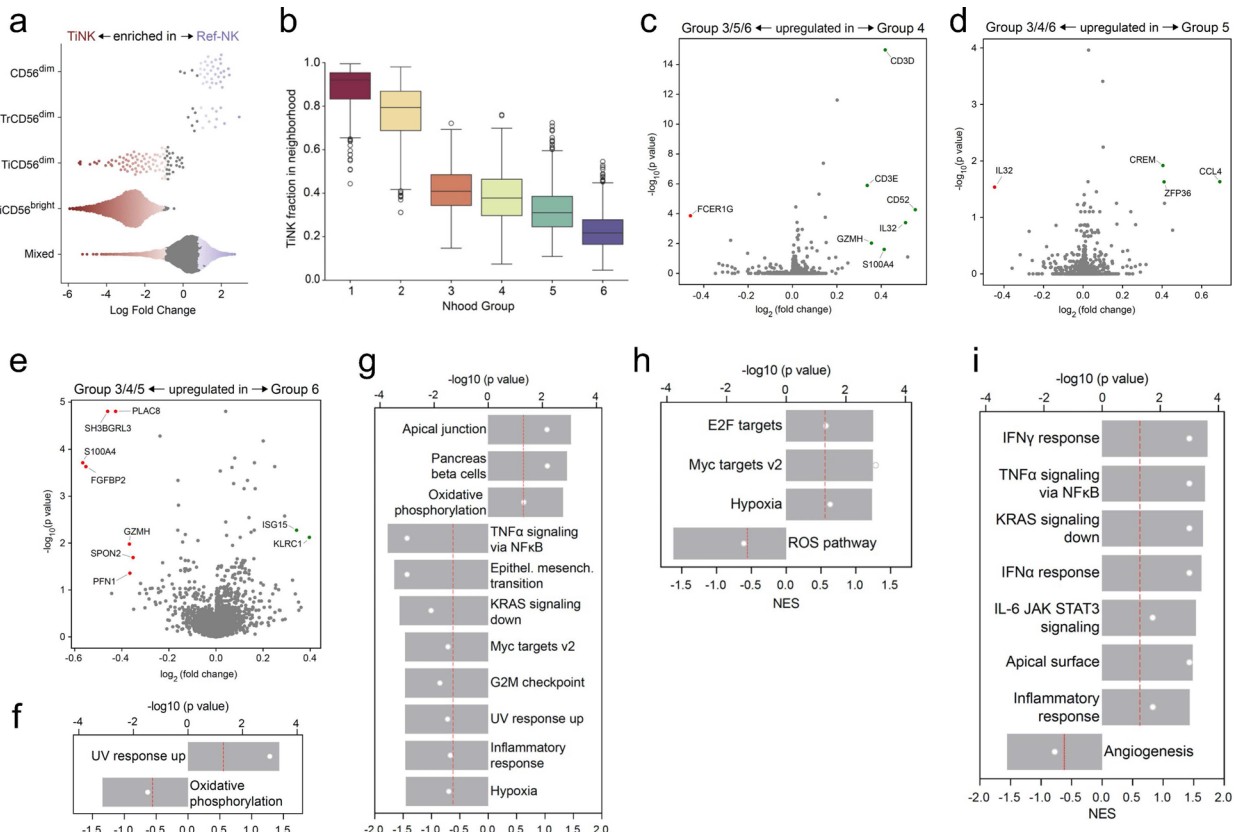

**Extended Data Fig. 7 | Characterization of cellular states of NK cells identified in pan-cancer cell atlas. (a)** Beaswarm plot depicting differential abundance of TiNK or Ref-NK (PB-NK, TrNK) enriched neighborhoods, clustered based on subset annotation of individual neighborhoods. **(b)** TiNK fraction of cells in neighborhoods within each neighborhood group. The boxplot indicates the median with the interquartile range (IQR), whiskers extend to the farthest point within 1.5 times the IQR from the box. n is the number of neighborhoods in each group: group 1, n = 382; group 2, n = 1261; group 3, n = 1239; group 4, n = 871; group 5, n = 1427; group 6, n = 1752. **(c-e)** Volcano plots depicting

differentially expressed genes (DEGs) between Group 4 vs. Group 3/5/6 (**c**), Group 5 vs. Group 3/4/6 (**d**), Group 6 vs. Group 3/4/5 (**e**). Differential expression analysis was performed using the findNhoodGroupMarkers method within the miloR package. Counts were aggregated per sample; groups were compared using edgeR and the adjusted p-values were used for the plots. **(f-i)** Gene set enrichment analysis (GSEA) for DEGs identified between Group 1 vs Group 2 (**f**), Group 3 vs Group 4/5/6 (**g**), Group 5 vs Group 3/4/6 (**h**) and Group 6 vs Group 3/4/5 (**i**). Volcano plots: log fold change cutoff at 0.5, p < 0.05. GSEA plots: p value cutoff 0.5 (red line).

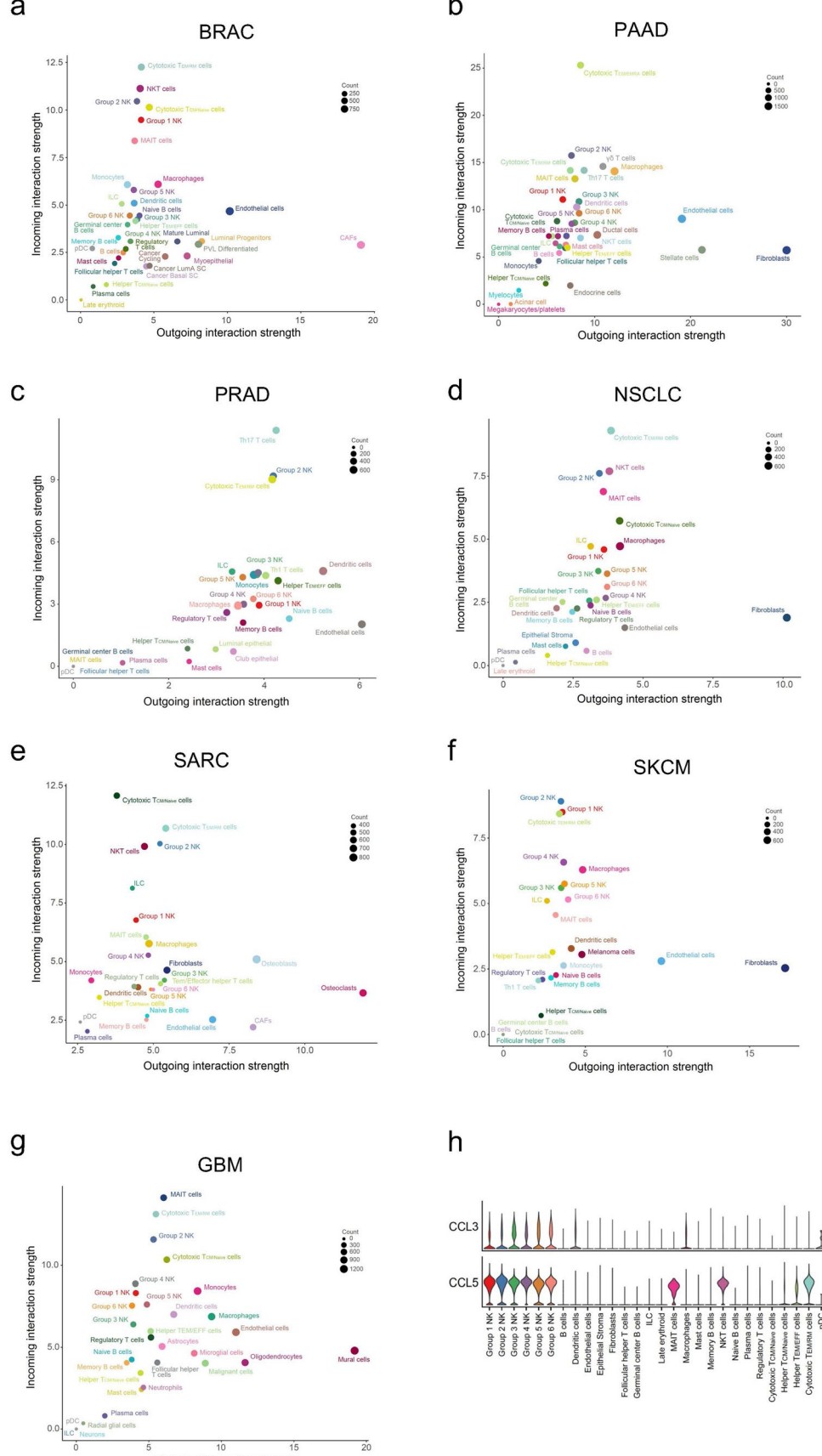

**Extended Data Fig. 8 | Intercellular communication in TME. (a-g)** Scatterplot depicting incoming and outgoing interaction strength of individual cell types in BRAC (**a**), PAAD (**b**), PRAD (**c**), NSCLC (**d**), SARC (**e**), SKCM (**f**), GBM (**g**) as identified by CellChat. (**h**) Violin plots showing expression of ligands for the CCL (CCL3, CCL5) communication pathway in NSCLC.

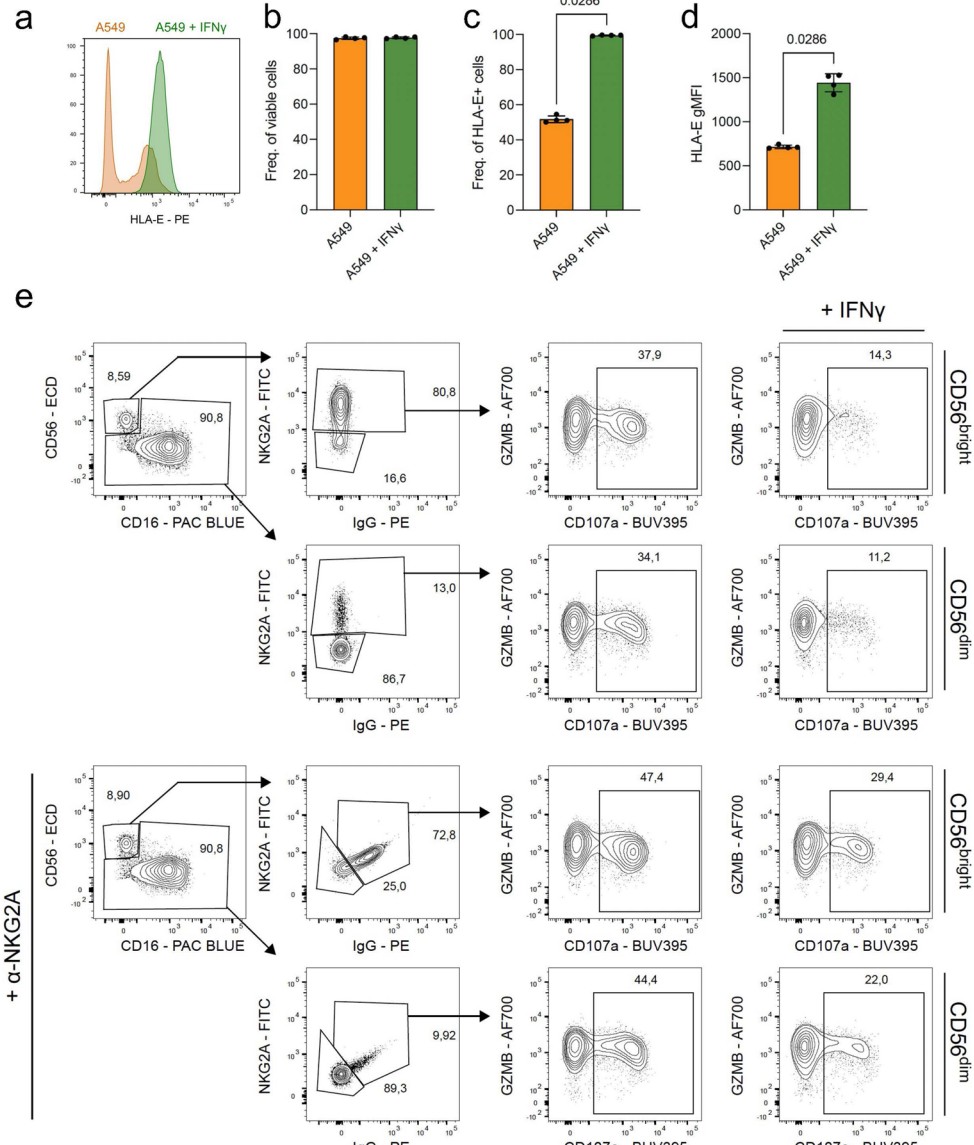

**Extended Data Fig. 9 | In vitro validation of IFNG-HLA-E-KLRC1 axis in NSCLC.** (**a**) Representative histogram of HLA-E expression of A549 cells pre-treated (24 h) with and without IFNγ. (**b-d**) Viability (**b**), frequency (**c**) and geometric MFI (**d**) of HLA-E + A549 cells (n = 4, biological replicates from two independent experiments). (**e**) Gating strategy and representative contour plots for functional readout of CD56^bright and CD56^dim NK cells against A549 target cells pre-treated with and without IFNγ (24 h) in presence and absence of α-NKG2A antibody (E:T 1:1, 4 h). Data were analyzed using two-tailed Mann-Whitney test (**b-d**). All bar graphs represent the mean ± s.d. Actual p values are indicated.

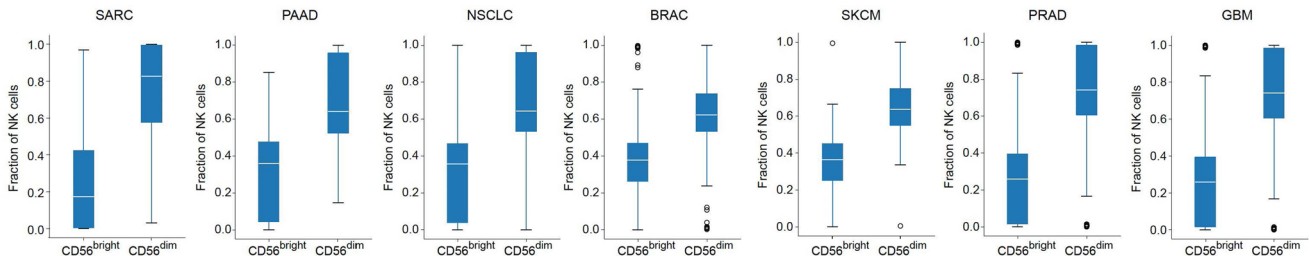

**Extended Data Fig. 10 | Deconvolution of TCGA datasets.** Distribution of CD56[bright] and CD56[dim] NK cells in deconvoluted TCGA datasets. The boxplots indicates the median with the interquartile range (IQR), whiskers extend to the farthest point within 1.5 times the IQR from the box. For each plot and each subset n is the number of patients for each tumor type: SARC, n = 88; PAAD, n = 183; NSCLC, n = 600; BRAC, n = 1231; SKCM, n = 473; PRAD, n = 554; GBM, n = 175.

# Reporting Summary

## Statistics

For all statistical analyses, confirm that the following items are present in the figure legend, table legend, main text, or Methods section.

| n/a | Confirmed | |
|---|---|---|
| ☐ | ☒ | The exact sample size (*n*) for each experimental group/condition, given as a discrete number and unit of measurement |
| ☐ | ☒ | A statement on whether measurements were taken from distinct samples or whether the same sample was measured repeatedly |
| ☐ | ☒ | The statistical test(s) used AND whether they are one- or two-sided<br>*Only common tests should be described solely by name; describe more complex techniques in the Methods section.* |
| ☐ | ☒ | A description of all covariates tested |
| ☐ | ☒ | A description of any assumptions or corrections, such as tests of normality and adjustment for multiple comparisons |
| ☐ | ☒ | A full description of the statistical parameters including central tendency (e.g. means) or other basic estimates (e.g. regression coefficient) AND variation (e.g. standard deviation) or associated estimates of uncertainty (e.g. confidence intervals) |
| ☐ | ☒ | For null hypothesis testing, the test statistic (e.g. $F$, $t$, $r$) with confidence intervals, effect sizes, degrees of freedom and $P$ value noted<br>*Give P values as exact values whenever suitable.* |
| ☐ | ☒ | For Bayesian analysis, information on the choice of priors and Markov chain Monte Carlo settings |
| ☒ | ☐ | For hierarchical and complex designs, identification of the appropriate level for tests and full reporting of outcomes |
| ☐ | ☒ | Estimates of effect sizes (e.g. Cohen's *d*, Pearson's *r*), indicating how they were calculated |

*Our web collection on statistics for biologists contains articles on many of the points above.*

## Software and code

Policy information about availability of computer code

| Data collection | R/Bioconductor package TCGAbiolinks (v 2.25.3) |
|---|---|
| Data analysis | 10x Genomics Cell Ranger (v 7.0.0), r-base (v 4.2.3), python (v 3.9.16), scvi (v 0.20), CellChat (v 1.6.1), survival (v 3.5-5), survminer (0.4.9), scanpy (v 1.9.3), scikit-learn (v 1.2.2), scvelo (v 0.2.5), palantir (v 1.2), numpy (v 1.23.5), pyscenic (v 0.12.1), phenograph (v 1.5.7), arboreto (v 0.1.6), anndata (v 0.9.1), GSEA (v 4.2.3), edgeR (v 3.40.2), miloR (v 1.7.1), BayesPrism (v 2.0), decontX (v 1.0.0), Tangram (v 1.0.4), CellTypist (v 1.6.2), SOLO (v 1.0)<br><br>Code from our own analysis have been made available on GitHub: https://github.com/hernet/transcriptional-map-nk |

For manuscripts utilizing custom algorithms or software that are central to the research but not yet described in published literature, software must be made available to editors and reviewers. We strongly encourage code deposition in a community repository (e.g. GitHub). See the Nature Portfolio guidelines for submitting code & software for further information.

## Data

All manuscripts must include a data availability statement. This statement should provide the following information, where applicable:

- Accession codes, unique identifiers, or web links for publicly available datasets
- A description of any restrictions on data availability
- For clinical datasets or third party data, please ensure that the statement adheres to our policy

The gene expression data generated for this paper is available at NCBI GEO with accession number GSE245690 and raw sequencing data is available at EGA with accession number EGAS50000000014. The details about the publicly available data included in the analysis are available in Supplemental tables S1, S2, S3 and S5. Processed data and models have also been made available on Zenodo (https://zenodo.org/doi/10.5281/zenodo.8434223) and as an online resource at http://nk-scrna.malmberglab.com/. For GSEA the Molecular Signature Database (v2023.2.Hs) available at https://www.gsea-msigdb.org/gsea/msigdb/ was used. Relevant gene sets for scoring were also retrieved from this database. Bulk RNA-seq data was downloaded from TCGA and TARGET. Curated survival data was downloaded from Xena.

## Research involving human participants, their data, or biological material

Policy information about studies with human participants or human data. See also policy information about sex, gender (identity/presentation), and sexual orientation and race, ethnicity and racism.

| | |
|---|---|
| Reporting on sex and gender | Donors are anonymous and sex/gender is not discussed. |
| Reporting on race, ethnicity, or other socially relevant groupings | Donors are anonymous and race/ethnicity/social groupings are not discussed. |
| Population characteristics | Donors are anonymous. |
| Recruitment | Peripheral mononuclear cells (PBMC) were isolated using density gradient centrifugation from anonymized healthy blood donors (Oslo University Hospital; Karolinska University Hospital) with informed consent. |
| Ethics oversight | The study was approved by the regional ethics committee in Norway (Regional etisk komité (REK): 2018/2482) and Sweden (Regionala etikprövningsnämnden i Stockholm: 2016/1415-32, Etikprövningsmyndigheten: 2020-05289). |

Note that full information on the approval of the study protocol must also be provided in the manuscript.

# Field-specific reporting

Please select the one below that is the best fit for your research. If you are not sure, read the appropriate sections before making your selection.

☒ Life sciences   ☐ Behavioural & social sciences   ☐ Ecological, evolutionary & environmental sciences

For a reference copy of the document with all sections, see nature.com/documents/nr-reporting-summary-flat.pdf

# Life sciences study design

All studies must disclose on these points even when the disclosure is negative.

| | |
|---|---|
| Sample size | No sample size calculation was performed. Instead the sample size was chosen based on availability of material and published datasets at the time of study. |
| Data exclusions | No data was excluded from the analysis. Filtering and quality control of sequencing datasets are describe in the method section. |
| Replication | Different datasets were utilized as replicates for statistical analysis. More than 3 biologically independent replicates were used for all in vitro experiments. Data in Fig 4 f-g and Extended Data Fig. 6c-e is from 23 independent experiments, Fig. 6h is from 9 independent experiments, Fig. 6i is from 6 independent experiments, Fig. 6j-k from one independent experiment, Extended Data Fig. 9b-d is from 2 independent experiments |
| Randomization | Randomization was not relevant to this study as it involves the analysis of pre-existing datasets where conditions already have been applied. The analysis is also descriptive in nature and does not involve an intervention and there is no experimental manipulation to test. |
| Blinding | Blinding was not relevant to this study. Data was collected from existing datasets and the analysis aims to identify patterns and describe the transcriptional landscape of NK cells using computational methods, and blinding is not relevant in this context. |

# Behavioural & social sciences study design

All studies must disclose on these points even when the disclosure is negative.

| | |
|---|---|
| Study description | *Briefly describe the study type including whether data are quantitative, qualitative, or mixed-methods (e.g. qualitative cross-sectional, quantitative experimental, mixed-methods case study).* |
| Research sample | *State the research sample (e.g. Harvard university undergraduates, villagers in rural India) and provide relevant demographic information (e.g. age, sex) and indicate whether the sample is representative. Provide a rationale for the study sample chosen. For studies involving existing datasets, please describe the dataset and source.* |
| Sampling strategy | *Describe the sampling procedure (e.g. random, snowball, stratified, convenience). Describe the statistical methods that were used to predetermine sample size OR if no sample-size calculation was performed, describe how sample sizes were chosen and provide a rationale for why these sample sizes are sufficient. For qualitative data, please indicate whether data saturation was considered, and what criteria were used to decide that no further sampling was needed.* |
| Data collection | *Provide details about the data collection procedure, including the instruments or devices used to record the data (e.g. pen and paper, computer, eye tracker, video or audio equipment) whether anyone was present besides the participant(s) and the researcher, and whether the researcher was blind to experimental condition and/or the study hypothesis during data collection.* |
| Timing | *Indicate the start and stop dates of data collection. If there is a gap between collection periods, state the dates for each sample cohort.* |
| Data exclusions | *If no data were excluded from the analyses, state so OR if data were excluded, provide the exact number of exclusions and the rationale behind them, indicating whether exclusion criteria were pre-established.* |
| Non-participation | *State how many participants dropped out/declined participation and the reason(s) given OR provide response rate OR state that no participants dropped out/declined participation.* |
| Randomization | *If participants were not allocated into experimental groups, state so OR describe how participants were allocated to groups, and if allocation was not random, describe how covariates were controlled.* |

# Ecological, evolutionary & environmental sciences study design

All studies must disclose on these points even when the disclosure is negative.

| | |
|---|---|
| Study description | *Briefly describe the study. For quantitative data include treatment factors and interactions, design structure (e.g. factorial, nested, hierarchical), nature and number of experimental units and replicates.* |
| Research sample | *Describe the research sample (e.g. a group of tagged Passer domesticus, all Stenocereus thurberi within Organ Pipe Cactus National Monument), and provide a rationale for the sample choice. When relevant, describe the organism taxa, source, sex, age range and any manipulations. State what population the sample is meant to represent when applicable. For studies involving existing datasets, describe the data and its source.* |
| Sampling strategy | *Note the sampling procedure. Describe the statistical methods that were used to predetermine sample size OR if no sample-size calculation was performed, describe how sample sizes were chosen and provide a rationale for why these sample sizes are sufficient.* |
| Data collection | *Describe the data collection procedure, including who recorded the data and how.* |
| Timing and spatial scale | *Indicate the start and stop dates of data collection, noting the frequency and periodicity of sampling and providing a rationale for these choices. If there is a gap between collection periods, state the dates for each sample cohort. Specify the spatial scale from which the data are taken* |
| Data exclusions | *If no data were excluded from the analyses, state so OR if data were excluded, describe the exclusions and the rationale behind them, indicating whether exclusion criteria were pre-established.* |
| Reproducibility | *Describe the measures taken to verify the reproducibility of experimental findings. For each experiment, note whether any attempts to repeat the experiment failed OR state that all attempts to repeat the experiment were successful.* |
| Randomization | *Describe how samples/organisms/participants were allocated into groups. If allocation was not random, describe how covariates were controlled. If this is not relevant to your study, explain why.* |
| Blinding | *Describe the extent of blinding used during data acquisition and analysis. If blinding was not possible, describe why OR explain why blinding was not relevant to your study.* |

Did the study involve field work? 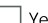 Yes 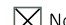 No

# Reporting for specific materials, systems and methods

We require information from authors about some types of materials, experimental systems and methods used in many studies. Here, indicate whether each material, system or method listed is relevant to your study. If you are not sure if a list item applies to your research, read the appropriate section before selecting a response.

## Materials & experimental systems

| n/a | Involved in the study |
|-----|------------------------|
| ☐ | ☒ Antibodies |
| ☐ | ☒ Eukaryotic cell lines |
| ☒ | ☐ Palaeontology and archaeology |
| ☒ | ☐ Animals and other organisms |
| ☒ | ☐ Clinical data |
| ☒ | ☐ Dual use research of concern |
| ☒ | ☐ Plants |

## Methods

| n/a | Involved in the study |
|-----|------------------------|
| ☒ | ☐ ChIP-seq |
| ☐ | ☒ Flow cytometry |
| ☒ | ☐ MRI-based neuroimaging |

## Antibodies

| | |
|---|---|
| Antibodies used | Flow cytometric analysis was performed with the following antibodies: PE-Cy7 mouse anti-human Perforin (eBioscience, deltaG9, cat # 25-9994-42, 1/100), PE goat anti-human IgG Fc Secondary Antibody (eBioscience, cat # 12-4998-82, 1/200), V500 mouse anti-human CD3 (BD Biosciences, UCHT1, cat # 561417, 1/100), V500 mouse anti-human CD14 (BD Biosciences, MφP9, cat # 561391, 1/100), V500 mouse anti-human CD19 (BD Biosciences, HIB19, cat # 561121, 1/100), Alexa Fluor 700 mouse anti-human Granzyme B (BD Biosciences, GB11, cat # 560213, 1/100), BUV395 mouse anti-human CD107a (BD Biosciences, H4A3, cat # 565113, 1/80), Pacific Blue mouse anti-human CD16 (BD Biosciences, 3G8, cat # 558122, 1/50), FITC mouse anti-human CD57 (BioLegend, HNK-1, cat # 359604, 1/50), Brilliant Violet 650 mouse anti-human CD38 (BioLegend, HB-7, cat # 356620, 1/50), Brilliant Violet 421 mouse anti-human CD158e1 (BioLegend, DX9, cat # 312714, 1/50), PE mouse anti-human HLA-E (BioLegend, 3D12, cat # 342604, 1/50), Brilliant Violet 650 mouse anti-human TNFa (BioLegend, Mab11, cat # 502938, 1/25), Brilliant Violet 785 mouse anti-human IFNg (BioLegend, 4S.B3, cat # 502542, 1/25), APC-Vio770 anti-human CD158a (Miltenyi Biotec, REA284, cat # 130-120-444, 1/10), PE-Vio770 mouse anti-human CD158a/h (Miltenyi Biotec, 11PB6, cat # 130-099-891, 1/10), PE anti-human CD159c (Miltenyi Biotec, REA205, cat # 130-119-776, 1/10), PE-Vio770 anti-human CD159a (Miltenyi Biotec, REA110, cat # 130-113-567, 1/10), VioBright FITC anti-human CD159a (Miltenyi Biotec, REA110, cat # 130-113-568, 1/100), PE-Cy5.5 mouse anti-human CD158b1/b2,j (Beckman Coulter, GL183, cat # A66900, 1/50), APC mouse anti-human CD159a (Beckman Coulter, Z199, cat # A60797, 1/25), ECD mouse anti-human CD56 (Beckman Coulter, N901, cat # A82943, 1/20), APC mouse anti-human CD158e1/e2 (Beckman Coulter, Z27.3.7, cat # A60795, 1/50), LIVE/DEAD Fixable Aqua Dead Stain kit, 405 nM (Life Technologies, cat # L34965, 1/200). |
| Validation | All antibodies used in this study were titrated on human PBMCs prior to usage. Validated staining was determined by FACS and compared to other validated antibodies. |

## Eukaryotic cell lines

Policy information about cell lines and Sex and Gender in Research

| | |
|---|---|
| Cell line source(s) | The A549 cell line was purchased from ATCC. |
| Authentication | The cell line was fingerprinted prior to usage. |
| Mycoplasma contamination | The cells are mycoplasma tested regularly (Eurofins). |
| Commonly misidentified lines (See ICLAC register) | No commonly misidentified cell lines were used in this study. |

## Palaeontology and Archaeology

| | |
|---|---|
| Specimen provenance | *Provide provenance information for specimens and describe permits that were obtained for the work (including the name of the issuing authority, the date of issue, and any identifying information). Permits should encompass collection and, where applicable, export.* |
| Specimen deposition | *Indicate where the specimens have been deposited to permit free access by other researchers.* |
| Dating methods | *If new dates are provided, describe how they were obtained (e.g. collection, storage, sample pretreatment and measurement), where they were obtained (i.e. lab name), the calibration program and the protocol for quality assurance OR state that no new dates are provided.* |

☐ Tick this box to confirm that the raw and calibrated dates are available in the paper or in Supplementary Information.

| Ethics oversight | *Identify the organization(s) that approved or provided guidance on the study protocol, OR state that no ethical approval or guidance was required and explain why not.* |
|---|---|

Note that full information on the approval of the study protocol must also be provided in the manuscript.

# Animals and other research organisms

Policy information about studies involving animals; ARRIVE guidelines recommended for reporting animal research, and Sex and Gender in Research

| Laboratory animals | *For laboratory animals, report species, strain and age OR state that the study did not involve laboratory animals.* |
|---|---|
| Wild animals | *Provide details on animals observed in or captured in the field; report species and age where possible. Describe how animals were caught and transported and what happened to captive animals after the study (if killed, explain why and describe method; if released, say where and when) OR state that the study did not involve wild animals.* |
| Reporting on sex | *Indicate if findings apply to only one sex; describe whether sex was considered in study design, methods used for assigning sex. Provide data disaggregated for sex where this information has been collected in the source data as appropriate; provide overall numbers in this Reporting Summary. Please state if this information has not been collected. Report sex-based analyses where performed, justify reasons for lack of sex-based analysis.* |
| Field-collected samples | *For laboratory work with field-collected samples, describe all relevant parameters such as housing, maintenance, temperature, photoperiod and end-of-experiment protocol OR state that the study did not involve samples collected from the field.* |
| Ethics oversight | *Identify the organization(s) that approved or provided guidance on the study protocol, OR state that no ethical approval or guidance was required and explain why not.* |

Note that full information on the approval of the study protocol must also be provided in the manuscript.

# Clinical data

Policy information about clinical studies
All manuscripts should comply with the ICMJE guidelines for publication of clinical research and a completed CONSORT checklist must be included with all submissions.

| Clinical trial registration | *Provide the trial registration number from ClinicalTrials.gov or an equivalent agency.* |
|---|---|
| Study protocol | *Note where the full trial protocol can be accessed OR if not available, explain why.* |
| Data collection | *Describe the settings and locales of data collection, noting the time periods of recruitment and data collection.* |
| Outcomes | *Describe how you pre-defined primary and secondary outcome measures and how you assessed these measures.* |

# Dual use research of concern

Policy information about dual use research of concern

## Hazards

Could the accidental, deliberate or reckless misuse of agents or technologies generated in the work, or the application of information presented in the manuscript, pose a threat to:

| No | Yes | |
|---|---|---|
| ☐ | ☐ | Public health |
| ☐ | ☐ | National security |
| ☐ | ☐ | Crops and/or livestock |
| ☐ | ☐ | Ecosystems |
| ☐ | ☐ | Any other significant area |

## Experiments of concern

Does the work involve any of these experiments of concern:

No Yes

☐ ☐ Demonstrate how to render a vaccine ineffective

☐ ☐ Confer resistance to therapeutically useful antibiotics or antiviral agents

☐ ☐ Enhance the virulence of a pathogen or render a nonpathogen virulent

☐ ☐ Increase transmissibility of a pathogen

☐ ☐ Alter the host range of a pathogen

☐ ☐ Enable evasion of diagnostic/detection modalities

☐ ☐ Enable the weaponization of a biological agent or toxin

☐ ☐ Any other potentially harmful combination of experiments and agents

# Plants

| | |
|---|---|
| Seed stocks | N/A |
| Novel plant genotypes | N/A |
| Authentication | N/A |

# ChIP-seq

## Data deposition

☒ Confirm that both raw and final processed data have been deposited in a public database such as GEO.

☒ Confirm that you have deposited or provided access to graph files (e.g. BED files) for the called peaks.

| | |
|---|---|
| Data access links<br>*May remain private before publication.* | Publicly available datasets were used. Relevant GEO accession numbers can be found in the supplemental tables. |
| Files in database submission | N/A |
| Genome browser session<br>(e.g. UCSC) | N/A |

## Methodology

| | |
|---|---|
| Replicates | N/A |
| Sequencing depth | N/A |
| Antibodies | N/A |
| Peak calling parameters | N/A |
| Data quality | N/A |
| Software | N/A |

# Flow Cytometry

## Plots

Confirm that:

☒ The axis labels state the marker and fluorochrome used (e.g. CD4-FITC).

☒ The axis scales are clearly visible. Include numbers along axes only for bottom left plot of group (a 'group' is an analysis of identical markers).

☒ All plots are contour plots with outliers or pseudocolor plots.

☒ A numerical value for number of cells or percentage (with statistics) is provided.

## Methodology

| | |
|---|---|
| Sample preparation | Peripheral mononuclear cells (PBMC) were isolated using density gradient centrifugation from anonymized healthy blood donors (Oslo University Hospital; Karolinska University Hospital) with informed consent (Norway: Regional etisk komité (REK): 2018/2482, Sweden: Regionala etikprövningsnämnden i Stockholm: 2016/1415-32, Etikprövningsmyndigheten: 2020-05289). PBMC were stained for surface antigens and viability in a 96 V-bottom plate, followed by fixation/permeabilization and intracellular staining at room temperature. |
| Instrument | Samples were acquired on an LSR-Fortessa equipped with a blue, red and violet laser or sorted using a FACSAriaII (Beckton Dickinson). |
| Software | Data was analyzed in FlowJo version 9 and 10 (TreeStar, Inc.). |
| Cell population abundance | All cell populations contained > 100 cells and 12,000 cells were sorted for each sample. |
| Gating strategy | Gating strategies are show in the supplemental figures. Restrictive gates were used to ensure clean sorted populations. Post-sort purity testing was performed. Single-color stains and fluorescence minus one (FMO) were used as controls to set PMT voltages. |

☒ Tick this box to confirm that a figure exemplifying the gating strategy is provided in the Supplementary Information.

# Magnetic resonance imaging

## Experimental design

| | |
|---|---|
| Design type | *Indicate task or resting state; event-related or block design.* |
| Design specifications | *Specify the number of blocks, trials or experimental units per session and/or subject, and specify the length of each trial or block (if trials are blocked) and interval between trials.* |
| Behavioral performance measures | *State number and/or type of variables recorded (e.g. correct button press, response time) and what statistics were used to establish that the subjects were performing the task as expected (e.g. mean, range, and/or standard deviation across subjects).* |

## Acquisition

| | |
|---|---|
| Imaging type(s) | *Specify: functional, structural, diffusion, perfusion.* |
| Field strength | *Specify in Tesla* |
| Sequence & imaging parameters | *Specify the pulse sequence type (gradient echo, spin echo, etc.), imaging type (EPI, spiral, etc.), field of view, matrix size, slice thickness, orientation and TE/TR/flip angle.* |
| Area of acquisition | *State whether a whole brain scan was used OR define the area of acquisition, describing how the region was determined.* |

Diffusion MRI   ☐ Used   ☐ Not used

## Preprocessing

| | |
|---|---|
| Preprocessing software | *Provide detail on software version and revision number and on specific parameters (model/functions, brain extraction, segmentation, smoothing kernel size, etc.).* |
| Normalization | *If data were normalized/standardized, describe the approach(es): specify linear or non-linear and define image types used for transformation OR indicate that data were not normalized and explain rationale for lack of normalization.* |
| Normalization template | *Describe the template used for normalization/transformation, specifying subject space or group standardized space (e.g. original Talairach, MNI305, ICBM152) OR indicate that the data were not normalized.* |

| Noise and artifact removal | *Describe your procedure(s) for artifact and structured noise removal, specifying motion parameters, tissue signals and physiological signals (heart rate, respiration).* |
|---|---|
| Volume censoring | *Define your software and/or method and criteria for volume censoring, and state the extent of such censoring.* |

## Statistical modeling & inference

| Model type and settings | *Specify type (mass univariate, multivariate, RSA, predictive, etc.) and describe essential details of the model at the first and second levels (e.g. fixed, random or mixed effects; drift or auto-correlation).* |
|---|---|
| Effect(s) tested | *Define precise effect in terms of the task or stimulus conditions instead of psychological concepts and indicate whether ANOVA or factorial designs were used.* |

Specify type of analysis: ☐ Whole brain  ☐ ROI-based  ☐ Both

| Statistic type for inference<br><br>(See Eklund et al. 2016) | *Specify voxel-wise or cluster-wise and report all relevant parameters for cluster-wise methods.* |
|---|---|
| Correction | *Describe the type of correction and how it is obtained for multiple comparisons (e.g. FWE, FDR, permutation or Monte Carlo).* |

## Models & analysis

| n/a | Involved in the study |
|---|---|
| ☐ | ☐ Functional and/or effective connectivity |
| ☐ | ☐ Graph analysis |
| ☐ | ☐ Multivariate modeling or predictive analysis |

| Functional and/or effective connectivity | *Report the measures of dependence used and the model details (e.g. Pearson correlation, partial correlation, mutual information).* |
|---|---|
| Graph analysis | *Report the dependent variable and connectivity measure, specifying weighted graph or binarized graph, subject- or group-level, and the global and/or node summaries used (e.g. clustering coefficient, efficiency, etc.).* |
| Multivariate modeling and predictive analysis | *Specify independent variables, features extraction and dimension reduction, model, training and evaluation metrics.* |

