## [Peer Review File · Nature Immunology]

Peer Review Information

Journal: Nature Immunology

Manuscript Title: Pan-cancer profiling of tumor-infiltrating natural killer cells through transcriptional reference mapping

Corresponding author name(s): Professor Karl-Johan Malmberg; Dr Aline Pfefferle

Reviewer Comments & Decisions:

Decision Letter, initial version:
--

6th Dec 2023

Dear Professor Malmberg,

As you know your Resource, "Pan-cancer profiling of tumor-infiltrating natural killer cells through transcriptional reference mapping" has now been seen by 2 referees. In light of their comments, and your recent email describing how you would address them we would be very interested in considering a revised version that addresses these concerns.

If you choose to revise your manuscript taking into account all reviewer and editor comments, please highlight all changes in the manuscript text file [OPTIONAL: in Microsoft Word format].

* If you have not done so already please begin to revise your manuscript so that it conforms to our Resource format instructions at <http://www.nature.com/ni/authors/index.html>. Refer also to any guidelines provided in this letter.

The Reporting Summary can be found here:

When submitting the revised version of your manuscript, please pay close attention to our [href="https://www.nature.com/nature-portfolio/editorial-policies/image-integrity">Digital Image Integrity Guidelines](https://www.nature.com/nature-portfolio/editorial-policies/image-integrity). and to the following points below:

[REDACTED]

If you wish to submit a suitably revised manuscript we would hope to receive it within 6 months. If you cannot send it within this time, please let us know. We will be happy to consider your revision so long as nothing similar has been accepted for publication at Nature Immunology or published elsewhere.

Nature Immunology is committed to improving transparency in authorship. As part of our efforts in this direction, we are now requesting that all authors identified as 'corresponding author' on published papers create and link their Open Researcher and Contributor Identifier (ORCID) with their account on the Manuscript Tracking System (MTS), prior to acceptance. ORCID helps the scientific community achieve unambiguous attribution of all scholarly contributions. You can create and link your ORCID from the home page of the MTS by clicking on 'Modify my Springer Nature account'. For more information please visit please visit www.springernature.com/orcid.

Thank you for the opportunity to review your work.

Sincerely,

Jamie D.K. Wilson, D.Phil
Chief Editor
Nature Immunology
212 726 9207
j.wilson@us.nature.com

Reviewers' Comments:

Reviewer #1:

Remarks to the Author:

In the resource "Pan-cancer profiling of tumor-infiltrating natural killer cells through 1 transcriptional reference mapping" the group of Malmberg have generated some highly useful reference maps of NK cell transcriptional state from both public and in-house datasets.

The paper is well written, logical and offers up a rich dataset for the field. My only general comment worth addressing with further work is that there could be more effort to highlight some biologically meaningful outputs or validation from the dataset. Even if speculative, it would be a nice value-add if some emerging questions of NK cell biology in the context of cancer immunotherapy were investigated further (as highlighted below)

Specific comments:

The authors found that TrNK and TiNK cells have a clear tissue residency signature but still share the dominant regulons of blood CD56bright and CD56dim NK cells, when defining "residency" are there any incoming signals that are discussed in Figure 8 that add to this definition? There is a dialogue around how "residency" is achieved and whether "residency signatures" are really "activation signatures" rather than defined by location.

A dysfunctional 'stressed' CD56bright state susceptible to TME-associated cellular communication and a cytotoxic 'effector' CD56dim state resistant to TME-associated cellular communication were commonly enriched across tumor types. Are there inputs or incoming signals from conventional suppressive sources (PGE2, A2, TGFb, IL-10) that in fact on this TME-associated cellular communication?

However, while adaptive donors NK cells continued their progression to intermediate CD56dim cells, terminating in the adaptive population, conventional donors instead branched into the intermediate or late CD56dim populations. What are some biological implications for this finding?

PRDM1 (BLIMP1) in mouse is a terminal TF (decreasing in late differentiation in this resource dataset) and typically associated with reduction in proliferation and cytokine responsiveness. Were these last 2 parameters inversely correlated with PRDM1 expression in these human datasets?

Regarding the relevance of this dataset in revealing novel actionable pathways in tumor-resident NK cells, It would be helpful to know how the 6 subsets change with age since most cancer patients are 50 years and above. Is there skewing of peripheral NK cells towards later subsets with age (skewing that follows the pseudotime predictions of differentiation in figure 2)? I would imagine that this impacts the quality and type of tumor infiltrating NK cell in elderly tumors versus younger patients but

I fear there might not be adequate data to look into this, even superficially? 427 patient scRNAseq data were analysed so I think some thought into this question could add to this resource utility. Even banking these into 25% eldest v 25% youngest of a similar tumor type?

Understanding of NK cell metabolism is growing and differential metabolic programs in NK cells from healthy tissue versus tumors, or from tumors with high NK cell infiltration (SKCM) versus low NK cell infiltration may shed light on NK cell fate in these circumstances. Can the metabolome be investigated further, again, superficially from these datasets to glean some novel learnings and outputs of the resource? They mention the "stressed state" having more metabolic activation (glycolysis, cholesterol homeostasis, fatty acid metabolism) but it could be insightful to dig a little deeper than GSEA and look at specific regulators of Ox Phos/mito stress v Glycolysis v fatty acid met v nutrient sensing for example.

There is a generalization of effector function to be cytotoxicity genes, but pre-clinical evidence suggests that NK cells might be an important source of chemokines that contributes to an inflamed TME. Is there evidence supporting this from looking at inflamed melanoma v excluded prostate cancer samples? There is interest in this question and some debate so a helpful biology output from this data could shed some light on this question. Specifically, can you segregate cytotoxic programs from chemokine programs in "effector" NK cells? Can this be prognostic?

Similarly, how does the chemokine receptor expression and signaling differentially impact the tumor-infiltrating NK cell subsets and states? How do they impact NK-rich/inflamed v NK-poor/desert? GPCRs such as CCR5, CCR2, CXCR3, CMKLR1 are putative regulators of NK cell infiltration into solid tumors so some analysis of these in the context of the resource would be helpful to the readers.

In figure 6, can one make any inference of incoming signals from with HLA-E on tumor versus HLA-E on NK cells and KLRC1/KLRD1 status? In NSCLC where maybe a signal is being observed with Monalizumab, is there stronger evidence of this incoming signal in TRANS v CIS?

Reviewer #2:

In this manuscript, Netskar and colleagues describe a set of bioinformatics analyses conducted on single-cell RNA sequencing data of NK cells extracted from peripheral blood, healthy tissues, and solid tumours. Employing knowledge of conventional stages of NK cell development in peripheral blood, defined by surface markers, the authors construct transfer learning models aimed at identifying distinct and common gene expression profiles. Subsequently, these models are applied to tissue- and tumour-derived NK cells, revealing their inherent ability to maintain either CD56bright or CD56dim expression patterns within their respective environments. Furthermore, they show that NK cells from these datasets can be categorised into six groups, where unique expression profiles can be linked to better survival in specific solid-tumour cancer types.

Overall, this paper provides an extensive analysis of NK cell transcriptional programs and has the potential to be a thoroughly useful resource for NK cell biologists. Collecting NK cell signatures from multiple tissues and tumours and showing that there are conserved programs is of high interest to the field, particularly for translational scientists looking to genetically modify NK cells for tailored treatments of cancers.

I have the following main concerns

1. The NK cell differentiation model proposed is not entirely supported by the analysis

In Figure 1c, the authors sort NK cell populations according to surface markers used in the field to define NK cell differentiation stages (CD56, NKG2A, self MHC specific (s)-KIR, CD57 and NKG2C). From this analysis it results that there is a considerable overlap of dimensional representation (and possibly of gene expression) between intermediate and late NK, questioning the ability of these markers to resolve different and subsequent stages of NK cell differentiation.

However, the findings from this experiment do not completely align with subsequent analyses in Figures 1D and 1E, where these populations are annotated to form separate clusters with no apparent major overlap. It is unclear which genes separate these populations, as these do not separate them in 1C. From which data this representation is generated? For example, how is the scVI representation in 1A, where there is no obvious cluster structure different from the plots in 1E? Signatures are shown in 1F, but this dot-plot contains several redundant and overlapping genes and can be more difficult to follow than a conventional dot-plot of each population and unique per-cluster genes.

The limitations of the proposed differentiation model are further highlighted by analysis in Figures 2C and 2D, where PAGA trajectory shows cells from the Early CD56dim population differentiating into either Intermediate CD56dim or Late CD56dim, with no description as to signatures that may influence this directionality. This phenomenon might relate to the large similarity between Intermediate or Late CD56dim. The authors should clarify these points and critically discuss the validity of the differentiation model proposed.

2. Validity of tissue cell-type annotation and tissue-specific NK cell signatures

The authors perform an extensive and impressive re-analysis of 136 scRNA-seq datasets (Figure 3A, detailed in supplementary figure 3), with the goal of extracting tumour NK cells for comparison with those in peripheral blood.

I have two major criticisms here:

A-The validity of cell lineage annotation in tissues (NK cells versus "ILC") needs to be ascertained. In Figure 3B, the authors use a combination of the prediction algorithm CellTypist (which model is used, is it the same for all tissue samples?), CITE-seq (where available) and their own clustering methods to identify cell types. Could the authors expand on how they define NK cells, in the context of the entire tissue dataset? What really separates NK cells from what is loosely defined as "ILCs"? Which ILC lineages, ILC3, ILC2, ILC1?

Some examples: in Figure 1F of peripheral NK cells, the authors show high expression of CST7, CMC1 and KLRB1 across all NK subsets. This is in contrast to where the authors now propose that TRDC, a primarily T cell-specific gene, is specific to CD56bright NK cells, a plausible difference for tissue NK cells. The presence of XCL1 being a unique marker of ILCs (ILC2? ILC3? Are there overlapping tissue signatures with NK?), which the authors have included in the NK CD56bright score from Figure 1B.

B-Tissue-specific NK cell signatures are underappreciated

By integrating the tissue-derived NK cell data with those from peripheral blood NK, I think the authors miss the chance here to highlight what would be most interesting to the readers: identifying reliably tissue-resident NK cells as well as the heterogeneity of the tissue-specific signatures.

The authors conclude from Figures 3E and 3F that there is a high degree of heterogeneity within tissues, but a conservation of either a CD56bright or CD56dim peripheral identity, shown by a measure of 'connectivity' but no specific gene signature overlap/divergence. The context of tissue NK cells within the total tissue dataset is critical, as placing any cell between the two states (here, CD56bright or CD56dim), will always place it somewhere between one of them. There should be more

detailed content on the signature behind the overlap and the divergence between tissue and PB NK cells, and on the signatures across different tissues. An integrative analysis combining all those datasets depicted in Supplementary Figure 3 could not only remove doubts from NK cell type annotation but also provide an extensive comparison of tissue-resident signatures that have been notoriously difficult to define for human NK cells versus ILC1s.

Finally, bearing in mind that this dataset is comprised of several donors, across many batches, labs and tissue origins - how is the data integrated, and how was this determined to be the best method of integration and representation?

3. Identification of stressed NK cells in tumours should be further validated

The presence of a 'stressed' NK population (neighbourhood 1 in figure 5A) could certainly be an interesting finding; however, several steps are needed to validate that this is not an artefactual finding (for example those found in 10.1038/s41593-022-01022-8):

- i. Are count matrices corrected for ambient RNA removal with a tool like CellBender, soupX or decontX?
- ii. Are tissue dissociation methods for skin different from those for solid tumours? The distribution of SKCM (Fig 3H) suggests that these NK cells either have a highly-specific signature, which will skew the representation and work against the method milo attempts to use, or that during dissociation of skin samples, a signature is generated which is highly different from other cancer types - both entirely plausible possibilities
- iii. Previous single cell RNA sequencing papers describe cell populations with up-regulation of genes for heat shock proteins (10.4110/in.2020.20.e34) and several reports even suggest dissociation methods which minimise these effects, specifically for tumours (10.1186/s13059-019-1830-0)

4. Relevance for tumour immunology

The focus of NK cells in tumours and tissues is the particular strength of this paper, and even with other criticisms above it should not be lost how valuable this analysis is. However, the final set of figures in 6B-E about cell interaction analysis remains the weakest part of this paper, particularly as it is a very straightforward piece of analysis within one R package and could be expanded.

The clarity of each of these plots' accuracy is questionable, as each ligand/receptor is measured by an arbitrary 'strength', and no validation is given of any of these interactions. If one of these pathways were validated in vitro, it would lend significant strength to the study.

Specific points:

The utility of the representation in Figure 3C is unclear. There have been 6 representations of peripheral blood NK cells by this point in the paper. Some more uniformity and clear description in each Figure legend of which data set is used for which representation would be very useful.

In Figure 5A, the authors utilise their final model to understand and interrogate inter-tissue and inter-tumour differences by using milo to detect neighbourhoods of cells in a cluster-free fashion. This method works by comparing intra-neighbourhood differences, instead of against each other, and is highly dependent on the data's preparation, including normalisation, variable features, representation method and number of neighbours used.

It is unclear how the authors determined that this is the optimal way to represent this data, which this method is so sensitive to. Would the results be different if they used the diffusion map, tSNE, or scVI representation?

Figure 6A shows survival curve analysis of 7 cancer types, split by whether they are enriched for

signatures of NK cells of neighbourhoods 1 or 3. It is unclear which genes are used to split these samples, such as differential gene expression between group 1 and group 3, or group 1 and everything else, etc. Are these NK cell-specific, or are they found in the wide tissue/tumour dataset?

Author Rebuttal to Initial comments

See inserted PDF

Dear editor and reviewers

We are grateful for the positive evaluation and thorough review of our pan-cancer NK cell reference atlas. During the past months we have addressed the technical questions and validated some of the key findings related to stress signatures and CellChat pathways. Key changes to the manuscript are described below and specifically in the point-by-point response.

Technical questions

Ambient RNA. We ran decontX, one of the tools suggested by reviewer #2, to adjust the count matrices for ambient RNA. decontX can run without empty droplet-data, allowing us to adjust the matrices also for the data where only the filtered count matrices have been made available. We adjusted the matrices for each sample for all the tissue and tumor types. The biggest differences between the adjusted and non-adjusted matrices were often seen in ribosomal and mitochondrial genes, and other genes that have not been important in the downstream analysis. However, since this step is upstream in the data analysis workflow, we had to recreate the whole atlas and redo all downstream analyses, including GRN networks, CellChat, Milo, Bayes prism deconvolution and survival analysis. As a consequence, many panels in Figure 3-6 have been slightly modified but all previous findings and interpretations could be corroborated. All models have been updated in the online resource.

Dissociation stress. We scored each sample for the dissociation related stress signature identified in one of the papers cited by reviewer #2 (10.1186/s13059-019-1830-0). From analysis of these scores we found no evidence that individual samples or tissue/tumor types are particularly affected by upstream processing of tissues. Specifically, we did not see a particularly high expression of these signatures in the melanoma and skin samples as opposed to the other tumor/tissue types. These samples also don't have different dissociation methods compared to the other tumor/tissue types. Perhaps more importantly, we could validate the cell type composition and presence of a strong Group 1 NK cell signature, including ROS signalling, hypoxia and stress response pathways in spatial RNA-seq data from melanoma and lung cancer samples, which have not undergone any tissue dissociation. New Fig. 7.

Data integration. Another computationally challenging effort during the revision was to integrate all immune cells across all tissues and tumors. We reintegrated the adjusted count matrices for each tumor and tissue type. As suggested, we also integrated all the immune cells across all tumors and all tissues and redid the cell type annotations for the integrated datasets using Celltypist v2. This led to some small difference in the cell type annotations. We then redid the rest of the downstream analysis using these new cell type labels. We also looked further into different ILC signatures and scored these on the cells that we had annotated as ILCs. See further details in the point-by-point response. We are grateful for this suggestion, that clearly has made the resource more powerful since it will allow studies of common patterns and those unique to specific tumor types. Perhaps the most striking outcome of this exercise was the definition of a new atlas-derived tissue residence signature that outperform the previous literature-based score in identifying both CD56^{bright} and CD56^{dim} NK cells in the tumor (Revised Fig. 3e and Extended Data Fig. 5e-g).

Validation

Given the amount of data integrated in this resource it would be impossible to follow all leads and dive into all facets of the resource. Likewise, it is not feasible to validate all the possible angles and pathways across all tissues and tumor types. After rebuilding the reference atlas, we choose to validate four aspects related to i) the composition, ii) the stress signature, iii) incoming signals in CellChat, iv) outgoing signals in the CellChat analysis.

In terms of validating the compositional analysis, we performed flow cytometry analysis on tumor infiltrating NK cells in 25 patients with lung cancer, with 19 healthy donors as control, and found that the subset distribution largely matched the one extracted from the resource, with an increased frequency of CD56^{bright} TiNK cells in central areas of the tumor and a similar subset distribution as the one derived

from scRNAseq (New Fig. 4f, g). We also analysed the deconvoluted myeloid and lymphocyte compartment in select spatial RNA-seq samples of GBM, NSCLC and SKCM, showing a very good match with the average composition in the resource (scRNAseq) (New Fig. 7a-b).

In terms of validating the stress signature, in addition to measures described above to exclude dissociation stress, we display a set of selected genes contributing to each of the programs, including the metabolism genes requested by Reviewer 1 (Revised Fig. 5). As described above, these same programs were high in Group 1 NK cells found in spatial transcriptomics data from SKCM (Fig. 7).

In terms of validating incoming signals, we choose to look closer at the NKG2A/HLA-E pathway in the context of anti-NKG2A (monalizumab biosimilar) therapy as suggested by Reviewer 1. This was one of the dominating incoming signals to CD56^{bright} TiNK cells in the CellChat analysis across all tumor types (Revised Fig. 6). Although, the NKG2A-HLA-E checkpoint is well described, its role in regulating CD56^{bright} NK cell responses and the potential reversal of this by therapeutic anti-NKG2A has not been formally addressed. We therefore performed functional experiments on PB-NK cells and show that degranulation and cytokine production by CD56^{bright} NK cells is shut down by IFN-treatment of lung cancer cell lines, resulting in HLA-E upregulation, and efficiently restored by anti-NKG2A treatment. Revised Fig. 6j-k and Extended data Fig. 9). These data are discussed in the context of significant outgoing signals specifically from Group 1 CD56^{bright} NK cells in the tumor, releasing IFN- γ (Revised Fig. 6e-g).

Finally, we found that one of the major outgoing signals from cytotoxic Group 3 CD56^{dim} NK cells was granzyme A. Although we cannot formally prove specific release of Granzyme A in the tumor we found that tumor infiltrating NK cells (primary NSCLC samples) displayed lower levels of granzyme A (Revised Fig. 6h-i). Notably, the granzyme A signature was also corroborated in spatial transcriptomics data (New Fig. 7e).

Reviewer #1

(Remarks to the Author)

In the resource “Pan-cancer profiling of tumor-infiltrating natural killer cells through 1 transcriptional reference mapping” the group of Malmberg have generated some highly useful reference maps of NK cell transcriptional state from both public and in-house datasets.

The paper is well written, logical and offers up a rich dataset for the field. My only general comment worth addressing with further work is that there could be more effort to highlight some biologically meaningful outputs or validation from the dataset. Even if speculative, it would be a nice value-add if some emerging questions of NK cell biology in the context of cancer immunotherapy were investigated further (as highlighted below)

Author response: We thank the reviewer for these positive remarks and for the careful review and helpful suggestions.

Specific comments:

1. The authors found that TrNK and TiNK cells have a clear tissue residency signature but still share the dominant regulons of blood CD56^{bright} and CD56^{dim} NK cells, when defining “residency” are there any incoming signals that are discussed in Figure 8 that add to this definition? There is a dialogue around how “residency” is achieved and whether “residency signatures” are really “activation signatures” rather than defined by location.

Author response: The discussion of tissue residency signatures was also brought up by reviewer 2, point 2b. We agree with both reviewers that the definition of tissue residency remains a bit vague and has so

far not been robustly defined for NK cells and is an intense area of research. We originally used a set of genes commonly associated with tissue residency in both T cells and NK cells to score tissue residency of immune cells in the tissue/tumor samples analysed in this resource (now referred to as literature-based tissue residence score “literature-TR signature”). However, after integration of all immune cells across all tissues, as requested by Reviewer 2, we made a new comparison of differentially expressed genes (DEG) between TrNK cells and PB-NK cells. This turned out to be a very powerful analysis and we identified several new DEGs, not previously used to score tissue residency in NK cells, which were enriched in both CD56^{bright} and CD56^{dim} NK cells in tissues. We were able to generate a solely NK-derived tissue-residency signature (atlas-TR: PSMA2, SLC5A3, CCL4L2, CLN3, SCGB1A1, AREG), which outperformed the conventional literature-derived TR signature across tissue and tumor type (Revised Fig 3e and Extended Data Fig 5e-g.). Expression of CCL4L2, encoding a chemokine which induces chemotaxis of CCR5 and CCR1-expressing cells, such as T cells, dendritic cells and macrophages, has previously been described in NK cells isolated from melanoma samples (ref 63). This represents an independent verification, as this dataset was not included in our study. These melanoma-infiltrating NK cells also exhibited high AREG expression, an EGF receptor ligand. Notably, upregulation of AREG has also been described in the setting of healthy and cirrhotic liver-resident NK cells (ref 64), a tissue type not included in our pan-cancer atlas. Intriguingly, SCGB1A1, a member of the secretoglobulin family, functions as a potent inhibitor of phospholipase A2 (ref 65), a well described immunosuppressive molecule contributing to development of the TME. Hence, it is tempting to speculate that secretion of the SCGB1A1-encoded protein could be another effector mechanism through which TiNK cells can positively contribute to remodeling of the TME. These new results are discussed on page 20.

2. A dysfunctional ‘stressed’ CD56bright state susceptible to TME-associated cellular communication and a cytotoxic ‘effector’ CD56dim state resistant to TME-associated cellular communication were commonly enriched across tumor types. Are there inputs or incoming signals from conventional suppressive sources (PGE2, A2, TGFb, IL-10) that in fact on this TME-associated cellular communication?

Author response: These suppressive pathways were not captured as dominant NK cell specific incoming signals in the CellChat analysis. However, TGFb signaling was increased in stressed Group 1 NK cells and key genes making up this score are now depicted in Revised Fig. 5k, as are genes related to prostaglandin signaling. In the revised manuscript we have described the stress signature in more detail and validated its presence in spatial transcriptomics data (New Fig. 7). Additionally, SCGB1A1 (part of the atlas-TR signature), encodes for a secreted protein that functions as a potent phospholipase A2 inhibitor and expression of which is commonly upregulated on Tr- and Ti-NK cells.

3. However, while adaptive donors NK cells continued their progression to intermediate CD56dim cells, terminating in the adaptive population, conventional donors instead branched into the intermediate or late CD56dim populations. What are some biological implications for this finding?

Author response: This is related to the concern raised by Reviewer 2, point 1 regarding the transcriptional similarity of intermediate and late NK cells in conventional donors where the repertoire has not undergone adaptive reprogramming following CMV infection. We do not have evidence to suggest a branching differentiation. These two differentiation stages within the CD56^{dim} subset are very similar transcriptionally. Nevertheless, unbiased clustering algorithms consistently identify three clusters within the CD56^{dim} subset, here referred to as early, intermediate, and late.

We revised the wording on Page 8-9: “However, while adaptive donor NK cells continued their progression to intermediate CD56^{dim} cells, terminating in the transcriptionally distinct adaptive population, conventional donors instead progressed towards intermediate/late CD56^{dim} populations (Fig. 2c-d).”

The functional consequences of this gradual progression through differentiation, with loss of NKG2A, acquisition of killer cell immunoglobulin-like receptors (KIR) and CD57, is well established in the field and carefully referenced in the introduction of the manuscript. In brief, early CD56^{dim} NK cells express NKG2A, are cytokine responsive, secrete high amounts of IFN γ , and show prominent proliferative responses. Intermediate CD56^{dim} NK cells have acquired KIR and are therefore functionally tuned (through education) by interactions with self HLA-class I. We have expanded the discussion of the biological implication of the established knowledge of the functional specialization of NK cells during differentiation. See page 19.

4. PRDM1 (BLIMP1) in mouse is a terminal TF (decreasing in late differentiation in this resource dataset) and typically associated with reduction in proliferation and cytokine responsiveness. Were these last 2 parameters inversely correlated with PRDM1 expression in these human datasets?

Author response: Trend 5 represents an average of all the genes that show a gradually increased expression over pseudotime. Within each trend there is a range of expression for individual TFs and genes. It is well established that cytokine responsiveness and proliferation is highest in CD56^{bright}s (beginning of pseudotime) and lowest in late CD56^{dim} NK cells (CD57+ and/or Adaptive NK cells at the end of pseudotime). Hence, PRDM1 show an expression pattern indicative of being inversely correlated with cytokine responsiveness and proliferation. We highlight this inverse correlation in the description of the results (Fig. 2g, PRDM1 regulon has been added) and cite a relevant paper describing the role of *BLIMP1/PRDM1* (Kallies, A. et al. A role for Blimp1 in the transcriptional network controlling natural killer cell maturation. Blood 117, 1869-1879 (2011)).

5. Regarding the relevance of this dataset in revealing novel actionable pathways in tumor-resident NK cells, it would be helpful to know how the 6 subsets change with age since most cancer patients are 50 years and above. Is there skewing of peripheral NK cells towards later subsets with age (skewing that follows the pseudotime predictions of differentiation in figure 2)? I would imagine that this impacts the quality and type of tumor infiltrating NK cell in elderly tumors versus younger patients but I fear there might not be adequate data to look into this, even superficially? 427 patient scRNAseq data were analysed so I think some thought into this question could add to this resource utility. Even banking these into 25% eldest v 25% youngest of a similar tumor type?

Author response: This is indeed an interesting question but as suspected by the reviewer, the age data in the compiled resource from 39 studies are too sparse to make any statements regarding how this impacts the quality and type of tumor infiltrating NK cell in elderly versus younger patients.

6. Understanding of NK cell metabolism is growing and differential metabolic programs in NK cells from healthy tissue versus tumors, or from tumors with high NK cell infiltration (SKCM) versus low NK cell infiltration may shed light on NK cell fate in these circumstances. Can the metabolome be investigated further, again, superficially from these datasets to glean some novel learnings and outputs of the resource? They mention the “stressed state” having more metabolic activation (glycolysis, cholesterol homeostasis, fatty acid metabolism) but it could be insightful to dig a little deeper than GSEA and look at specific regulators of Ox Phos/mito stress v Glycolysis v fatty acid met v nutrient sensing for example.

Author response: We agree with the reviewer and have performed a deeper analysis of NK cell metabolism in the resource and across the identified states. We find that the stressed CD56^{bright} cell state exhibited increased metabolic activation (glycolysis, cholesterol homeostasis, mTORC1) (Revised Fig. 5g, i, l).

7. There is a generalization of effector function to be cytotoxicity genes, but pre-clinical evidence suggests that NK cells might be an important source of chemokines that contributes to an inflamed TME. Is there evidence supporting this from looking at inflamed melanoma v excluded prostate cancer samples? There is interest in this question and some debate so a helpful biology output from this data

could shed some light on this question. Specifically, can you segregate cytotoxic programs from chemokine programs in “effector” NK cells? Can this be prognostic?

Author response: This is another great suggestion. We have revisited the CellChat analysis in the revised manuscript and now include an analysis of all seven tumors in Revised Fig. 6. To understand how NK cells contribute to shaping the TME via an immunomodulatory role, we focused our analysis on outgoing signaling largely restricted to NK cells. We identified three signaling pathways (CCL, PARs, IFN-II) through which NK cells predominantly communicated with dendritic cells, macrophages, fibroblasts and endothelial cells (Revised Fig. 6e-f). CCL3 and CCL5, expressed across all states, can lead to the recruitment of ACKR1, CCR1 and CCR4 expressing cells (Extended Data Fig 8h, Fig. 5n). These results are discussed on page 21.

8. Similarly, how does the chemokine receptor expression and signaling differentially impact the tumor-infiltrating NK cell subsets and states? How do they impact NK-rich/inflamed v NK-poor/desert? GPCRs such as CCR5, CCR2, CXCR3, CMKLR1 are putative regulators of NK cell infiltration into solid tumors so some analysis of these in the context of the resource would be helpful to the readers.

Author response: These are also very relevant questions that can be addressed in the current resource. However, in the present version of the resource manuscript, we have not investigated the entire ecosystem within the tumor to extract information on the relationships between chemokine receptor repertoires (on all cell types), cell type and subset distribution, NK cell states, and the hot/cold status of the tumor. This is a very exciting problem but it has not been feasible to perform this analysis properly within the timeframe of this revision. We therefore limited the analysis to the signals that stood out as top pathways in the CellChat analysis, eg input to CXCR4. Notably, we recently reported a thorough investigation of the chemokine receptor repertoire and migratory responses in NK cells at different stages of differentiation alongside an analysis of chemokine expression in solid tumors (Lachota et al. “Mapping the chemotactic landscape in NK cells reveals subset-specific synergistic migratory responses to dual chemokine receptor ligation”. EBioMedicine. 2023 Oct;96:104811.). We discuss this topic briefly in the revised manuscript with reference to this recent publication and the regulation of NK cell infiltration. See page 21.

9. In figure 6, can one make any inference of incoming signals from with HLA-E on tumor versus HLA-E on NK cells and KLRC1/KLRD1 status? In NSCLC where maybe a signal is being observed with Monalizumab, is there stronger evidence of this incoming signal in TRANS v CIS?

Author response: HLA-E is one of the dominating incoming (and outgoing) signals to NK cells derived from the CellChat analysis. As part of the general request to validate one or more of the inferred CellChat pathways, we focused on the HLA-E/NKG2A checkpoint in NSCLC. While we cannot make any robust statements on cis/trans regulation on single cells, NK cells and other lymphocytes in the TME express high levels of HLA-E and therefore contribute to the strength of this interaction in CellChat. Such interaction could theoretically be beneficial in terms of NK cell education but also problematic in the recognition of tumor cells (eg as a negative check-point). We analysed tumor infiltrating NK cells in 25 patients with NSCLC by flow cytometry and found that the repertoires closely resembled those derived from the scRNA-seq analysis with increased frequencies of CD56^{bright} NK cells.

Furthermore, Group 1 and 2 NK cells preferentially received inhibitory input via the MHC-I (HLA-E/KLRC1) pathway due to high KLRC1 expression on these cellular states (Fig. 6a, d). Inhibitory signaling via the HLA-E axis significantly inhibited degranulation and granzyme B release of both CD56^{bright} and CD56^{dim} NK cells when co-cultured with A549 (NSCLC) targets cells pre-stimulated with IFN γ to upregulate HLA-E expression (Fig. 6j-k, Extended Data Fig. 9a-e). Blocking the NKG2A/HLA-E axis, using an anti-NKG2A antibody, manifested in significant recovery of function, both degranulation and granzyme B release (Fig. 6j-k, Extended Fig. 9e).

Reviewer #2

(Remarks to the Author)

In this manuscript, Netskar and colleagues describe a set of bioinformatics analyses conducted on single-cell RNA sequencing data of NK cells extracted from peripheral blood, healthy tissues, and solid tumours. Employing knowledge of conventional stages of NK cell development in peripheral blood, defined by surface markers, the authors construct transfer learning models aimed at identifying distinct and common gene expression profiles. Subsequently, these models are applied to tissue- and tumour-derived NK cells, revealing their inherent ability to maintain either CD56^{bright} or CD56^{dim} expression patterns within their respective environments. Furthermore, they show that NK cells from these datasets can be categorised into six groups, where unique expression profiles can be linked to better survival in specific solid-tumour cancer types.

Overall, this paper provides an extensive analysis of NK cell transcriptional programs and has the potential to be a thoroughly useful resource for NK cell biologists. Collecting NK cell signatures from multiple tissues and tumours and showing that there are conserved programs is of high interest to the field, particularly for translational scientists looking to genetically modify NK cells for tailored treatments of cancers.

Author response: We thank the reviewer for these positive remarks and for the careful review and helpful suggestions.

I have the following main concerns

1. The NK cell differentiation model proposed is not entirely supported by the analysis
In Figure 1c, the authors sort NK cell populations according to surface markers used in the field to define NK cell differentiation stages (CD56, NKG2A, self MHC specific (s)-KIR, CD57 and NKG2C). From this analysis it results that there is a considerable overlap of dimensional representation (and possibly of gene expression) between intermediate and late NK, questioning the ability of these markers to resolve different and subsequent stages of NK cell differentiation.
However, the findings from this experiment do not completely align with subsequent analyses in Figures 1D and 1E, where these populations are annotated to form separate clusters with no apparent major overlap. It is unclear which genes separate these populations, as these do not separate them in 1C. From which data this representation is generated? For example, how is the scVI representation in 1A, where there is no obvious cluster structure different from the plots in 1E?

Author response: In response to this comment, we have modified the text/legend in Fig. 1 to better explain which representation is shown in each panel, which samples that are included and which labels that are used. First, we analyze the bulk NK data sets only and observe that we can integrate them and that the batch effects associated with the donor and lab are accounted for (Fig. 1a). We represent this also using diffusion map components as we believe that better captures the differentiation trajectory. Next, we trained a new model, a scANVI model, where we include both bulk NK and sorted subsets of NK cells (visualized in Fig. 1e). This model leverages cell type labels, in our case the subset labels, and in the resulting representation the cells belonging to the various subsets are better separated. The data displayed in Fig. 1e also contains many more cells/subset, specifically the CD56^{bright} subset since there are much fewer cells in the bulk, which also leads to the separation of the clusters/subsets being much clearer. We only use this scANVI model for annotation of our data and the downstream analysis relies on the bulk data with varying proportions of the individual subsets which then allows us to consider the subset composition of the data analyzed.

Signatures are shown in 1F, but this dot-plot contains several redundant and overlapping genes and can be more difficult to follow than a conventional dot-plot of each population and unique per-cluster genes.

Author response: Unlike analysis of different cell types, there are few (if any) unique per cluster genes defining stages of NK cell differentiation. We feel the current plot is the best way of depicting the subset transitions, since it reflects the outcome of the scANVI differential expression analysis, in terms of the most differentially expressed genes in all possible comparisons of subsets and not just the ones high in one subset compared to all others. We discuss these signatures in the context of a continuous differentiation with relatively small transcriptional changes between some of these subsets. Notably, this classification of intermediate stages matches those defined previously by phenotypes (Fig. 1c), and unbiased Leiden clustering (Fig. 1g).

The limitations of the proposed differentiation model are further highlighted by analysis in Figures 2C and 2D, where PAGA trajectory shows cells from the Early CD56^{dim} population differentiating into either Intermediate CD56^{dim} or Late CD56^{dim}, with no description as to signatures that may influence this directionality. This phenomenon might relate to the large similarity between Intermediate or Late CD56^{dim}. The authors should clarify these points and critically discuss the validity of the differentiation model proposed.

Author response: We think the reviewer is correct in suggesting that the branching tendency in the PAGA trajectory seen within the conventional donors, that have not undergone CMV-driven “terminal” differentiation into the adaptive state, is related to the large similarity between intermediate and late CD56^{dim} NK cells. This similarity is both evident from Fig. 1d as well as the PAGA analysis where we see high levels of connectivity between these two subsets. See also response to reviewer 1, comment 3. As suggested, we have revised the legend of Fig. 1, removed the unsupported statements of branching differentiation in the results section, and briefly discuss the implications of the model on the functional traits along the differentiation axis.

New wording on Page 8-9: “However, while adaptive donor NK cells continued their progression to intermediate CD56^{dim} cells, terminating in the transcriptionally distinct adaptive population, conventional donors instead progressed towards intermediate/late CD56^{dim} populations (Fig. 2c-d).”

2. Validity of tissue cell-type annotation and tissue-specific NK cell signatures

The authors perform an extensive and impressive re-analysis of 136 scRNA-seq datasets (Figure 3A, detailed in supplementary figure 3), with the goal of extracting tumour NK cells for comparison with those in peripheral blood.

I have two major criticisms here:

2A-The validity of cell lineage annotation in tissues (NK cells versus “ILC”) needs to be ascertained

In Figure 3B, the authors use a combination of the prediction algorithm CellTypist (which model is used, is it the same for all tissue samples?), CITE-seq (where available) and their own clustering methods to identify cell types. Could the authors expand on how they define NK cells, in the context of the entire tissue dataset?

Author response: In the submitted version of the manuscript, we used CellTypist (v1). After the upstream removal of ambient RNA and reintegration of the atlas, including a complete integration of all immune cells, we re-ran all cell type annotations, using CellTypist (v2), and all analyses downstream of the annotations. The CITE-seq data and our own clustering/scoring of CD56^{bright} and CD56^{dim} NK cells, works as a validation of the annotation. This is further strengthened in Fig. 4d-e, where we provide a fine-grained annotation of CellTypist-identified NK cells using our trained model for subset annotation based on FACS-sorted subsets. Altogether, we are confident that this approach captures diverse stages of NK cells in normal tissues and tumors.

What really separates NK cells from what is loosely defined as “ILCs”? Which ILC lineages, ILC3, ILC2, ILC1?

Author response: Primarily we rely on the annotations resulting from running the CellTypist prediction algorithm and “ILCs” are defined by the signature provided by the CellTypist model. The model does include signatures also for ILC1, ILC2 and ILC3, but we noticed that the general ILC signature scored highest for these cells. The cell type label “ILCs” in the revised Fig. **3b** is the cell type ILC from CellTypist (v2). To further interrogate the separation between ILCs and NK cells and to analyse the presence of ILC1, ILC2 and ILC3 in tissues and tumors, we performed a new subanalysis presented in Extended Data Fig. **5**. We extracted the cells annotated as ILCs from the integrated data set from tumors and tissues. We performed a clustering (Leiden) of these cells and scored them based on described ILC1, ILC2 and ILC3 signatures. Based on this scoring we labeled some cells as ILC2 and some cells as ILC3, while all the cells scored low on the ILC1 signature. The NK cells in our data scored low on all three signatures. When looking at the prediction probability of the ILC1, ILC2 and ILC3 signatures from CellTypist we also noticed that ILC2 and ILC3 scored the highest, while ILC1 scored very low. New Extended Data Fig. **5c-d** shows expression of a key marker of ILCs, IL7R, across the NK cells and identified ILC clusters.

Some examples: in Figure 1F of peripheral NK cells, the authors show high expression of CST7, CMC1 and KLRB1 across all NK subsets. This is in contrast to where the authors now propose that TRDC, a primarily T cell-specific gene, is specific to CD56bright NK cells, a plausible difference for tissue NK cells. The presence of XCL1 being a unique marker of ILCs (ILC2? ILC3? Are there overlapping tissue signatures with NK?), which the authors have included in the NK CD56bright score from Figure 1B.

Author response: In Fig. **1f**, CST7, CMC1 and KLRB1 are consistent across NK cell subsets, but these genes are not exclusive to NK cells. This is not necessary for the genes to be informative and contribute to gene signatures that separates NK cell subsets. Please see the response above about the use of the dot plot in Fig. **1f**. The performance of the prediction score based on these signatures, generated by the scANVI tool, is tested and shown in Fig. **1d**.

In the original version of Fig. **3b**, we highlighted three genes, from a longer list of signature genes, that showed up as markers for each of the immune cell types in the lung tumor data set only. As for all cell-typist derived gene signatures the individual genes are not uniquely expressed in each defined cell type. The signature represents the composition of genes and expression that together best define and separates each cell type from all other cell types. As proposed by the reviewer in comment 2Bii (below), we have now integrated all immune cell signatures across all tumors and tissues and then re-annotated all cell types using CellTypist (v2). We performed multiple comparisons to identify the marker genes to highlight. First, we compared NK, ILC, NKT and T cells. Then we compared the two NK cell subset to each other, and all the T cell subsets among themselves. This resulted in lists of markers for each cell type that were differentially expressed. We curated these lists to highlight genes we found most relevant (New Extended Data Fig. **3a**).

2B-Tissue-specific NK cell signatures are underappreciated

2Bi: By integrating the tissue-derived NK cell data with those from peripheral blood NK, I think the authors miss the chance here to highlight what would be most interesting to the readers: identifying reliably tissue-resident NK cells as well as the heterogeneity of the tissue-specific signatures.

Author response: To address the state of NK cells in the tumor relative to NK cells in normal tissues we felt it made most sense to create a reference of normality that include both peripheral blood NK cells and tissue-derived NK cells. We believe this is the only way to discriminate putative perturbations in the TME from “normal” tissue signatures. Nevertheless, we agree that it is a good idea to perform a deeper analysis of tissue resident NK cells and possibly use this comparison to define an even better NK cell specific tissue resident signatures and to capture underlying heterogeneity. Therefore, we performed a global DEG analysis of all integrated TrNK cells and compared these to PB-NK cell counterparts. We did this analysis for CD56^{bright} and CD56^{dim} NK cells separately. We were able to generate a solely NK-derived tissue-residency signature (atlas-TR: PSMA2, SLC5A3, CCL4L2, CLN3, SCGB1A1, AREG), which outperformed the conventional literature-derived TR signature across tissue

and tumor type (Revised Fig 3e and Extended Data Fig 5.). Expression of CCL4L2, encoding a chemokine which induces chemotaxis of CCR5 and CCR1-expressing cells, such as T cells, dendritic cells and macrophages, has previously been described in NK cells isolated from melanoma samples (ref 63). This represents an independent verification, as this dataset was not included in our study. These melanoma-infiltrating NK cells also exhibited high AREG expression, an EGF receptor ligand. Notably, upregulation of AREG has also been described in the setting of healthy and cirrhotic liver-resident NK cells (ref 64), a tissue type not included in our pan-cancer atlas. Intriguingly, SCGB1A1, a member of the secretoglobulin family, functions as a potent inhibitor of phospholipase A2 (ref 65), a well described immunosuppressive molecule contributing to development of the TME. Hence, it is tempting to speculate that secretion of the SCGB1A1-encoded protein could be another effector mechanism through which TiNK cells can positively contribute to remodeling of the TME.

2Bii: The authors conclude from Figures 3E and 3F that there is a high degree of heterogeneity within tissues, but a conservation of either a CD56^{bright} or CD56^{dim} peripheral identity, shown by a measure of ‘connectivity’ but no specific gene signature overlap/divergence. The context of tissue NK cells within the total tissue dataset is critical, as placing any cell between the two states (here, CD56^{bright} or CD56^{dim}), will always place it somewhere between one of them. There should be more detailed content on the signature behind the overlap and the divergence between tissue and PB NK cells, and on the signatures across different tissues. An integrative analysis combining all those datasets depicted in Supplementary Figure 3 could not only remove doubts from NK cell type annotation but also provide an extensive comparison of tissue-resident signatures that have been notoriously difficult to define for human NK cells versus ILCs.

Author response: This concern is partially addressed in response to comment 2Bi. We agree with the reviewer regarding the usefulness of expanded analysis and have therefore integrated all immune cells in the resource across tissues. This turned out to be a massive undertaking as it led to us having to redo large parts of the downstream analysis, requiring us to run this on multiple computer clusters. This led to a revised Fig. 3b and d, as described above (before one cancer type, now pan-cancer) and to a deeper analysis of tissue residency (described above and shown in Extended Data Fig. 5e-g). We have also been able to corroborate the NK cell annotation and describe how it differs from ILCs (Extended Data Fig. 5c-d).

We would like to clarify that when integrating the tissue-derived cells with the peripheral blood, the tissue-derived cells are not restricted to be placed between these two states, but we observe that the CD56^{bright} from the tissue place themselves near the CD56^{bright} from peripheral blood and the CD56^{dim} from the tissue place themselves near the CD56^{dim} from peripheral blood. The model that we refer to as our reference model is trained by integration of blood and tissue NK cells. It does not build on our subset-trained model and does not force any cell to be placed between CD56^{bright} or CD56^{dim}. The tissue and blood data are treated the exact same way. The fact that both tissue-resident and tumor-infiltrating NK cells retain their overarching bright/dim signature (the two dominant transcriptional signatures in peripheral blood) is important as it allows us to specifically delineate how these cells have been perturbed from their presumed baseline signatures.

2Biii: Finally, bearing in mind that this dataset is comprised of several donors, across many batches, labs and tissue origins - how is the data integrated, and how was this determined to be the best method of integration and representation?

Author response: The NK data where we leverage the known subset annotations was integrated using scANVI and the bulk NK data was integrated using scVI. The data from individual tumors/tissues was integrated using scANVI, allowing us to harmonize annotations. The integration of all immune cells across all tumor and tissue types was done using scVI. For establishing what we refer to as our normal reference we used scVI for integration. scVI/scANVI was selected based on published benchmarking studies of integration methods, but we did not benchmark this ourselves. We cite the following paper [10.1038/s41592-021-01336-8](https://doi.org/10.1038/s41592-021-01336-8) and described this in the methods section: “The probabilistic models scVI and scANVI as implemented in scvi-tools3 were used for integration of scRNA-seq data. These

methods have been shown to perform well for integration of scRNA-seq data, especially when dealing with complex batch effects and integrating atlas-level data". This article nicely shows that autoencoder-based frameworks such as scVI and scANVI tended to perform better on tasks with more cells and complex batch structure as we are dealing with in our analysis. We used the representations of the data provided by scVI/scANVI for all downstream analysis tasks.

3. Identification of stressed NK cells in tumours should be further validated

The presence of a 'stressed' NK population (neighbourhood 1 in figure 5A) could certainly be an interesting finding; however, several steps are needed to validate that this is not an artefactual finding (for example those found in 10.1038/s41593-022-01022-8):

i. Are count matrices corrected for ambient RNA removal with a tool like CellBender, soupX or decontX?

Author response: This is a very important point that has been carefully addressed during revision at multiple levels. We ran decontX, one of the tools suggested, to adjust the count matrices for ambient RNA. decontX can run without empty droplet-data, allowing us to adjust the matrices also for the data where only the filtered count matrices have been made available. We adjusted the matrices for each sample for all the tissue and tumor types. Thereafter the whole atlas was rebuilt, and all downstream analyses were rerun (CellChat, annotation, GRN analysis, Milo and survival analyses). The biggest differences between the adjusted and non-adjusted matrices were seen in ribosomal and mitochondrial genes and there was no impact on the main results. The online methods file has been updated to describe these additional steps.

ii. Are tissue dissociation methods for skin different from those for solid tumours? The distribution of SKCM (Fig 3H) suggests that these NK cells either have a highly-specific signature, which will skew the representation and work against the method milo attempts to use, or that during dissociation of skin samples, a signature is generated which is highly different from other cancer types - both entirely plausible possibilities

iii. Previous single cell RNA sequencing papers describe cell populations with up-regulation of genes for heat shock proteins (10.4110/in.2020.20.e34) and several reports even suggest dissociation methods which minimise these effects, specifically for tumours (10.1186/s13059-019-1830-0).

Author response: We scored each sample for the dissociation related stress signature identified in one of the papers cited by reviewer #2 (10.1186/s13059-019-1830-0). From analysis of these scores, we found no evidence that individual samples or tissue/tumor types are particularly affected by upstream processing of tissues. Specifically, we did not see a particularly high expression of these signatures in the melanoma and skin samples as opposed to the other tumor/tissue types. These samples also don't have different dissociation methods compared to the other tumor/tissue types. Directly regressing out these dissociation related stress signatures, which partially captures a general stress signature, can be problematic since we risk removing biologically relevant signals. Neuschulz et al. (10.15252/msb.202211147) notes that: "While it should be possible to regress out dissociation response from scRNA-seq datasets based on [dissociation related stress signatures], we wish to note that this can be potentially problematic, since cellular stress response may not only be due to dissociation but can also be caused by biological factors." We therefore have not regressed out dissociation response before performing the downstream analysis downstream of removing ambient RNA. Other indirect validation of the stress signature includes a deeper analysis of the individual programs, including metabolism, ROS, hypoxia etc. (Revised Fig. 5). We have also been able to identify stressed group 1 NK cells in spatial transcriptomics data, not subject to any upstream dissociation/digestion (New Fig. 7).

4. Relevance for tumour immunology

The focus of NK cells in tumours and tissues is the particular strength of this paper, and even with other criticisms above it should not be lost how valuable this analysis is. However, the final set of figures in 6B-E about cell interaction analysis remains the weakest part of this paper, particularly as

it is a very straightforward piece of analysis within one R package and could be expanded. The clarity of each of these plots' accuracy is questionable, as each ligand/receptor is measured by an arbitrary 'strength', and no validation is given of any of these interactions. If one of these pathways were validated in vitro, it would lend significant strength to the study.

Author response: We thank the reviewer for this positive remark and believe the resource have improved further through addressing the many constructive comments and suggestions, not the least the upstream removal of ambient RNA and the integration of all immune cells across all tissues and tumors. With regards to the limitations of CellChat, we agree with the reviewer that the inferred interactions have limited value without functional validation. In the revised manuscript we moved the CellChat analysis to a separate descriptive figure (New Fig. 6) and scored the major incoming and outgoing signals to/from the six NK cell states across ALL tumor types. In terms of validating incoming signals, we choose to look closer at the NKG2A/HLA-E pathway in the context of anti-NKG2A (biosimilar of monalizumab) therapy as suggested by Reviewer 1. This was one of the dominating incoming signals to CD56^{bright} TiNK cells in the CellChat analysis across all tumor types (Revised Fig. 6). Although, the NKG2A-HLA-E checkpoint is well described, its role in regulating CD56^{bright} NK cell responses and the potential reversal of this by therapeutic anti-NKG2A has not been formally addressed. We therefore performed functional experiments on PB-NK cells and show that degranulation and cytokine production by CD56^{bright} NK cells is shut down by IFN-treatment of lung cancer cell lines, resulting in HLA-E upregulation, and efficiently restored by anti-NKG2A treatment. Revised Fig. 6j-k and Extended data Fig. 9). These data are discussed in the context of significant outgoing signals specifically from group 1 CD56^{bright} NK cells in the tumor, releasing IFN- γ (Revised Fig. 6e-g).

Specific points:

The utility of the representation in Figure 3C is unclear. There have been 6 representations of peripheral blood NK cells by this point in the paper. Some more uniformity and clear description in each Figure legend of which data set is used for which representation would be very useful.

Author response: We appreciate that this representation may have been confusing and decided to remove it from the ms. We have also clarified the underlying data used in the various representations in the legend of Fig. 1 and Fig. 3.

In Figure 5A, the authors utilise their final model to understand and interrogate inter-tissue and inter-tumour differences by using milo to detect neighbourhoods of cells in a cluster-free fashion. This method works by comparing intra-neighbourhood differences, instead of against each other, and is highly dependent on the data's preparation, including normalisation, variable features, representation method and number of neighbours used.

It is unclear how the authors determined that this is the optimal way to represent this data, which this method is so sensitive to. Would the results be different if they used the diffusion map, tSNE, or scVI representation?

Author response: The neighborhood graph used for Milo is computed using the scVI representation, which is the only representation of the total NK dataset we use for increased consistency. This has now been further clarified in the methods. We avoided tSNE or UMAP representation for this analysis (or any other analysis, expect for visualization) as this representation can lead to distortions of high dimensional data (10.1371/journal.pcbi.1011288).

Figure 6A shows survival curve analysis of 7 cancer types, split by whether they are enriched for signatures of NK cells of neighbourhoods 1 or 3. It is unclear which genes are used to split these samples, such as differential gene expression between group 1 and group 3, or group 1 and everything else, etc. Are these NK cell-specific, or are they found in the wide tissue/tumour dataset?

Author response: We compare the groups using differential expression analysis implemented in Milo and identify top differentially expressed genes. We score these signatures in the inferred NK expression

matrix from BayesPrism. Therefore, it is not directly relevant if they are expressed in other cell types, since this will have no impact on the survival analysis performed here. Although not done here, the resource offers extensive possibilities to extract whole tumor immune ecosystems and look at more dependable effects between states of multiple cell types, such as for example EcoTyper. We plan to do so but this is not feasible within the timeframe of the current revision.

Decision Letter, first revision:

25th Apr 2024

Dear Dr. Malmberg,

Thank you for submitting your revised manuscript "Pan-cancer profiling of tumor-infiltrating natural killer cells through transcriptional reference mapping" (NI-RS36733A). It has now been seen by the original referees and their comments are below. The reviewers find that the paper has improved in revision, and therefore we'll be happy in principle to publish it in Nature Immunology, pending minor revisions to comply with our editorial and formatting guidelines.

We will now perform detailed checks on your paper and will send you a checklist detailing our editorial and formatting requirements in about a week. Please do not upload the final materials and make any revisions until you receive this additional information from us.

If you had not uploaded a Word file for the current version of the manuscript, we will need one before beginning the editing process; please email that to immunology@us.nature.com at your earliest convenience.

Thank you again for your interest in Nature Immunology Please do not hesitate to contact me if you have any questions.

Sincerely,

Jamie D.K. Wilson, D.Phil
Chief Editor
Nature Immunology
212 726 9207
j.wilson@us.nature.com

Reviewer #1 (Remarks to the Author):

I am satisfied with the authors rebuttal and find the manuscript greatly improved. Congratulations to the authors on a excellent body of work

Reviewer #2 (Remarks to the Author):

The authors have addressed convincingly all my criticisms, I have no further comments.

Final Decision Letter:

Dear Dr. Malmberg,

I am delighted to accept your manuscript entitled "Pan-cancer profiling of tumor-infiltrating natural killer cells through transcriptional reference mapping" for publication in an upcoming issue of *Nature Immunology*.

Over the next few weeks, your paper will be copyedited to ensure that it conforms to *Nature Immunology* style. Once your paper is typeset, you will receive an email with a link to choose the appropriate publishing options for your paper and our Author Services team will be in touch regarding any additional information that may be required.

Please note that *Nature Immunology* is a Transformative Journal (TJ). Authors may publish their research with us through the traditional subscription access route or make their paper immediately open access through payment of an article-processing charge (APC). Authors will not be required to make a final decision about access to their article until it has been accepted. Find out more about Transformative Journals.

Authors may need to take specific actions to achieve compliance with funder and institutional open access mandates. If your research is supported by a funder that requires immediate open access (e.g. according to Plan S principles) then you should select the gold OA route, and we will direct you to the compliant route where possible. For authors selecting the subscription publication route, the journal's standard licensing terms will need to be accepted, including self-archiving policies. Those licensing terms will supersede any other terms that the author or any third party may assert apply to any version of the manuscript.

Your paper will be published online soon after we receive your corrections and will appear in print in the next available issue.

You may wish to make your media relations office aware of your accepted publication, in case they

consider it appropriate to organize some internal or external publicity. Once your paper has been scheduled you will receive an email confirming the publication details. This is normally 3-4 working days in advance of publication. If you need additional notice of the date and time of publication, please let the production team know when you receive the proof of your article to ensure there is sufficient time to coordinate. Further information on our embargo policies can be found here: <https://www.nature.com/authors/policies/embargo.html>

Also, if you have any spectacular or outstanding figures or graphics associated with your manuscript - though not necessarily included with your submission - we'd be delighted to consider them as candidates for our cover. Simply send an electronic version (accompanied by a hard copy) to us with a possible cover caption enclosed.

If you have not already done so, we strongly recommend that you upload the step-by-step protocols used in this manuscript to the Protocol Exchange. Protocol Exchange is an open online resource that allows researchers to share their detailed experimental know-how. All uploaded protocols are made freely available, assigned DOIs for ease of citation and fully searchable through nature.com. Protocols can be linked to any publications in which they are used and will be linked to from your article. You can also establish a dedicated page to collect all your lab Protocols. By uploading your Protocols to Protocol Exchange, you are enabling researchers to more readily reproduce or adapt the methodology you use, as well as increasing the visibility of your protocols and papers. Upload your Protocols at www.nature.com/protocolexchange/. Further information can be found at www.nature.com/protocolexchange/about .

Please note that we encourage the authors to self-archive their manuscript (the accepted version before copy editing) in their institutional repository, and in their funders' archives, six months after publication. Nature Portfolio recognizes the efforts of funding bodies to increase access of the research they fund, and strongly encourages authors to participate in such efforts. For information about our editorial policy, including license agreement and author copyright, please visit www.nature.com/ni/about/ed_policies/index.html

Sincerely,

Jamie D.K. Wilson, D.Phil
Chief Editor
Nature Immunology
212 726 9207
j.wilson@us.nature.com